# Statistical inference with a manifold-constrained RNA velocity model uncovers cell cycle speed modulations

Alex R. Lederer ®[1], Maxine Leonardi ®[2,7], Lorenzo Talamanca ®[2,7], Daniil M. Bobrovskiy ®[1], Antonio Herrera ®[1], Colas Droin ®[2], Irina Khven[1], Hugo J. F. Carvalho[2], Alessandro Valente[1], Albert Dominguez Mantes[1,3], Pau Mulet Arabí[2], Luca Pinello ®[4,5,6], Felix Naef ®[2] ✉ & Gioele La Manno ®[1] ✉

Across biological systems, cells undergo coordinated changes in gene expression, resulting in transcriptome dynamics that unfold within a low-dimensional manifold. While low-dimensional dynamics can be extracted using RNA velocity, these algorithms can be fragile and rely on heuristics lacking statistical control. Moreover, the estimated vector field is not dynamically consistent with the traversed gene expression manifold. To address these challenges, we introduce a Bayesian model of RNA velocity that couples velocity field and manifold estimation in a reformulated, unified framework, identifying the parameters of an explicit dynamical system. Focusing on the cell cycle, we implement VeloCycle to study gene regulation dynamics on one-dimensional periodic manifolds and validate its ability to infer cell cycle periods using live imaging. We also apply VeloCycle to reveal speed differences in regionally defined progenitors and Perturb-seq gene knockdowns. Overall, VeloCycle expands the single-cell RNA sequencing analysis toolkit with a modular and statistically consistent RNA velocity inference framework.

Single-cell RNA sequencing (scRNA-seq) captures a static snapshot of gene expression in a destructive manner, making it difficult to interpret dynamical aspects of biological processes. To address this issue, computational approaches have emerged that reconstruct temporal information among cellular states from scRNA-seq data[1]. For example, RNA velocity exploits the ratio between unspliced and spliced transcripts to estimate a vector that describes the rate of change of gene expression[2]. The model considers a system of first-order ordinary differential equations describing the mRNA life cycle and whose key parameters are splicing and degradation rates. Under simplified assumptions, it is possible to estimate these parameters from data[3].

The original RNA velocity framework, implemented in velocyto, fixes a common splicing rate across genes to infer a relative gene-dependent degradation rate from spliced–unspliced phase portraits[2]. This parameter is then plugged into the differential equations to obtain a gene-specific velocity. An extended model for the estimation

[1]Laboratory of Brain Development and Biological Data Science, Brain Mind Institute, Faculty of Life Sciences, École Polytechnique Fédérale de Lausanne (EPFL), Lausanne, Switzerland. [2]Laboratory of Computational and Systems Biology, Institute of Bioengineering, Faculty of Life Sciences, École Polytechnique Fédérale de Lausanne (EPFL), Lausanne, Switzerland. [3]Laboratory of Bioimage Analysis and Computational Microscopy, Institute of Bioengineering, Faculty of Life Sciences, École Polytechnique Fédérale de Lausanne (EPFL), Lausanne, Switzerland. [4]Molecular Pathology Unit, Massachusetts General Research Institute, Charlestown, MA, USA. [5]Massachusetts General Hospital Cancer Center, Harvard Medical School, Charlestown, MA, USA. [6]Broad Institute of MIT and Harvard, Cambridge, MA, USA. [7]These authors contributed equally: Maxine Leonardi, Lorenzo Talamanca. ✉e-mail: felix.naef@epfl.ch; gioele.lamanno@epfl.ch

of RNA velocity is the 'dynamical model', implemented for the first time in the tool scvelo, which introduced for each gene a cell-wise latent time to support the estimation of kinetic parameters varying across a pseudotemporal axis, making them directly identifiable[4]. By exploiting expectation-maximization, scvelo estimates latent time and kinetic parameters. Other methods have harnessed these modeling ideas or worked toward extending them[5–14]; however, RNA velocity analysis remains highly sensitive to preprocessing choices and requires various heuristics to obtain the final estimates.

A pervasive yet potentially dangerous heuristic is the nearest-neighbor smoothing used to approximate expectations on the RNA counts; this procedure can let information bleed from some genes to others and cause distortions[15]. Additionally, the use of general nonlinear dimensionality reduction techniques to bring the high-dimensional velocity vector onto a two-dimensional embedding (for example, Uniform Manifold Approximation and Projection (UMAP) and *t*-distributed stochastic neighbor embedding (*t*-SNE)) risks introducing artifacts[16]. For instance, velocities associated with orthogonal processes, such as proliferation and differentiation, may be blended together, and adjacent yet unrelated cell populations might affect the resulting vector. Other algorithmic steps and corner cases that typically require attention have already been noted[2,17]; single-cell metabolic-labeling measurements methods can solve some of these problems[8], but their applicability is limited to specific experimental designs and in vitro settings.

However, a seldom discussed limitation of some of the earliest and still commonly used RNA velocity models[2,4] is their reliance on the gene-wise fit kinetic parameters and velocities. In this setting, even when global reconciliation is sought post hoc, the estimated kinetic parameters remain independent; this leads to a physically and geometrically inconsistent velocity vector, whose gene-specific components are on different time scales and whose resulting direction is not necessarily tangent to the low-dimensional manifold cells traverse. Therefore, it is desirable to perform a joint gene fit to regularize the estimates, a strategy introduced by recent methods[18–20] where the manifold, nonconstant kinetic parameters and velocities are all the output of a nonlinear function (for example, a neural network) with a shared latent representation. Yet, this unstructured interdependence does not fully control the information flow from data to estimates and makes it difficult to understand in which way regularization is applied.

Finally, the lack of established ground truths for RNA velocity limits the rigorousness of sensitivity analyses that can be performed on newly developed methods, creating a challenging environment to benchmark advanced extensions[18,21–23]. In particular, overparameterization becomes a concern, especially for models with less stringent assumptions, several nonlinearities or many degrees of freedom. Furthermore, proposed Bayesian formulations of the 'dynamical model' return a high-dimensional mean-field posterior, which is not consistent with the assumption of low-rank dynamics and is poorly suited to inference on the velocity and statistical comparisons of cell population dynamics.

We addressed these challenges by reformulating RNA velocity analysis as an inferential framework rooted in a manifold-constrained probabilistic model. Adopting this approach, we propose an explicit parametrization of RNA velocity as a field defined on the manifold coordinates. We focus on one-dimensional (1D) periodic manifolds in a framework called VeloCycle, enabling model validation and application to cell cycle dynamics. The cell cycle is the most ubiquitous periodic process in biology and plays a fundamental role in embryonic development, tissue regeneration and disease[24,25]. Despite being pervasive in scRNA-seq datasets, default cell cycle analysis pipelines[26,27] are still restricted to categorical phase assignment based on a small selection of marker genes[28–30]. In this work, we not only tackle the broader issue of maintaining geometrical constraints during velocity estimation, but we also make strides in improving cell cycle analysis in scRNA-seq

data, highlighting its continuous nature and providing control over the actual biological time scales. We apply VeloCycle across different biological contexts, experimentally benchmark against time-lapse microscopy measurements and illustrate the ability to perform statistical tests.

## Results

### Manifold-constrained RNA velocity addresses shortcomings

We first sought to redesign RNA velocity estimation by unifying manifold and velocity inference into a single probabilistic framework (Fig. 1a, left). This framework is articulated around a generative model with explicit low-dimensional dynamics at its core. In our model, cells move in time as points on a low-dimensional manifold *x* embedded within the space of all measured genes. Spliced and unspliced molecules are taken only as a function of *x* (*s(x)* and *u(x)*). Then, by parameterizing the velocity vector field as an autonomous function of the manifold coordinates *V(x)*, we constrain RNA velocity vectors to lie tangent to the manifold (Fig. 1a, right). This is contrary to previous approaches, where velocity direction is unconstrained and the result of gene-wise estimates[15–17] (Fig. 1b). We take the derivative of the expected spliced counts, apply the chain rule and plug in the kinetic equations to obtain a velocity vector field interlocking the kinetic parameters of all genes and the dynamics of the latent coordinates (Methods). Noise in the measured raw read counts is modeled as a negative binomial and also as a function of the manifold. Biochemically informed priors are chosen for all other parameters, including splicing (*β*) and degradation (*γ*) rates for each gene (Fig. 1c and Methods).

This formulation constitutes a latent variable framework for estimation of the gene expression manifold and RNA velocity. The choice of a specific dimensionality, topology and associated functional parametrization constraining its geometry can be tailored in an application-specific manner (Fig. 1d; Discussion). We propose inference in two statistical learning procedures: (1) manifold learning to jointly learn the parameters defining the geometry of the gene expression space and assign each cell a manifold (latent) coordinate; and (2) velocity learning to find a velocity field and kinetic parameters, conditioned on the manifold geometry and cell coordinates (Fig. 1d,e).

We implemented this scheme considering a scenario where the prior information on manifold topology is strong: the cell cycle, a 1D periodic space on which gene expression varies smoothly and can be parametrized using a Fourier series. Our framework, VeloCycle, constitutes a generative probabilistic model with two groups of latent variables and is solved in Pyro[31] (Methods). The first group relates to manifold learning and defines the low-dimensional manifold *x*, parameterized as cell cycle phase (*φ*) and gene-specific Fourier coefficients ($v_0$, $v_{1sin}$ and $v_{1cos}$) using the expected spliced counts as a function of the phase (Fig. 1e and Extended Data Fig. 1a,b). The second group relates to velocity learning from the expected unspliced counts and includes the gene-specific degradation rates ($γ_g$), effective splicing rates ($β_g$) and velocity harmonic coefficients (*vω*), which parameterize an angular speed function $ω(φ)$ describing how cell cycle velocity changes along the manifold (*φ*) (Fig. 1e and Extended Data Fig. 1c,d; Methods). Using stochastic variational inference (SVI), VeloCycle returns the joint posterior probability of the latent variables, which can be used to (1) perform statistical velocity significance testing; (2) characterize underlying correlations between the uncertainty of latent variables; (3) estimate cell cycle velocities on a biologically relevant time scale; and (4) facilitate the application of velocity to small datasets by transfer learning (Fig. 1f).

### Sensitivity analysis on simulated data validates VeloCycle

After designing our model, we sought to evaluate its performance on simulated data, as no real dataset is endowed with ground-truth information for phases, speed and RNA kinetic parameters. We employed a simulation intended to preserve important relations expected in

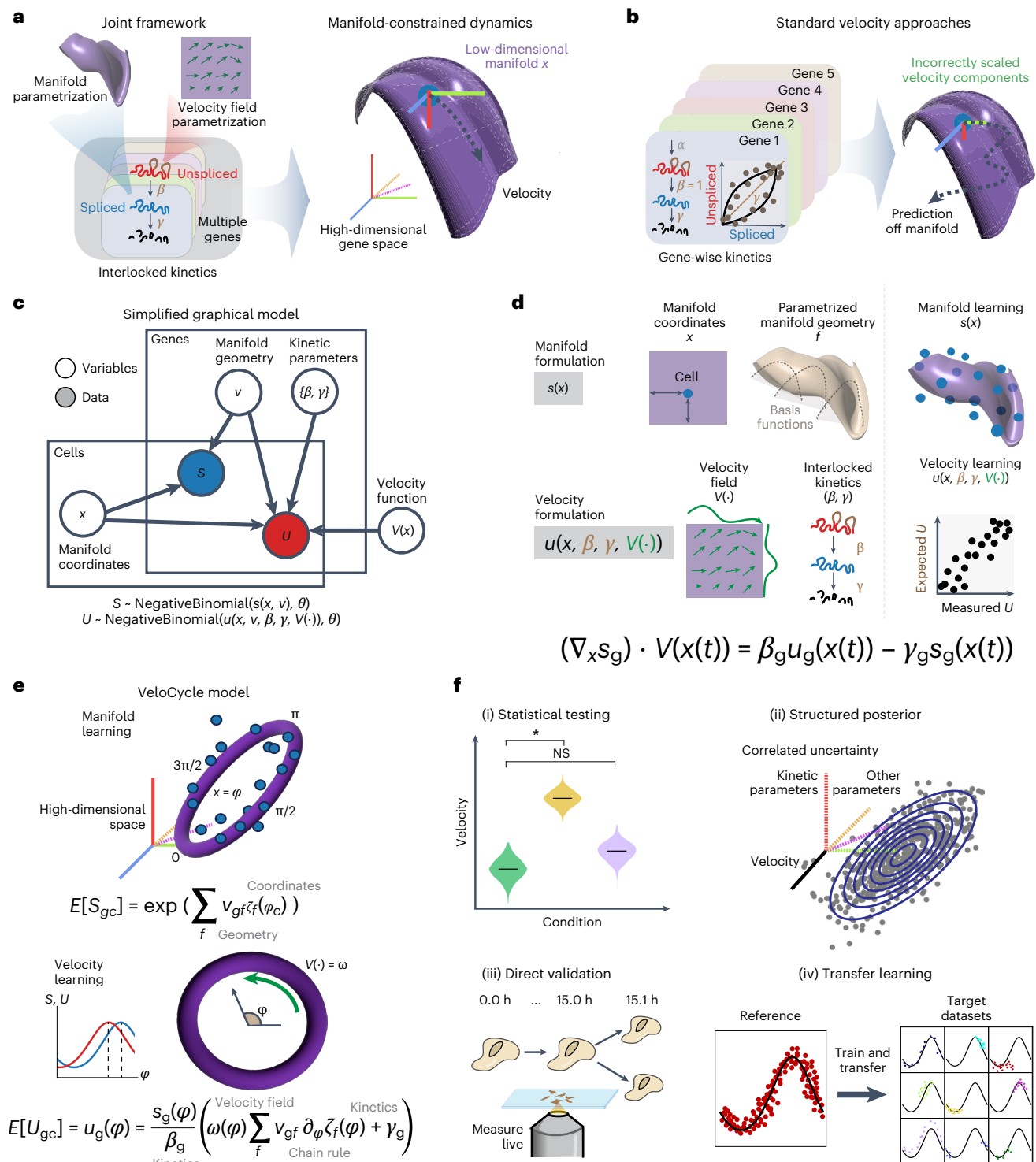

**Fig. 1 | Statistical inference of RNA velocity with a manifold-constrained framework for the cell cycle. a**, Schematic of a joint framework for parameterization of the gene expression manifold and RNA velocity field. **b**, Schematic of unconstrained velocity estimation described by standard approaches. **c**, Plate diagram of the probabilistic relationship among latent variables and observable data. $S$ is sampled from the expectation, manifold coordinates and manifold geometry. $U$ is sampled from the manifold information, kinetic parameters and velocity function. Coordinates define each cell's position on the latent space and geometry defines expression changes along the manifold. **d**, Manifold formulation is defined for the spliced counts ($s$) using cell-specific coordinates ($x$) and a gene-specific geometric family ($f$), with which observed data can be directly mapped to the high-dimensional space (top). Bottom: velocity formulation is defined for unspliced counts ($u$)

as a velocity field function ($V$) with interlocked kinetic parameters ($\beta$, $\gamma$). We obtain a velocity estimate by taking the chain rule over these entities, describing velocity as a direct function of the manifold $x(t)$. **e**, Schematic of manifold-constrained velocity estimation for periodic processes. First, manifold learning estimates the coordinates and geometry; second, velocity learning estimates the kinetic parameters and velocity function. **f**, Schematic of the new types of velocity analyses possible with VeloCycle: (i) statistical credibility testing between multiple samples and against a null hypothesis; (ii) posterior marginal distribution analysis of model parameters by MCMC sampling; (iii) velocity extrapolation to real biological time, verifiable by live microscopy; and (iv) transfer learning of the gene manifold from large references to small target datasets. The asterisk indicates statistical significance. NS, not significant.

real data[2] and avoid biologically improbable scenarios (Methods and Extended Data Fig. 2a–c). Specifically, we incorporated positive correlations among the splicing rates, degradation rates and baseline (mean) expression levels ($r = 0.30$), which helps ensure biological plausibility and avoid unrealistic parameter configurations that diverge from empirical observations (Extended Data Fig. 2a). This structure naturally imposed a positive correlation between the splicing rate and total spliced counts as well as a negative correlation between the splicing rate and total unspliced counts (Extended Data Fig. 2b,c).

First, we evaluated manifold learning across 20 individually simulated datasets each containing 3,000 cells and 300 genes and found VeloCycle inferred phases that closely matched the ground truth, with a circular correlation of $r_\varphi = 0.95$ (Fig. 2a,b). The estimation error was consistently smaller than the uncertainty defined by the posterior, with true values falling within the 5–95% credible interval for 99.2% of cells (Extended Data Fig. 2d). We also verified that the gene-specific Fourier series coefficients closely tracked the original ground truths ($r_{v0} = 0.95$, $r_{v1sin} = 0.98$ and $r_{v1cos} = 0.98$) (Fig. 2c and Extended Data Fig. 2e). For these parameters, wider credible intervals corresponded to more noisy genes with a larger coefficient of variation (Extended Data Fig. 2f). These results confirmed that VeloCycle correctly identified the manifold geometry and cell coordinates.

To assess robustness of the model on different dataset sizes, we performed sensitivity analysis, varying the number of cells and genes (Methods). We found that estimates were broadly accurate, with a circular correlation coefficient greater than 0.70 obtained using as few as 100 cells or 100 genes (Fig. 2d). We further benchmarked our inference against DeepCycle, a recent autoencoder-based method[32]. This comparison showed that VeloCycle was typically more accurate (60% lower mean squared error (MSE) on average, $r_\varphi = 0.95$) than Deep-Cycle ($r_\varphi = 0.73$), despite the latter using velocity moments to achieve its estimations (Fig. 2e–h).

Next, we conditioned VeloCycle on the simulated phase and gene harmonics to assess velocity learning. We observed accurate estimation of gene-wise kinetic parameters across 20 individually simulated datasets, with a particularly close match of degradation-splicing rate ratios to the ground truth ($r_{\gamma/\beta} = 0.997$, $r_\beta = 0.918$, $r_\gamma = 0.617$; Fig. 2i and Extended Data Fig. 2g,h). Of note, VeloCycle was capable of returning an accurate estimate of the mean angular velocity (percent error running 5.4–22.6%; Fig. 2j). VeloCycle recovered the biological correlation structure among estimated kinetic parameters and total counts without imposing them in the model formulation (Fig. 2k and Extended Data Fig. 2a,b).

We performed sensitivity analysis to understand how the estimations behaved at different ground-truth velocities. We considered a large span of cell cycle velocities fully encompassing the range of biologically plausible ones (16 values from 0 to 1.5 radians per mean half-life (rpmh), four simulations each). The results highlighted stable method performance, with estimates 0.2–35.8% away from the ground truth (Fig. 2l,m). Error increased at slower velocities, with a lower Pearson's correlation between kinetic parameters and ground truths (Extended Data Fig. 2i, left). Indeed, slower velocities corresponded to shorter delays between unspliced and spliced RNAs (Fig. 2n and Methods), which are more difficult to characterize accurately. In all simulations, the degradation-splicing rate ratios almost perfectly matched the ground truth (mean $r_{\gamma/\beta} = 0.99$) (Extended Data Fig. 2i, right). Finally, we investigated whether velocity learning performance was affected by dataset size. We detected a dependence on the number of cells and genes, with the highest accuracy and tightest posterior ranges obtained on larger datasets; however, using more cells could compensate for fewer genes, and vice versa (Fig. 2o and Extended Data Fig. 2j). We established 500 cells (and a minimum of 50 genes) or 350 genes (and a minimum of 50 cells) as the lower limits of accurate velocity estimation.

Finally, we assessed the impact of increasing fractions of simulated noncycling cells (that is, between 0 and 200 noncycling cells per 100 cycling cells) on manifold learning, obtaining a circular correlation greater than 0.70 in mixed populations containing up to 50 noncycling cells per 100 cycling cells (Extended Data Fig. 2k). Velocity estimates also remained within 25% of the ground truth using up to 50 noncycling cells per 100 cycling cells (Extended Data Fig. 2l).

## Manifold learning robustly estimates accurate phases

After validating on simulated data, we deployed VeloCycle on real datasets produced with different scRNA-seq chemistries. We reasoned that access to a cell cycle phase ground truth, even if categorical (for example, G1, S and G2/M), would facilitate the evaluation of our phase assignments. Thus, we performed manifold learning on a Smart-seq2 dataset of fluorescent ubiquitination-based cell cycle indicator (FUCCI) system-transduced mouse embryonic stem (mES) cells that were index-sorted using fluorescence-activated cell sorting (FACS)[33]. We fit the cell cycle phase on spliced counts using a gene set representing a broad Gene Ontology (GO) query[34] (Methods) and evaluated the results against FUCCI-FACS categories. Cells belonging to the same category were assigned to similar phases (Fig. 3a,b); a classifier based on two thresholds and trained on VeloCycle phases achieved 82.7% accuracy in predicting the annotations, almost matching the 87.8% accuracy obtained when training a logistic classifier on all genes (Fig. 3c). Furthermore, gene fits underlying manifold learning closely replicated the expected sequential patterns of cell cycle genes. Among fits of high confidence were early-peaking histone acetylase *Hat1*, followed by transcription factor *Trp53*, and the anaphase-promoting complex

---

**Fig. 2 | Sensitivity analysis of VeloCycle on simulated data. a**, Scatterplot of cell cycle phase assignment (estimated) compared to the simulated ground truth (GT). **b**, Box plot of circular correlation coefficients (min, 0.932; max, 0.963; median, 0.957) between estimated and GT phases. **c**, Scatterplots of estimated and GT values for gene harmonic (Fourier) coefficients ($v_0$, $v_{1sin}$, $v_{1cos}$) using the dataset in **a**. **d**, Heatmap of the mean circular correlation coefficient between estimated and GT phases computed with varying numbers of cells and genes (average of three simulations). **e**, Scatterplot of cell cycle phase estimation obtained by DeepCycle compared to GT using the dataset in **a**. **f**, Box plot of circular correlation coefficients (min, 0.416; max, 0.851; median, 0.788) between DeepCycle-estimated and GT phases across the datasets in **b**. **g**, Box plots of per-cell MSE for phase estimation with VeloCycle (min, 0.22; max, 0.29; median, 0.23) and DeepCycle (min, 0.43; max, 1.03; median, 0.53) across 20 simulations. **h**, Polar plots representing the phase difference between estimated (Est.) and simulated GT for 30 randomly chosen cells from one simulated dataset using VeloCycle (left) and DeepCycle (right). Each dot represents a cell, and lines connect the estimated phase assignment (light gray) to simulated GT (dark gray). **i**, Scatterplot of estimated kinetic ratio compared to simulated GT for 300 genes.

**j**, Box plot of percent error (min, 2.0; max, 23.0; median, 14.5) between estimated and GT velocity ($\omega = 0.4$). **k**, Scatterplots illustrating the recovered relationships among splicing rate ($\log\beta_g$), degradation rate ($\log\gamma_g$), spliced counts and unspliced counts for 300 simulated genes. E[log l, Top: scatterplot of estimated and GT estimates for 16 different simulated velocities between 0.0 to 1.5 rpmh for four simulations. Bottom: point plots with s.d. of posterior uncertainty intervals corresponding to above simulations. The black dot for each plot represents the mean posterior interval across four simulations; error bands indicate 1 × s.d. **m**, Scatterplot of percent error between estimated and GT velocity across conditions in **l. n**, Scatterplot of mean unspliced–spliced expression delay across conditions in **l. o**, Sensitivity analysis heatmap of the range among velocity estimates for three independent simulations, using varying numbers of cells and genes. The text value in each box represents the mean velocity over the three datasets and heatmap intensity represents absolute range. The Pearson's correlation coefficient ($r$) over 20 individual simulated datasets is indicated in red (**a,c,e,i,k**). Each green dot represents a single gene (**a,e**). Each purple dot represents a single gene (**c,i,k**). Box plot bounds (**b,f,g,j**) are defined by the interquartile range (IQR); whiskers extend each box by 1.5× IQR.

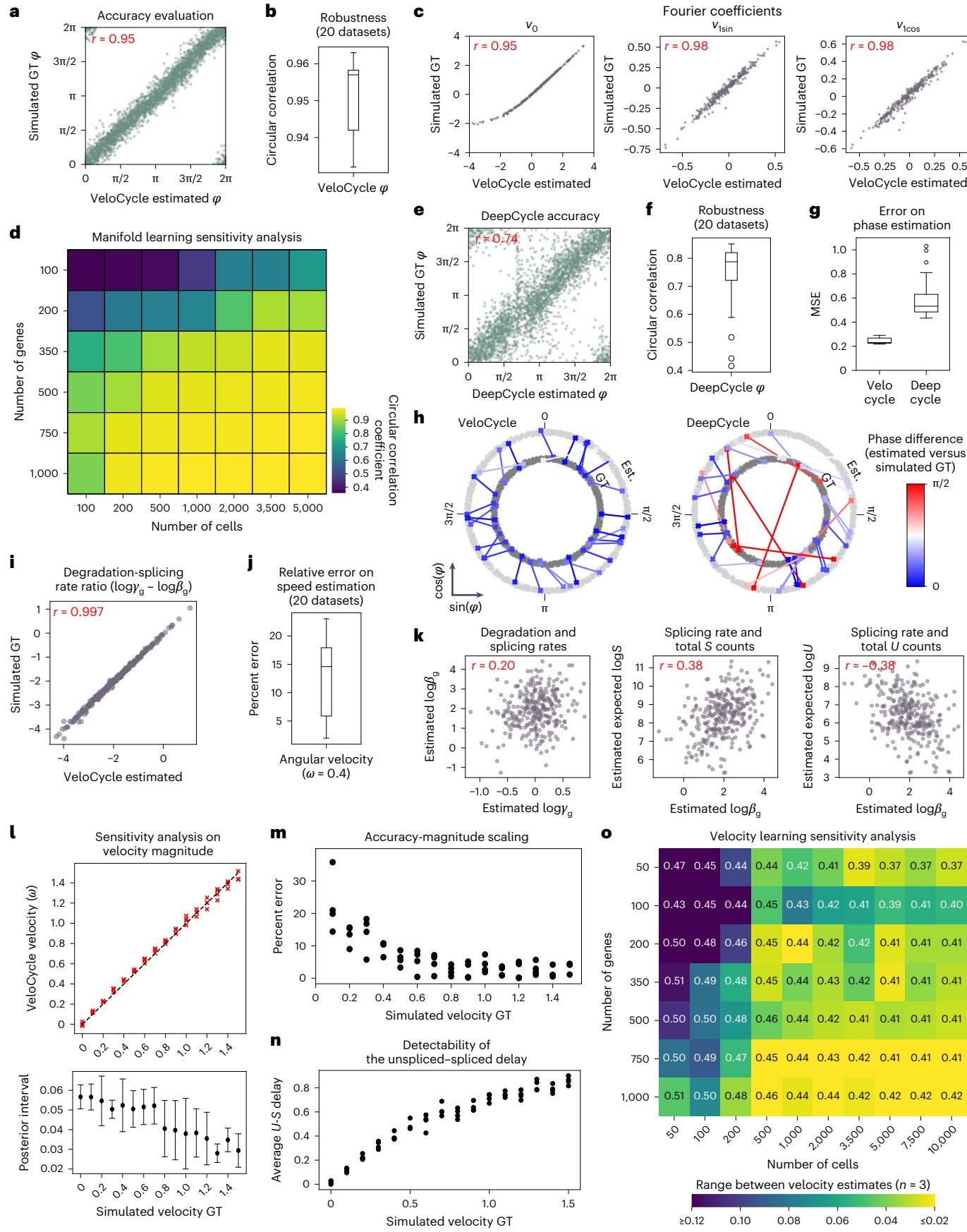

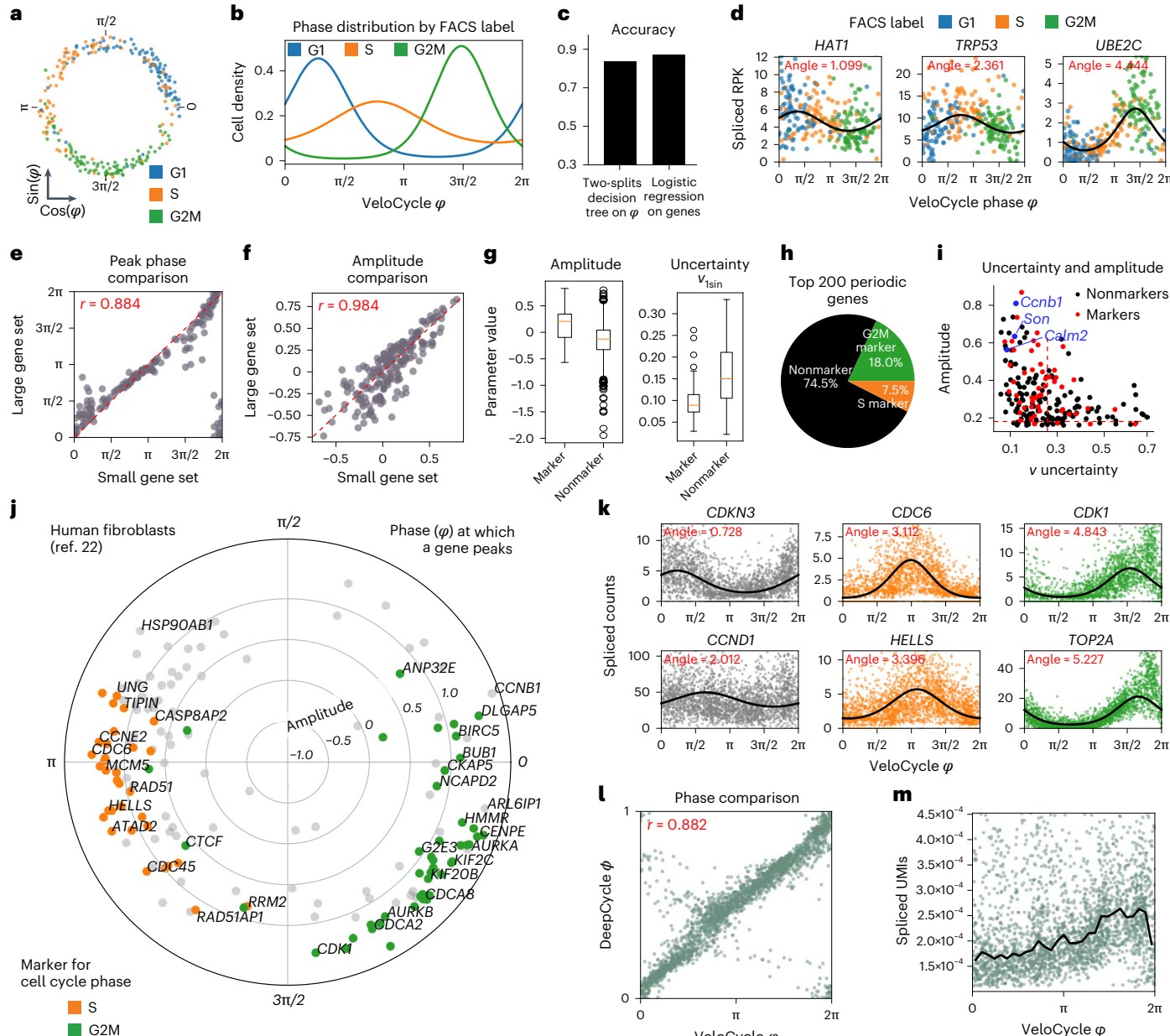

**Fig. 3 | Manifold learning and gene periodicity on different datasets and technologies. a**, Scatterplot of phase assignment for 279 mES cells, colored by FACS-sorted categorical phase[33]. **b**, Density plot for FACS-sorted labels across VeloCycle-assigned phases. **c**, Bar plot reporting categorical phase predictor obtained using a two-threshold decision tree trained on VeloCycle phase estimates alone versus a logistic regression classifier trained on the entire expression matrix. **d**, Representative scatterplots of genes fits. Curved black lines indicate a gene-specific Fourier series obtained with manifold learning. The 'peak' indicates the position of maximum expression along the cell cycle manifold ($\varphi$). **e,f**, Scatterplot of gene-wise peak position (**e**) and amplitude (**f**) using a small ($x$ axis) or large ($y$ axis) gene set during manifold learning. **g**, Left: box plot of gene-wise amplitude for 1,358 marker (min. −0.56; max, 0.84; median, 0.18) and nonmarker (min: −1.5; max; 0.80; median, −0.13) genes. Right: box plot of harmonic coefficient uncertainties for marker (min, 0.03; max, 0.25; median, 0.09) and nonmarker (min, 0.02; max, 0.31; median, 0.15)

genes. **h**, Pie chart of categorical composition for the 200 periodic genes with greatest amplitude. **i**, Scatterplot of gene-wise total harmonic coefficient ($v$) uncertainty and amplitude. Gene dots are colored as standard 'markers' or 'nonmarkers'. Red dashed lines represent mean values for markers. **j**, Polar plot of estimated gene harmonics for human fibroblasts[32]. Each dot represents a gene ($n = 160$). The position along the circle represents the phase of maximum expression, and distance from the center represents total amplitude. Colored genes (orange/green) are those used to compute a standard cell cycle score with scanpy[26] or Seurat[27]. **k**, Selected scatter-plots of genes fits for early (*CDKN3, CCND1*), mid (*CDC6* and *HELLS*) and late (*CDK1* and *TOP2A*) cell cycle markers. **l**, Scatter-plot of VeloCycle-estimated phases compared to DeepCycle. Circular correlation is indicated in red. **m**, Scatter-plot of total raw spliced UMI counts by VeloCycle phase. Black lines indicate binned mean UMI level. Box plot bounds in **b,f,g,j** are defined by the interquartile range (IQR); whiskers extend each box by 1.5× the IQR.

---

member *Ube2c* (Fig. 3d). The gene succession and oscillation amplitude were recapitulated when performing manifold learning on a smaller set of 209 genes, illustrating that the method is effective on chemistries with lower sensitivity (Fig. 3e,f).

Given our Fourier parametrization, we could classify genes by the phase of peak expression, oscillation amplitude and estimation uncertainty (Supplementary Table 1). Inspection of phase–amplitude relationships revealed that marker genes typically used for scoring in

packages such as Seurat and scanpy[26,27] (henceforth 'standard markers') clustered by phase, consistent with the FACS-based ground truth (Fig. 3g,h). Compared to nonmarkers, standard markers on average had a higher amplitude (mean 0.14 versus −0.15) and lower posterior uncertainty (s.d. 0.26 versus 0.43) (Fig. 3g); however, of the top 200 periodic genes based on amplitude, the majority (74.5%) were not standard markers (Fig. 3h) and many ($n = 78$) could be equally or more confidently trusted (tighter posterior probability) as cell phase predictors (Fig. 3i). Among those were calcium-binding protein *Calm2*, splicing cofactor *Son* and cyclin *Ccnb1*, which all play roles in cell proliferation[35–37].

We continued our scrutiny of manifold learning using 10x Chromium data of human fibroblasts (Fig. 3j,k and Supplementary Table 2). To put VeloCycle in relation to other approaches, we compared its estimated phases to those obtained by DeepCycle[32], finding a strong correspondence (human fibroblasts, $r = 0.882$; Fig. 3l). Therefore, VeloCycle accomplishes similar phase estimation to DeepCycle but without using velocity and in tandem with fitting individual gene harmonics. As further validation that correct cell cycle dynamics were captured, we observed a gradual increase in total unique molecular identifiers (UMIs) along the phase, followed by a sharp drop corresponding to cytoplasm partitioning during cytokinesis (Fig. 3m). These results highlight that manifold learning estimates a biologically meaningful 1D geometric space that tracks with the cell cycle across chemistries.

## Unspliced–spliced delays identify cell cycle speeds

We next investigated whether unspliced molecule counts together with the VeloCycle phase are sufficiently informative to estimate cell cycle velocity. To explore this intuitively before performing the full inference, one can extract phases and gene harmonics with manifold learning for unspliced and spliced UMIs independently and use an approximate formula for the velocity that we derived (Methods). We applied this approach on two cultures of human RPE1 cells that were grown in parallel and under identical conditions so that we could also assess robustness by replicate comparison. First, we extracted the phases for each of the datasets by manifold learning and measured the delays (the phase difference) between peak unspliced and spliced expression for each gene[38] (Fig. 4a). We observed consistent and positive delays for the genes (Fig. 4b) that correlated well between replicates ($r = 0.90$; Fig. 4c). We interpreted this correlation as the first evidence that the data contains velocity information, so we proceeded to estimate a cell cycle period with the aforementioned approximate formula. The calculation returned a period 18.5-times the average half-life, which corresponds to 18.5 h, assuming a realistic average half-life of 1 h (Fig. 4d).

In addition to being an approximation, other limitations of the point estimate are that it is not based on a proper noise model and is not associated with an uncertainty measure. To obtain a more accurate estimate and statistical measures of confidence, we learned the complete Bayesian model (velocity learning) on both RPE1 replicates, conditioning on the random variables inferred by manifold learning.

Scaling the obtained velocity by fitted average half-lives yielded average cell cycle periods of 20.1 h ± 0.2 h and 20.0 h ± 0.2 h (mean ± 95% credible intervals) for the two replicates (Fig. 4e and Extended Data Fig. 3a,b). The posterior distributions broadly overlapped (71.2% overlap), indicating no credible velocity difference between replicates. To confirm on real data that VeloCycle can estimate cell cycle speed along a biologically relevant dynamic range, we performed velocity learning on mES cells, a rapidly cycling cell type[32,39]. For this dataset, VeloCycle returned an estimation of 10.5 ± 0.3 average half-life (Extended Data Fig. 3c). As with RPE1 cells, the model recovered kinetic parameters with expected relationships among total UMI counts and gene-specific splicing and degradation rates, as previously observed in simulations (Extended Data Fig. 3d and Fig. 2i). Taken together, these findings confirm VeloCycle can estimate a cell cycle velocity and sample informative posterior distributions.

## A structured distribution accurately models uncertainty

Although we showed our variational formulation recovers accurate estimates of cell cycle phase and velocity in simulated and real data using SVI, it is reasonable to question the limits of a simplified mean-field variational family in representing the structure of joint uncertainty among latent variables. We hypothesized that such a parametrization choice may lead to an overconfidence in the estimated velocity posterior because uncertainties on these latent variables may be inherently correlated (Fig. 4f). A piece of evidence in this direction was the observation that estimates on random gene subsets fell outside the posterior credible interval of the fit on all genes (Fig. 4g). To eliminate this bias toward the underestimation of velocity uncertainty, we decided to characterize the model joint posterior by sampling it with Markov chain Monte Carlo (MCMC; Methods). Using a No U-Turn Sampler, we studied the posterior for human fibroblasts[32] with MCMC, revealing a five-times wider uncertainty compared to mean-field SVI (0.10 rpmh versus 0.02 rpmh; Fig. 4h).

Consistent with our hypothesis, this wider credible interval manifested along with a correlated joint posterior, capturing dependencies among the uncertainty of different latent variables. Examining the posterior, we found samples of the angular speed ($v\omega$) and degradation rate ($\log\gamma_g$) for certain genes that exposed a correlation structure (mean $r = 0.26$; Fig. 4i). Moreover, for each gene we noticed a strong correlation (mean $r = 0.96$) between posterior samples of splicing ($\log\beta_g$) and degradation ($\log\gamma_g$) rates (Fig. 4j). Both features cannot be captured by a mean-field variational distribution.

These findings advocated for a recrafting of our variational distribution to accommodate typical features of the posterior inferred by MCMC, to maintain inferential accuracy but avoid significantly time-consuming sampling procedures. We reformulated our variational distribution with $\log\gamma_g$ and $v\omega$ modeled as a low-rank multivariate normal (LRMN) and with $\log\beta_g$ modeled as a normal conditional on the corresponding $\log\gamma_g$ (Methods). Upon retraining this new SVI + LRMN model, we obtained a velocity estimate with a larger uncertainty range

**Fig. 4 | Analysis of delays, velocity scale, and parameter uncertainties in the choice of variational distribution. a**, Polar plot of peak unspliced–spliced expression for 106 marker genes across two replicates of RPE1 cells analyzed with manifold learning. Genes are colored by their categorical annotation in Cyclebase 3.0 (ref. 38). Unspliced gene fits were inferred separately, conditioned on cell phases obtained when running manifold learning on spliced UMIs. **b**, Histogram of unspliced–spliced delays (in radians). **c**, Scatterplot of unspliced–spliced delays ($r = 0.90$) between replicates from **a**. **d**, Bar plot of cell cycle periods obtained with a first-order-approximate point estimate (Methods). **e**, Posterior estimate plot of constant, scaled cell cycle speed (rpmh) for the two replicates. Black dashed lines indicate a mean of 500 posterior predictions and the colored bar indicates the credibility interval (5th to 95th percentile). **f**, Schematic of the hypothetical scenarios where a gene has uncorrelated (left) and correlated (right) posterior uncertainty between $\log\gamma_g$ and $v\omega$. Blue circles represent the Gaussian kernel distribution density; red lines represent an uncertainty interval between two arbitrary fixed points. **g**, Posterior estimated velocity plot inferred for cultured human fibroblasts[32] using the original SVI mode of VeloCycle and either all genes (left) or random gene subsets (50% of total genes; right). **h**, Violin plots of scaled velocity (in rpmh) after estimation using SVI, MCMC and LRMV (SVI + LRMN) velocity learning models. **i**, Violin plots of Pearson's correlations between the degradation rate ($\log\gamma_g$) and angular speed ($v\omega$) posterior uncertainties across 160 genes. **j**, Violin plots of Pearson's correlations between degradation ($\log\gamma_g$) and splicing ($\log\beta_g$) uncertainties. **k**, Density representation of overlapping $\log\gamma_g$–$v\omega$ posterior distributions between MCMC and either SVI (top) or SVI + LRMN (bottom) for *TOP2A* and *RRM2* (black, MCMC; blue, SVI; red, SVI + LRMN). Kullback–Leibler (KL) divergence scores are in red. Violin plots in **h**–**j** are built from 500 predictive samples; the white line indicates the mean.

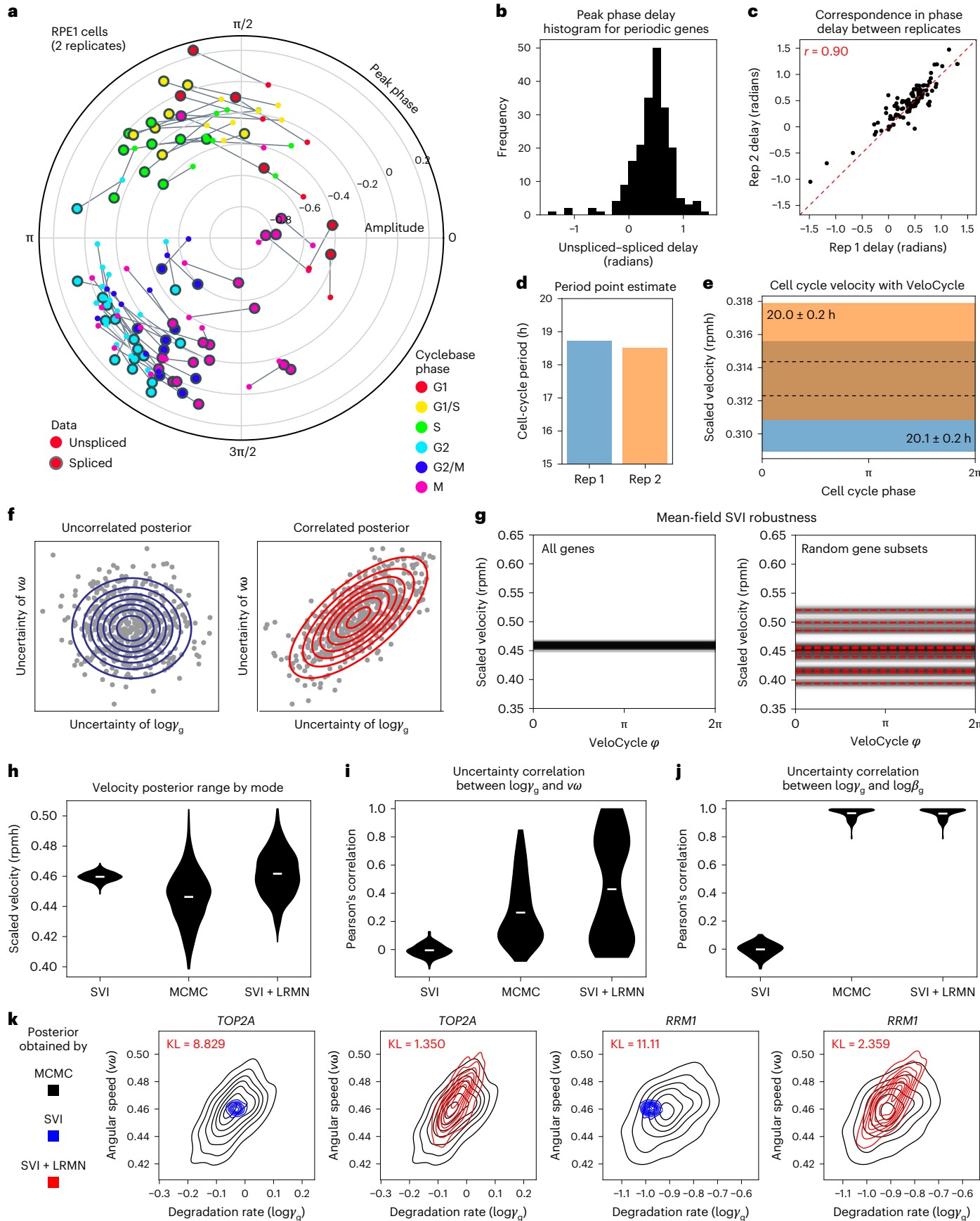

(0.08 rpmh) than with mean-field SVI (Fig. 4h–j). Additionally, we detected a correlation among the SVI + LRMN posterior samples between $\log\gamma_g$ and $v\omega$ for a subset of genes overlapping with the results of MCMC; this resulted in a decreased Kullback–Leibler (KL) divergence between the SVI + LRMN and MCMC posteriors than between the SVI and MCMC posteriors (Fig. 4k and Extended Data Fig. 3f).

Notably, there was a correspondence between the specific genes with high $\log\gamma_g$ and $v\omega$ uncertainty correlation in both SVI + LRMN and MCMC (Extended Data Fig. 3g). Genes with a greater correlation between $\log\gamma_g$ and $v\omega$ tended to be those with larger unspliced–spliced delay (Extended Data Fig. 3h). We speculated the degree of dependence between a gene's $\log\gamma_g$ and $v\omega$ is related to the extent it contributes to the velocity estimate. This was supported by a leave-one-out experiment, where individual genes with smaller degradation rates were those most strongly influencing velocity estimates (Extended Data Fig. 3i). The correlation between $\log\gamma_g$ and $v\omega$ posterior uncertainty was also reproducible when SVI + LRMN was applied to mES cells (Extended Data Fig. 3j,k). Overall, these implementation changes led to generation of a more robust model that can be confidently used for inference, while preserving the underlying correlation structure of the true posterior.

## Cell tracking and labeling validate the inferred velocities

Estimates of a manifold-constrained cell cycle speed with VeloCycle are most conveniently expressed in units of mean half-lives (Methods). Since the average values of half-lives are typically known in many cell types, real-time estimates of RNA velocity can be obtained and validated along the cycle. In this respect, we reasoned that time-lapse microscopy offers a compelling means for comparing VeloCycle estimates to a ground truth.

To benchmark our velocity estimation framework against an experimentally determined cell cycle period, we examined a dataset of dermal human fibroblasts (dHFs) monitored by time-lapse microscopy and for which scRNA-seq data was collected (Methods)[40]. Our SVI + LRMN model inferred a constant cell cycle period of 15.3 ± 1.2 h, assuming an average half-life of the modeled transcripts of 1 h (Fig. 5a and Extended Data Fig. 4a–d). Next, we used VeloCycle to infer a nonconstant (periodic) cell cycle velocity, and we obtained a similar estimated duration of 16.5 ± 2.1 h, with maximal velocity near mitosis ($\varphi$ between approximately $3\pi/2$ and $2\pi$) (Fig. 5b). We then reconstructed the cell cycle period using cellpose and TrackMate for 268 individual cells followed by time-lapse imaging (Fig. 5c)[41,42]. From these data, we recovered a median cell cycle of 15.8 h (s.d. 3.1 h), which overlapped with the posterior credibility interval of the VeloCycle estimate (Fig. 5a,b,d). Comparable results were obtained when using the smaller set of cycling genes[32] (Extended Data Fig. 4e), and results of sensitivity analyses incorporating different fractions of noncycling G0 cells aligned with the moderate robustness observed in simulations (Extended Data Fig. 4f,g). Taken together, these results indicate an ability to obtain comparable cell cycle speed estimates from live imaging and VeloCycle.

We next stratified velocity by an independent categorical cell cycle phase to gain further granularity on these evaluations and model behavior. We observed a faster progression through the cell cycle during the G2/M phase (mean scaled velocity of 0.47 rpmh) compared to a slower progression during G1 (0.37 rpmh) and S (0.36 rpmh) phases (Fig. 5e). Kinetic parameters and their posterior uncertainties were strongly correlated between constant and periodic velocity models (Extended Data Fig. 4h,i). Notably, when estimating the average unspliced–spliced delay for genes peaking at different cell cycle phases, we found cell cycle phases with larger average delays corresponded to regions with faster velocity (Fig. 5f). Genes with larger delays were also those with smaller splicing and degradation rates, which is expected from the approximate model (Extended Data Fig. 4j,k and Methods). After examining the unspliced–spliced delay and the low-rank gene-wise posterior correlation between the angular speed and degradation rate, we could identify specific genes that most strongly contributed to the underlying velocity estimates (Fig. 5g).

To further scrutinize the degree to which cell cycle durations inferred by VeloCycle match those obtained experimentally, we performed time-lapse microscopy and scRNA-seq on the same cultured RPE1 cells. The speed obtained with VeloCycle was approximately 17.7 ± 2.1 h (Fig. 5h and Methods). As in dHFs, this computational estimate overlapped with the mean cell cycle duration of 17.7 h (s.d. of 3.4 h) obtained from tracking dividing cells by time-lapse imaging (338 cells) (Fig. 5i,j). We next sought to compare our cell cycle duration measurements from time-lapse microscopy and VeloCycle to those obtained using an orthogonal experimental technique. Therefore, we performed continuous 5-ethynyl-2′-deoxyuridine (EdU) labeling to independently estimate cell cycle length (Fig. 5k,l). Cycling cells incorporate EdU when they undergo DNA replication during the S phase; thus, the duration of

**Fig. 5 | Validation of computationally inferred velocities by cell tracking and labeling experiments. a,b**, Posterior estimate plot of constant (**a**) and periodic (**b**) cell cycle speed in dHFs[40]. **c**, Top: schematic of time-lapse microscopy to track consecutive cell divisions. Bottom: example images at multiple time points to illustrate tracking a single segmented dHF (pink) through two divisions. Following division of the mother cell (16:40 h), one daughter cell (indicated by white arrow) is tracked for 15 h until dividing itself (31:40 h). **d**, Histogram of cell cycle period for 282 dHFs tracked by live imaging. **e**, Violin plot of dHF cell cycle speed, stratified by categorical phase assignment (G1, 514 cells; S, 383 cells; G2M, 325 cells). Median velocities are indicated by black lines (G1, 0.35; S, 0.37; G2M, 0.48). **f**, Dual-axis plot of the correspondence between unspliced–spliced (U–S) expression delay (left) and velocity (right). Left: genes were grouped by phase into 20 equal bins to calculate unspliced–spliced delay. The solid red line indicates binned mean delay; red bands indicate one standard deviation. Right: scaled velocity estimate from **b**. Bottom: categorical phase assignment probability. **g**, Gene expression scatterplots for genes peaking in S and M phases. Vertical lines correspond to the peak phase of spliced (blue) and unspliced (red) counts. **h**, Posterior estimate plot of periodic cell cycle speed in RPE1 cells. **i**, Images tracking a single RPE1 cell from birth (3:20 h) to subsequent division (20:00 h). **j**, Histogram of cell cycle period for 337 RPE1 cells tracked by live imaging. **k**, Diagram of the cumulative EdU/p21 experiment. Cells were continuously exposed to EdU, fixed at different time points and subjected to EdU detection and p21 immunostaining. **l**, Left: images of p21 (green), 4,6-diamidino-2-phenylindole (DAPI) (cyan) and EdU (magenta) staining after cumulative EdU labeling for 2 h, 8 h and 36 h (representative from one of three experimental replicates).

Scale bar, 100 μm. Right: images of individual cells with different staining combinations. Scale bar, 10 μm. **m**, Schematic of cumulative EdU labeling during cell cycle progression. **n**, Dot plot representing the average percentage of p21+ cells along the different time points. The black horizontal line indicates the mean (min, 0.24, max, 0.76, mean, 0.52), with an error bar for s.d.; each dot represents the percentage of p21+ cells for a single replicate ($n = 29$; a total of three replicates with ten, ten and nine time points). **o**, Top: line plot of the fraction of EdU+ cells at 13 time points (from 30 min to 73 h). Data show the mean of three replicates (except for 2 h, which is from two) and error bars indicate the s.d. Bottom: line plot of fraction of EdU-positive cells among quiescent cells (p21+) as a function of time. A, $x$ value at the intersection between growth and plateau; B, $y$ intercept of the linear fit; GF, $y$ value of the plateau. **p**, Illustration of scEU-seq[43] experimental design, which generates 24 tables used by Dynamo[8] to produce a gold standard cell cycle period estimate. RFP, red fluorescent protein; GFP, green fluorescent protein; RPEs, retinal pigmented epithelium. **q**, Left: schematic of the different experimental measurements, manifold and cell path inference approaches taken by VeloCycle, Dynamo (without metabolically labeled information) and Dynamo-Metabolic (with metabolically labeled information) models. Right: plot showing the estimated cell cycle period obtained by VeloCycle, Dynamo and Dynamo-Metabolic models. The violin plot displays the posterior distribution output by VeloCycle and the circles are individual evaluations of the LAP from different start/end cells; red stars indicate the means. The red dashed line indicates the median in **d** and **j**. The white dashed line indicates the mean of 500 posterior predictions and the black bar indicates the credibility interval (5th to 95th percentile) in **a**,**b**,**f** (right) and **h**. NS, not significant; **$P < 0.01$.

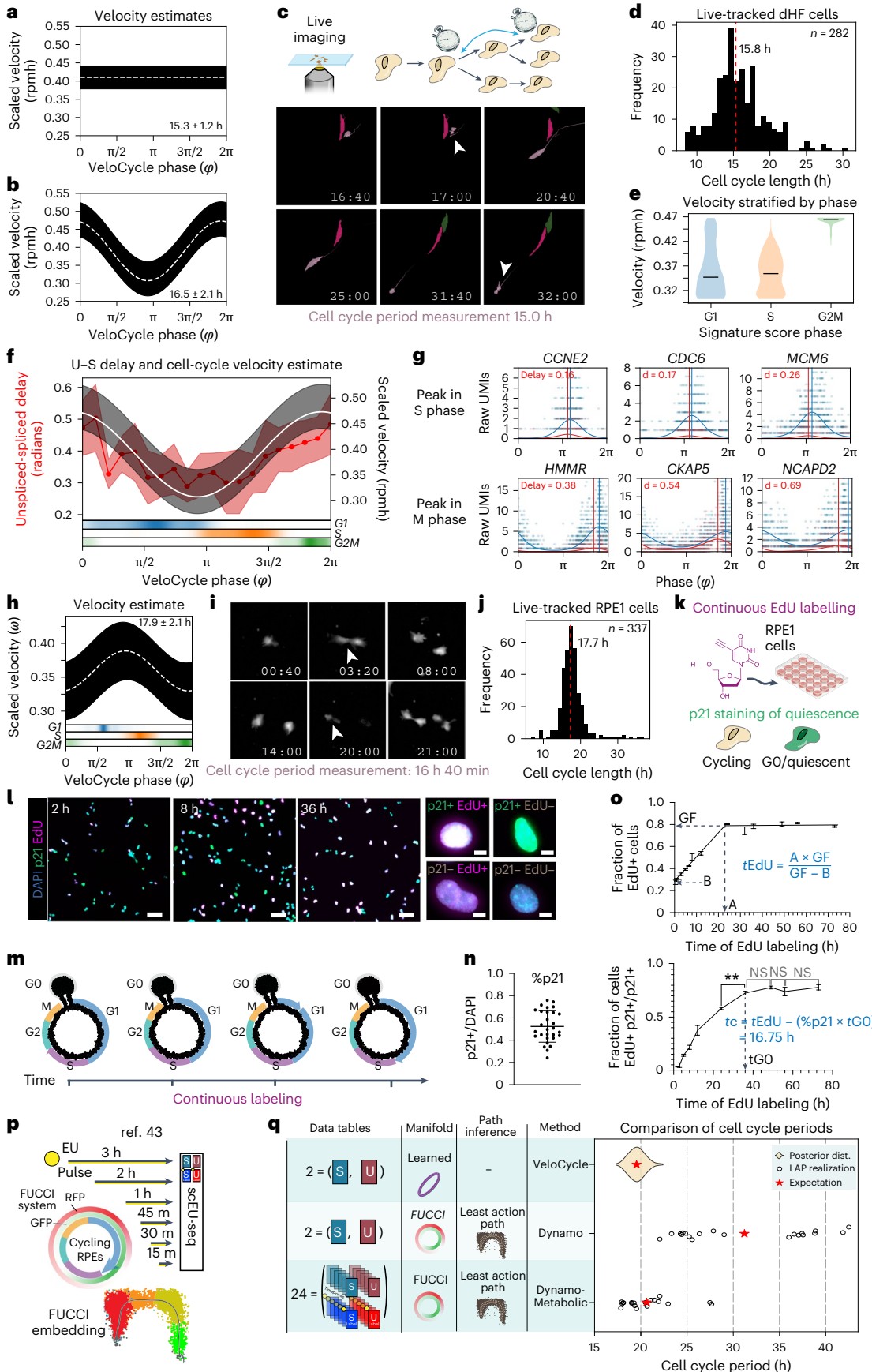

the EdU pulse is directly proportional to the fraction of EdU-positive cells. After monitoring EdU levels at 13 time points over 72 h (Fig. 5m), we used p21 (CDKN1A) staining to account for cells in G0 and determined a mean cell cycle length of 16.8 h (Fig. 5n,o and Methods). Taken together, these findings validated the computational RNA velocity estimates in the context of the cell cycle. To our knowledge, this is the first example of a direct validation of RNA velocity estimation with experimental methodologies and justifies the use of VeloCycle output in units of real (not pseudo) time.

Another approach to further validate cell cycle period estimates is to consider metabolically labeled single-cell 5-ethynyl-uridine (EU) sequencing (scEU-seq), which provides significantly richer information content. Using Dynamo's least action path estimation (LAP) routine[8], specifically designed to obtain a metabolically modeled velocity, we simultaneously processed the 24 data matrices obtained from a recent multi-pulse metabolic-labeling experiment[43] (including spliced and unspliced data, both labeled and unlabeled, across six pulse lengths). Notably, when we compared these LAP results to VeloCycle estimates based only on spliced–unspliced matrices, the posterior probability from VeloCycle closely overlapped with Dynamo estimates, which were obtained using the more-informative labeled dataset and by further leveraging FUCCI staining information as a ground-truth embedding (Fig. 5p,q and Extended Data Fig. 5). These findings further corroborate VeloCycle estimates for cell cycle speed on standard scRNA-seq data using gold standard estimates.

### VeloCycle benchmarks competitively across multiple datasets

Despite the conceptual challenge of comparing VeloCycle to methods of different scope and assumptions, we benchmarked the performance of VeloCycle on four independent datasets (simulated data (Fig. 2), dHFs (Fig. 5a), metabolically labeled A549 cells[44] and RPE1 cells (Fig. 4h)) against four RNA velocity estimation methods (scvelo[4], cellDancer[45], Dynamo[8] and VeloVAE[18]). VeloCycle achieved noticeably improved cross-boundary direction correctness[6] (CBDir; Extended Data Fig. 6a,b) and velocity consistency[21] scores across multiple datasets as compared to pre-existing methods (Extended Data Fig. 6c), even when their embeddings were used to generate the ground-truth clusters for evaluation (Extended Data Fig. 6b). This benchmarking analysis also revealed a higher mean-squared error (MSE) on the spliced and unspliced fits for VeloCycle, which we expected, as it reflects our choice to prioritize regularization over error minimization (Extended Data Fig. 6d–g).

### VeloCycle can test for drug treatment effects on velocity

Existing frameworks for RNA velocity do not propose an approach to test the statistical significance of obtained estimates, likely because it is challenging given a gene-wise velocity parametrization. For example,

it is currently not possible to determine whether RNA velocity estimates close to zero should just be interpreted as noise. Furthermore, direct comparisons between velocity estimates of two samples cannot be supported by a measure of confidence. With VeloCycle, statistical inference on velocity is possible for the first time, both against a specific null hypothesis and for differential velocity significance between cell populations.

To illustrate how our model can be used for statistical velocity tests in practice, we conducted RNA velocity analysis on a PC9 adenocarcinoma cancer cell line before (D0) and after (D3) treatment with the drug erlotinib[46] (Extended Data Fig. 7a–e). Statistical testing in a Bayesian setting can be achieved by calculating credible intervals from the posterior. First, we considered the velocity posterior of the D0 cells to ask whether there is statistical support for a nonzero velocity. Given no overlap between the credible interval and zero, we could conclude the data contains statistically significant evidence for progression through the cell cycle (Fig. 6a, left). We then compared the treated sample (D3) with the control (D0). We found significant velocity differences between the time points, where a slower mitotic cell cycle speed was detected at D3 (Fig. 6b). Such testing can be conducted globally and also locally. For example, we stratified by phase intervals and inspected the posterior samples, confirming a decreased speed during G2/M phase at D3 compared to D0, but not during G1 and S (Fig. 6b and Extended Data Fig. 7f). The reduced presence of cells in mitosis after erlotinib treatment was further suggested by the low density of D3 cells assigned an M phase coordinate (Extended Data Fig. 7a, bottom).

As the unspliced–spliced delay is linked with cell cycle velocity, we hypothesized there would be differential delays between the D0 and D3 time points, particularly for genes peaking during mitosis. After calculating the gene-wise unspliced–spliced delay before and after erlotinib treatment, we indeed noticed a subset of genes with peak expression during mitosis and larger phase delays in D0 than D3 (Fig. 6c); this included anaphase-promoting complex member *CDC27* (differential delay (dd) = 0.11 radians), cyclin-dependent kinase inhibitor *CDKN3* (dd = 0.10) and centrosome scaffolding factor *ODF2* (dd = 0.09) (Extended Data Fig. 7g). A decreased cell cycle speed specifically during M phase is consistent with the expected effect of erlotinib, an EGF-blocker inhibiting progression to G1 (ref. 47). The result also aligns with evidence that a complete arrest should not be observed for the PC9 cell line, which is reported to have some resistance to a full blockade[48–50].

### Cell cycle speed varies spatiotemporally in radial glia

Regulation of proliferation rate as well as symmetric and asymmetric divisions of radial glia (RG) cells in the ventricular zone plays a critical role in controlled developmental timing along an anterior–posterior

**Fig. 6 | Statistical velocity inference across diverse biological contexts and with transfer learning in genome-wide perturbation screens. a,** Posterior estimate plot of scaled velocity in PC9 lung adenocarcinoma cells[46] (D0) compared to a zero-velocity control (red). **b,** Posterior estimate plot of scaled velocity before (D0) and after (D3) erlotinib treatment. Areas where intervals do not overlap indicate statistically significant velocity differences. Bottom: categorical phase assignment probabilities. **c,** Scatterplot of mean unspliced–spliced expression delay for 273 genes between D0 and D3 samples. Gene dots are colored by peak expression phase. **d,** Violin plots of scaled velocity estimates obtained for mouse FB, MB and HB RG progenitors[52] at developmental stage E10. **e,** Spatial projection of single-cell clusters using BoneFight[52] onto four sections of a reference E11 embryo profiled with HybISS, colored by velocity estimates. Regional domains (FB, MB and HB) and the ventricular zone (VZ) are labeled accordingly. **f,** Violin plots of velocity estimates for regional domains at E14–E15. **g,** Bar plot of regional proportions by stage of RG. **h,i,** Kernel density estimation (KDE) plots of cell distributions along the cell cycle manifold at E10 (**h**) and E14–15 (**i**), colored by regional domain. **j,** Posterior estimate plot of cell cycle speed for RPE1 cells 7 days after CRISPR-induced single-gene knockdowns with Perturb-seq[62], stratified by NT control (green) and cell cycle knockdown

(beige) conditions. Manifold learning was performed using either a large (top) or small (bottom) gene set. **k,** Kernel density plot of continuous phase distributions for NT and cell cycle knockdown (CC-KD) samples from **j. l,** Schematic of the employed transfer-learning approach. Gene harmonic coefficients are obtained on NT controls (many cells) and applied to assign phases in specific gene knockdown conditions with few or unequally distributed cells. **m,** Scatterplot of velocity learning posterior estimates and s.d. for 986 individual gene knockdown (Δ) conditions in 167,119 RPE1 cells. Vertical lines correspond to mean velocity estimates for NT (green), cell cycle marker (tan) and other (blue) gene knockdowns. **n,** KDE (top) and binned unspliced–spliced delay (bottom) plots for NT, *MCM6Δ* and *DBR1Δ* conditions. The dark green line represents the mean delay; the light green line represents the s.d. **o,** Scatterplot of scaled cell cycle velocity estimates obtained for conditions in **m** using small and large gene sets. **p,** Scatterplot of total number of cells per condition and posterior velocity s.d. for conditions in **m**. In **a,b** and **j**, black dashed lines represent mean estimates over 500 posterior predictions; bars represent credibility intervals (5th to 95th percentile). In **d** and **f**, black lines indicate means over 500 posterior predictions. In **o** and **p**, Pearson's correlation coefficient is indicated in red.

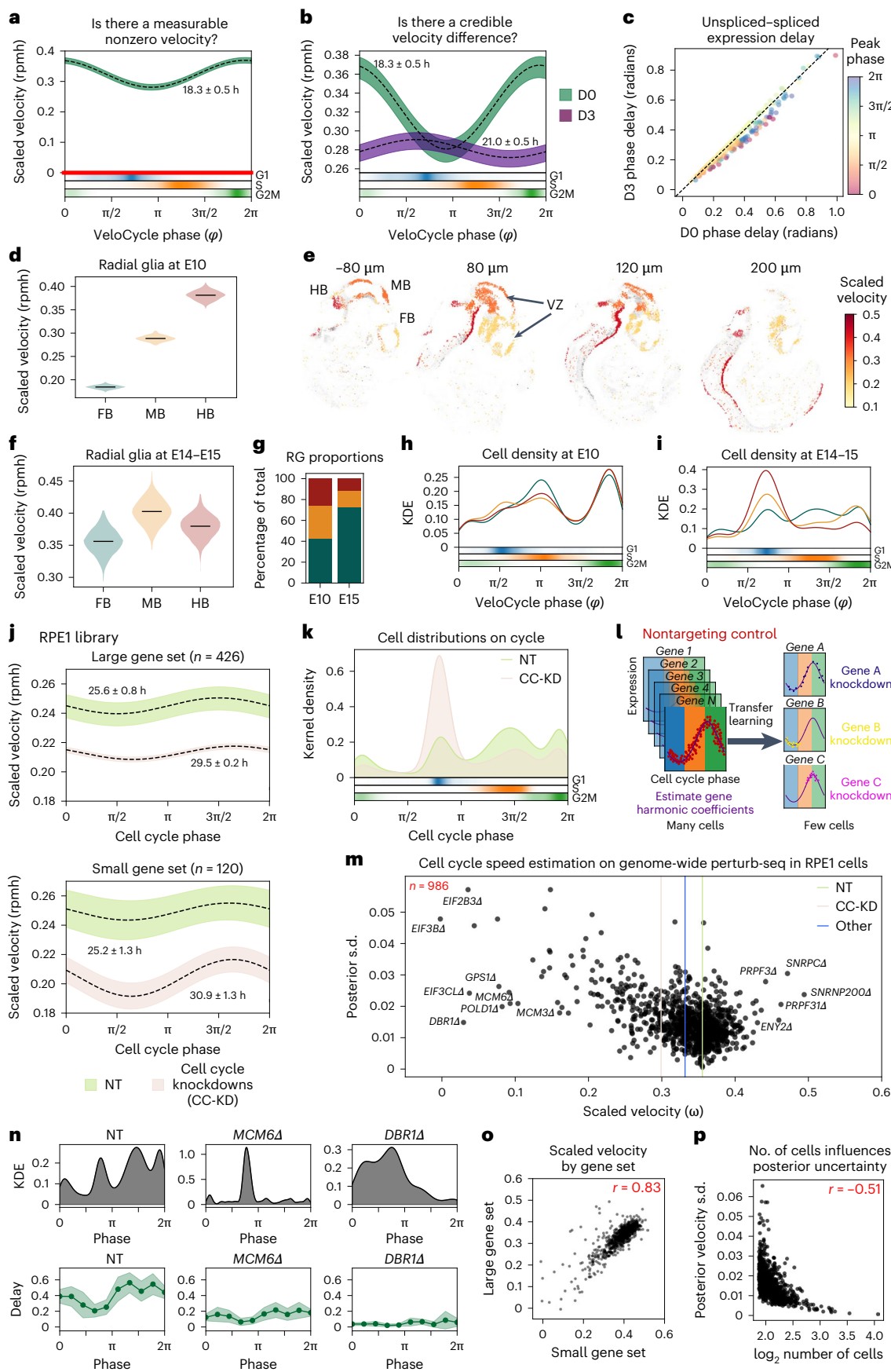

axis of the brain[51]. To elucidate whether there are differences in cell cycle speed among progenitors populating different spatial regions during mouse neurodevelopment, we performed VeloCycle estimation on forebrain (FB), midbrain (MB) and hindbrain (HB) RG cells at the embryonic day 10 (E10) stage[52]. Cell cycle speed varied along the forebrain–midbrain–hindbrain axis, with progenitors dividing more quickly posteriorly (HB) than anteriorly (FB) (Fig. 6d). A finer visualization of this gradient was allowed by computationally mapping the cell cycle speed inferred in these cells to the corresponding locations using in situ hybridization spatial transcriptomics (HybISS) data and the BoneFight algorithm[52] (Methods). We observed rapidly dividing RG cells localized close to the ventricular zones, highlighting that cell proliferation takes place along the ventricular zone and suggesting different segments of the zone proliferate at different rates (Fig. 6e)[53]. Conversely, at E14 and E15, RG cells from all three regions stabilized at a similar proliferation speed, with no credible velocity difference (Fig. 6f). At these later time points, the majority of RG cells in the MB and HB regions had accumulated in a nonproliferative state; the majority of RG cells present were from the FB, which more slowly developed at E10 (Fig. 6g–i). These results align with recent studies showing that HB specifies into nonproliferating, differentiated cell types more quickly; an increased proliferative capacity is thus likely required in the earlier stages of development[54–56]. Furthermore, the later slowdown is expected and in line with what has been reported in EdU tracking studies[57,58].

#### Speed modulation screening is achieved by transfer learning

Previous frameworks for RNA velocity have struggled to obtain reliable estimates using cell types or conditions for which only a limited number of cells are profiled. With recent single-cell technologies designed to screen the effects of hundreds of genetic, environmental or drug perturbations, there is a growing need to assess changes in cell dynamics using a small population of cells[59–61]. VeloCycle can explore RNA velocity contexts that were previously challenging: by conditioning the manifold learning model on gene harmonic coefficients previously inferred from a large reference dataset, one can perform velocity inference using either a smaller number of cells or cells belonging a single cell cycle phase (Methods).

To demonstrate this, we studied a large-scale, genome-wide Perturb-seq dataset where hundreds of individual gene knockdowns were introduced into the RPE1 cell line via a targeted, pooled CRISPR interference library, followed by scRNA-seq after 7 days in culture[62]. First, we ran VeloCycle on nontargeted (NT) control cells and a pooled group of gene knockdown conditions corresponding to well-characterized marker genes for the cell cycle (CC-KD). The cell cycle period was 25.6 ± 1.3 h for NT and 30.9 ± 1.3 h for CC-KD (Fig. 6j), using two differently sized gene sets (Extended Data Fig. 7h–l). When CC-KD conditions were stratified by genes typically considered S and G2/M markers, we observed an accumulation of cells in the G1 phase space compared to NT cells (Fig. 6k and Extended Data Fig. 7m). This suggests that the loss of function for some individual cell cycle-related genes disrupts cell cycle progression, either by slowing down the proliferation rate in certain phases or by halting progression altogether ahead of specific entry checkpoints.

To scrutinize the effect of individual gene knockdown conditions on cell cycle speed, we employed a transfer-learning approach in which we conditioned manifold learning on gene harmonics previously inferred from the NT and CC-KD data subsets, assigning phases to a substantially larger population of 167,119 cells and 986 individual conditions, some with as few as 75 cells (Fig. 6l). Consistent with coarser stratifications of the data, we observed a notable decrease in cell cycle speed in individual cell cycle-related-gene knockdown conditions compared to both NT control cells and cells with gene knockdowns unaffiliated with the cell cycle (Fig. 6m). Several of the most impaired cell cycle speeds were found in knockdowns of highly characterized

genes involved in DNA replication (*MCM3Δ* and *MCM6Δ*) and translation initiation (*E1F3BΔ*, *EIF2B3Δ* and *EIF3CLΔ*). Curiously, knockdown conditions for several splicing and mRNA processing genes either markedly decreased or increased the estimated cell cycle speed, including *DBR1Δ*, an intron-lariat splicing factor (11.7-fold decrease compared to NT condition), *PRPF3Δ* (1.2-fold increase) and *PRPF31Δ* (1.3-fold increase) (Fig. 6m,n). Given the dependence of RNA velocity estimation on the governing differential equations of the RNA metabolic life cycle, this result indicated that biological disruptions affiliated to RNA metabolism undermine the biophysical parameterization of the velocity framework. Moreover, the number of cells present in the dataset per condition had a direct influence on the estimate of velocity uncertainty, suggesting that more cells, and thereby less aggregated sparsity for a condition, increased the confidence of the VeloCycle model in the obtained velocity estimate (Fig. 6o,p). Ultimately, these analyses demonstrate that velocity can be applied, with transfer-learning approaches, in large-scale perturbation contexts as a metric to assess the impact of gene knockdowns on the dynamics of a biological process.

#### Statistical velocity generalizes across manifold geometries

VeloCycle is a probabilistic velocity model designed for 1D periodic manifolds (Fig. 1); however, our new manifold-constrained framework can also be harnessed to generate formulations of higher dimensionality and for various geometries, including a 1D (nonperiodic) interval and two-dimensional (2D) case (Methods and Discussion)[63,64]. To explore this possibility, we formulated, implemented and performed key tests on two new models: a 1D interval (nonperiodic) model designed to study differentiation speed (Extended Data Figs. 8 and 9) and a 2D model suited for examining more complex settings (Extended Data Fig. 10).

For the 1D interval model, we defined the manifold coordinates using an independently estimated pseudotime (diffusion pseudotime) and the manifold geometry with a B-spline basis function, rather than the periodic Fourier series basis used for VeloCycle (Extended Data Fig. 8a). First, we validated model performance conditioned on the pseudotime using ten independently simulated datasets, as with VeloCycle (Fig. 2), accurately recovering the simulated ground-truth velocity and kinetic parameters (Extended Data Fig. 8b–h). We next showcased the ability of this proof of principle extension to infer velocities on a reasonable real-time scale during pancreatic endocrinogenesis (Extended Data Fig. 8i–r and Methods) and the mouse dentate gyrus (Extended Data Fig. 9 and Methods). This demonstrated the potential of this framework to separately infer velocities describing two distinct biological processes co-occurring within a single sample, namely the cell cycle and β-cell differentiation (Extended Data Fig. 8l,o). Finally, we evaluated a more complex 2D case on simulated data (Extended Data Fig. 10a), successfully recovering both ground-truth velocity and kinetic parameters for two processes (differentiation and divergence) simultaneously (Extended Data Fig. 10b–g). Overall, we demonstrate that our manifold-constrained RNA velocity framework can be adapted to formulate other models beyond the periodic case; however, further validation and characterization of these models, as was performed with VeloCycle, will be needed to offer a more robust and standalone tool.

### Discussion

In this work, we address several limitations of current RNA velocity methods by designing a framework that unifies manifold and velocity inference into a single probabilistic generative model. We note that projections and smoothing methods of the velocity field on low-dimensional embeddings have been suggested and applied post hoc to achieve a smoother, less-overfit-prone vector field, which are particularly relevant for visualization purposes and data exploration[5,18–20]. Here, we propose, test and experimentally validate an explicit parametrization of RNA velocity as a vector field defined on the manifold coordinates that, from the beginning, considers tangency among its

core assumptions. The pivotal role of having a velocity estimate tangent to manifold structure has also been recognized by another recently proposed method, graph-dynamo[65], whose tangent space projection corrects the vector field post hoc for visualization and interpretation. VeloCycle accentuates the centrality of velocity tangency further by integrating manifold constraints directly into the estimation process, opening the possibility to exploit them for parameter identification and inference.

Our framework uses variational inference to infer directly from raw data the posterior parameters of our generative RNA velocity model, and it appropriately models the noise in the data instead of using heuristics such as nearest-neighbor smoothing. VeloCycle returns uncertainty estimates, enabling direct evaluation of the confidence about the estimation results and cell cycle speed comparisons between samples. These capabilities are relevant in different biological settings, such as in cancer biology, where alterations to cell cycle progression need to be scrutinized using snapshot single-cell data. RNA velocity has been previously applied to illustrate cell cycle progression, yet in ways that required several heuristics and with exclusive exploratory value, as no conclusion could be made from the inferential procedures[17,29,63,66]. Therefore, VeloCycle could yield new biological insight into disease progression, for example by characterizing differences in proliferation rates between tumors across microenvironments or patients.

Uncertainty measurements are central to statistical evaluation of RNA velocity. The first methods to introduce Bayesian variational inference for RNA velocity modeling, VeloVAE[18], VeloVI[21] and Pyro-Velocity[22], simplify the variational distribution in ways that limit usefulness of the estimated joint posterior, particularly given an unscaled gene-wise velocity parametrization. More generally, models with a high number of degrees of freedom and the assumption of independence risk overfitting noise and overestimating confidence in the velocity[17,23]. We anticipate that our strategy of constraining spliced–unspliced fits under a shared velocity function and controlling for uncertainty dependencies will be further explored by future velocity methods.

Regarding comparisons with other methods, which are overall quite favorable toward VeloCycle, such benchmarks should not be overinterpreted. For example, the optimal velocity consistency and cross-boundary direction correctness (CBDir) scores obtained by VeloCycle are expected given our self-consistent formulation. Analogously, we interpret higher MSE values on gene expression fits as the necessary cost to attain inferential capabilities. Overall, we advocate for selection of methods based on their intended application, proven validity and underlying goals, rather than crude metrics.

VeloCycle enriches velocity analysis by incorporating the manifold's constraints directly into the RNA velocity estimation process. This facilitates a structured regularization by unifying kinetic parameter estimation and the manifold's intrinsic geometry, ensuring coherence of the estimated quantities. Additionally, our approach stands out for offering a unified end-to-end model, which is fit on raw data, avoids heuristics, promotes interpretability and has rigorous inference capabilities. Inference on the primary latent variables of this model (manifold geometry, velocity and kinetic parameters) avoids the pitfalls associated with multiple gene-wise velocities[15]. Specifically, it does not lead to the overconfidence stemming from considering each gene as independent and neglecting the correlation of their uncertainties

While our model for RNA velocity estimation offers clear benefits, there remain open avenues for further development. First, VeloCycle focuses on the case of 1D periodic manifolds, yet extensions to latent spaces of different dimensionalities and topologies can be naturally pursued. We also demonstrate applicability to a 2D case; however, further experimental validation and characterization of these models, as we show for VeloCycle, will be needed.

Second, the issue of defining dimensionality intersects with that of gene selection; different subspaces defined by unique genes expose distinct manifolds traversed by varying fractions of cells[17].

Methods developed with this problem in mind have been recently proposed[67], and with appropriate modifications, these could be integrated into RNA velocity estimation methods to automate topology and gene set selection. In this direction, frameworks that consider multiple manifolds with varying topologies, spanned by cells in different subspaces, while also assigning specific cells and genes to these features, will notably enhance the general applicability and utility of manifold-consistent RNA velocity estimation. These could involve using either unbiased or GO-informed data factorization models to extract and study the speed along these modules[67–69]. This type of gene-modular modeling was successfully demonstrated by the method cell2fate[23].

Third, our model assumes a constant gene-specific splicing and degradation rate; in fact, for some genes, such rates likely change in different phases of the cell cycle[29,43]. A future extension to VeloCycle for which the kinetic parameters are defined by a parameterizable function could address this limitation. Yet, maintaining the model well-conditioned in these settings might be nontrivial.

Fourth, future refinements to VeloCycle could involve the addition of structured priors and other constraints: for the cell cycle, an auxiliary loss could be implemented to favor configurations reflecting total UMI drop after cytokinesis. More generally, the introduction of an entropy regularization could encourage an even distribution of cells across the manifold, and the imposition of low-rank constraints on basis coefficients could improve learning of high-dimensional manifolds. Similarly, priors that postulate a specific sequential activation of genes or leverage knowledge of gene regulatory networks could inject valuable biological information. Introducing genomic features as predictors for kinetic parameters is another promising strategy for regularizing the model. Finally, we have discussed the situation of autonomous dynamical systems $V(x)$; nonautonomous situations, for example owing to experimental designs using external interventions or perturbations[70,71], may in principle be envisaged within this framework, but the formulations and implementations will likely require case-by-case, nongeneric developments.

Widely used standard analysis pipelines rely on a small group of marker genes to attribute a categorical phase assignment to single cells, even though cell cycle progression is a continuous process[26,27]. Recent methods to infer continuous phase assignment represent a valuable improvement over scoring-based approaches[72–75]. The manifold learning of VeloCycle makes progress along this direction, also inferring individual gene periodicity patterns, providing posterior uncertainty and obtaining results that compare favorably with other methods[32]. Notably, the manifold learning step is flexible and facilitates transfer-learning: the manifold geometry can be estimated on a larger or higher quality dataset and serve as a prior for a smaller dataset. This enhances the robustness and applicability of velocity learning across diverse experimental conditions. This is particularly relevant given the increased use of barcoding strategies for single-cell-level screening. We expect future applications of such models in the context of drug screening and evaluation of genetic changes on heterogeneous pools of cells.

A way to validate the overall consistency of an RNA velocity vector field has been to correlate a heuristically estimated transition probability between populations with previous knowledge on their lineage relationships; however, this is correlative and indirect[14]. Here, we instead directly compare estimates with the real velocity of the process. By specifically biologically reasoned priors, velocities obtained with VeloCycle can be interpreted as the proliferation speed, which can vary in different tissue locations, at different moments of development or as a result of perturbations to the core gene regulatory network[53,62]. Although we advocate the use of metabolic-labeling techniques, which with their more-informative experimental design, targeted chemistry and structured data are likely to allow better estimation of velocity kinetic parameters, this practice is limited in application. Thus, the design of experimentally validated or manifold-constrained

RNA velocity methods, such as VeloCycle, is an important effort with general purpose applications by the community.

Ultimately, our framework represents an advancement in the rigor of dynamical estimations from single-cell data. The promising outcomes of tailoring RNA velocity to single processes advocates for the development of new models that dissect the high-dimensionality of single-cell data into individual biological axes with corresponding and interpretable RNA velocity fields.

## Online content

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

## Methods

### Model specifications for manifold-constrained RNA velocity

Gene expression measurements as obtained by scRNA-seq provide a high-dimensional snapshot of a cell's state, with typically $n \simeq 10^4$ genes being expressed in a cell, of which several thousands are experimentally detected per cell by a nonzero read count. Here, we use the notation $Y_c = (U_c, S_c)$ for the measurements, with $U_c$ for the unspliced and $S_c$ for the spliced RNA levels (counts), with $S_c, U_c \in \mathbb{N}^n$.

**The manifold.** Many biological processes of interest, such as the cell cycle or a differentiation event, unfold on low-dimensional manifolds $\mathcal{M}$. A manifold is defined as a surface of points existing in the high-dimensional space with an explicit parametric mapping to a low-dimensional latent space, along which the dynamical process of interest unfolds. Here, we will consider a parametric representation for $\mathcal{M}$, where a latent coordinate $x$ (defined for each cell) maps to the manifold of spliced gene expression levels $\mathcal{M}$ via a deterministic function $s(x)$ (where s indicates 'spliced') describing the expected level of spliced RNA for a cell of coordinate $x$. Moreover, we will choose the manifold topology based on the biological structure of the problem. For example, given a periodic process such as the cell cycle, we will take $x \in S_1$. Typically, the manifold dimension $m \ll n$ will be small, and in the case of the cell cycle $m = 1$. As we discuss later, we will learn the function $s(x)$ from the data (which we will refer to as the manifold learning procedure).

**Measurements and noise model.** Measurements for each cell $c$ will be linked to the corresponding locations on $\mathcal{M}$ via realistic noise models. In the case of scRNA-seq, relevant noise models consist of negative binomial (NB) distributions, so that $Y_{gc} \sim \text{NB}(y_g(x_c), \alpha_g)$, with $y_g(x_c) = E[Y_{gc}] = (s_g(x_c), u_g(x_c))$ and $\alpha_g = (\alpha_g^s, \alpha_g^u)$. Note that we are assuming for simplicity that $\alpha_g$ is independent of $x$ (but this can be relaxed at the expense of an increased number of parameters). This allows us to formulate a likelihood model for the data and approach inference using Bayesian or variational inference.

**RNA velocity and chemical kinetics.** In the high-dimensional gene expression space, we expect a rate equation describing the RNA velocity $\frac{d\tilde{s}}{dt}$ depending on both the expectation of spliced and unspliced RNA counts:

$$\frac{d\tilde{s}_g}{dt} = F(\tilde{s}_g, \tilde{u}_g) = \beta_g \tilde{u}_g - \gamma_g \tilde{s}_g \tag{1}$$

with time-dependent locations $\tilde{s}_g(t)$ and $\tilde{u}_g(t)$ and gene-dependent RNA splicing and degradation rates $\beta_g$ and $\gamma_g$. Note that here we do not include a corresponding equation for $\frac{d\tilde{u}}{dt}$ as it will not be needed for the application to the cell cycle. Also, $F$ is not explicitly time-dependent and the rates are taken as constants (which could however be relaxed, see below).

**Latent-space dynamics.** The key assumption in our approach is that there exists an autonomous (and here deterministic) equation for the dynamics of $x(t)$:

$$\frac{dx}{dt} = V(x) \tag{2}$$

which provides a low-dimensional approximation of the full dynamics (equation 1) and that $\tilde{s}(t), \tilde{u}(t)$ are time-dependent through $x(t)$:

$$\tilde{s}(t) = s(x(t)) \tag{3}$$

$$\tilde{u}(t) = u(x(t)) \tag{4}$$

$V(x)$ is the vector field describing the dynamics in the low-dimensional latent space.

**Manifold-constrained RNA velocity.** We can now link equations. (1)–(4) to obtain

$$\frac{ds_g(x(t))}{dt} = (\nabla_x s_g) \cdot V(x(t)) = \beta_g u_g(x(t)) - \gamma_g s_g(x(t)) \forall g \tag{5}$$

where we have introduced the gene index $g$ for clarity and applied the chain rule. $\beta_g$ and $\gamma_g$ are the gene-specific splicing and degradation rates.

Equation (5) provides the basis of our approach as it connects the topology of the low-dimensional manifold on the left-hand side with the biology on the right-hand side. Of note, the parameters governing gene dynamics ($\beta$ and $\gamma$) could, in principle, also depend on $x$.

**Geometric interpretation.** By construction, we see that the RNA velocity vector $\frac{ds(x(t))}{dt}$ lies in the tangent space of $\mathcal{M}$ at every point of a trajectory $s_g(x(t))$. Indeed $\nabla_x s$ forms an $m$-dimensional basis of the tangent space at each point and $V(x(t))$ forms the components of the velocity vector in that basis.

**$u(x)$ and inference.** Equation (5) can also be viewed as specifying $u(x)$, given $s(x)$, $V(x)$ and the parameters $\beta$ and $\gamma$. This will become central in the implementation. In essence, the optimization algorithm to identify $V(x)$ and $\gamma$ and $\beta$ coefficients (or functions if we would allow $\gamma = \gamma(x)$, etc.) such that the predicted RNA velocity $\frac{ds(x(t))}{dt}$ (which lies in tangent space over the entire manifold $\mathcal{M}$) is closest to that implied by chemical kinetics and the data $Y_c = (U_c, S_c)$.

**Duration of biological processes.** A benefit of this formulation is that it becomes accessible to estimate the actual duration of biological processes from the trajectories and $V(x)$:

$$\Delta t_{s_0, s_1} = \int_{\Gamma_{s_0}^{s_1}} \frac{1}{\dot{s}} ds = \int_{\Gamma_{x_0}^{x_1}} \frac{1}{V(x)} dx = \Delta t_{x_0, x_1} \tag{6}$$

where $\Gamma_{x_0}^{x_1}$ is the trajectory $x(t)$ that connects the two points $x_0$ and $x_1$, and where we have used the change of trajectory variable $s(x)$. For example, we will be able to estimate cell cycle periods. Moreover, this estimate is by construction independent of the parametrization of the low-dimensional manifold.

### Manifolds with $S^1$ topology: the cell cycle

Here, we assume that $\mathcal{M}$ is topologically a circle and therefore we write the coordinate $x$ as $\varphi \in S^1$. The equation of the dynamics (equation 5) becomes

$$\frac{d}{dt} s_g(\varphi(t)) = \frac{d}{d\varphi} s_g(\varphi)\omega(\varphi) = \beta_g u_g - \gamma_g s_g \tag{7}$$

$$E[S_{gc}] = s_g(\varphi_c) = \exp\left(\sum_f v_{gf}\zeta_f(\varphi_c)\right) \tag{8}$$

where we assume that $\beta_g$ and $\gamma_g$ are constant along the cell cycle. Of note is that the values of those parameters are constrained by the biology (see below), which we will enforce through appropriate priors. $S^1$ is convenient as it allows use of Fourier series to parameterize the various functions: $s(\varphi), u(\varphi), \omega(\varphi)$. Typical cell cycle genes exhibit profiles that can be described by only few harmonics; thus, we will consider up to $k$ Fourier components in our expansion (in practice we will by default use one harmonic). Moreover, as $s(\varphi)$ is positive, we will use the notation

$$\log(s_g(\varphi_c)) = \sum_f v_{gf} \zeta_f(\varphi_c) \tag{9}$$

with

$$v_g = \begin{pmatrix} a_g^0 \\ a_g^1 \\ b_g^1 \\ \vdots \\ a_g^k \\ b_g^k \end{pmatrix} \tag{10}$$

$$\zeta(\varphi) = \begin{pmatrix} 1 \\ \cos(\varphi) \\ \sin(\varphi) \\ \vdots \\ \cos(k\varphi) \\ \sin(k\varphi) \end{pmatrix} \tag{11}$$

Here $v_g$ is the vector of gene Fourier parameters written with real numbers.

Using the chain rule, we obtain $u(\varphi)$:

$$\frac{d}{dt} s_g(\varphi(t)) = \omega(\varphi) s_g(\varphi) \sum_f v_{gf} \frac{d}{d\varphi} \zeta_f(\varphi) \tag{12}$$

which leads to

$$\log(u_g(\varphi)) = -\log(\beta_g) + \log\left(\omega(\varphi) \sum_f v_{gf} \partial_\varphi \zeta_f(\varphi) + \gamma_g\right) + \log(s_g(\varphi)) \quad \forall g \tag{13}$$

$$E[U_{gc}] = u_g(\varphi) = \frac{s_g(\varphi)}{\beta_g}\left(\omega(\varphi) \sum_f v_{gf} \partial_\varphi \zeta_f(\varphi) + \gamma_g\right) \tag{14}$$

For $\omega(\varphi)$ we will also be using a Fourier series, limiting ourselves to either constant $\omega$ or $\omega(\varphi)$ functions with one harmonic.

**Likelihoods.** As explained above with the expressions for $u(\varphi)$ and $s(\varphi)$, we can calculate a likelihood for the count data over all cells $\{Y_c\} = \{(U_c, S_c)\}$. To simplify the implementation, we approximate the full joint likelihood for $\{(U_c, S_c)\}$ as a product of two factors:

$$P(\{(S_c, U_c)\}|\theta) = \prod_{gc} P(S_{gc}, U_{gc}|\omega(\varphi), \varphi_c, v_g, \beta_g, \gamma_g, \alpha_g) \text{ with}$$
$$P(S_{gc}, U_{gc}|\theta) = P_s\left(S_{gc}|v_g, \alpha_g^s, \varphi_c\right) \times P_u\left(U_{gc}|\omega(\varphi), \beta_g, \gamma_g, v_g, \varphi_c, \alpha_g^u\right) \tag{15}$$

$$P_s(S_{cg}|\ldots) = \text{NB}\left(s_g(\varphi_c) = F[v_g, \varphi_c], \alpha_g^s\right), \tag{16}$$

$$P_u(U_{cg}|\ldots) = \text{NB}\left(u_g(\varphi_c) = G[\omega(\varphi_c), \beta_g, \gamma_g, v_g, \varphi_c], \alpha_g^u\right) \tag{17}$$

where $\theta$ is a generic notation for parameters, and $F[\ldots], G[\ldots]$ show the dependencies of $s_g, u_g$ on the other quantities.

We combine these likelihoods with a set of priors into a full Bayesian model (see below) to estimate the joint posterior of $\theta$. As indicated above, in our current implementation we simplify the problem by taking two steps: first, we optimize $P_s$ to estimate the cell phases $\{\varphi_c\}$ and Fourier coefficients $\{v_g\}$. We call this step the manifold learning procedure. The second step optimizing $P_u$ is called velocity learning

and uses the posterior expectations for $(\{\varphi_c\}, \{v_g\}, (\alpha_g^s))$ obtained during manifold learning to estimate the remaining quantities $(\omega(\varphi), \beta_g, \gamma_g, \alpha_g^u)$.

**Bayesian model formulation for VeloCycle**

Our model includes a mix of biologically defined priors with empirical Bayes-style priors determined from the data. Our goal will be to estimate an approximation of the joint posterior probability distribution, based on the above expression of the likelihoods:

$$P(\theta|\{S_c, U_c\}) = \frac{P(\{S_c, U_c\}|\theta) P(\theta)}{P(\{S_c, U_c\})} = \frac{\prod_{gc} P(S_{gc}|\theta) P(U_{gc}|\theta) P(\theta)}{\int \prod_{gc} P(S_{gc}|\theta) P(U_{cg}|\theta) P(\theta) d\theta} \tag{18}$$

We specify the following priors $P(\theta)$.

$$v\omega_t \sim \mathcal{N}\left([0, 0, 0], [3^2, 0.05^2, 0.05^2]\right)$$
$$\log(\gamma_g) \sim \mathcal{N}\left(0, 0.5^2\right)$$
$$\log(\beta_g) \sim \mathcal{N}\left(2, 3^2\right)$$
$$\alpha_g \sim \text{Gamma}\left(1.0, 2.0\right)$$
$$v_{gt} \sim \mathcal{N}\left(\mu_{gt}^v, \sigma_{gt}^{v\,2}\right)$$
$$\varphi xy_c \sim \text{ProjNormal}\left(\varphi x_c, \varphi y_c\right)$$

Setting by empirical Bayes the following parameters:

$$\mu_{gt}^v = [\log(\text{mean}_c(S_{gc})), \mathbf{0}, \mathbf{0}]$$

$$\sigma_{gt}^v = \begin{bmatrix} \frac{1}{2}\text{std}_c(S_{gc} + 1), \\ \frac{1}{4}\text{std}_c(S_{gc} + 1), \\ \frac{1}{4}\text{std}_c(S_{gc} + 1) \end{bmatrix}$$

$$\varphi x_c = \varepsilon \cos(\Phi_c)$$
$$\varphi y_c = \varepsilon \sin(\Phi_c)$$

where $\Phi_c$ is obtained from the two first principal components $(w_{1c}, w_{2c})$ renormalized between $[-0.5, 0.5]$ and computing $\Phi_c = \tan^{-1}(w_{2c}, w_{1c})$.

Rotational invariance (for example, arbitrariness of the first cell c0 so that $\Phi_{c0} = 0$) is obtained by finding the global phase shift maximizing $\text{corr}(\Phi_c, \sum_g S_{gc})$. The concentration parameter of the projected normal $\varepsilon$ is set to 5 by default, but can be adjusted depending on the overall confidence in the data quality.

**Variational distribution: SVI.** The variational distribution we use in the base model is mean-field, with marginals of either normal or Dirac delta distributed. Specifically

$$P(\{v\omega_t\}, \{\varphi_c\}, \{v_{gt}\}, \{\beta_g\}, \{\gamma_g\}, \{\alpha_g\})$$
$$= \prod_c \prod_g \prod_t P(v\omega_t) P(\varphi_c) P(v_{gt}) P(\beta_g) P(\gamma_g) P(\alpha_g) \tag{19}$$

The variational distribution is parametrized as follows (^ indicates the parameters):

$$P(v\omega_t) = \mathcal{N}\left(\widehat{\mu v\omega_t}, \widehat{\sigma v\omega_t^2}\right)$$
$$P(v_{gt}) = \mathcal{N}\left(\widehat{\mu_{gt}^v}, \widehat{\sigma_{gt}^{v\,2}}\right)$$
$$P(\alpha_g) = \text{Delta}\left(\widehat{\alpha_g}\right)$$
$$P(\log(\gamma_g)) = \mathcal{N}\left(\widehat{\mu \log \gamma_g}, \widehat{\sigma \log \gamma_g^2}\right)$$
$$P(\log(\beta_g)) = \mathcal{N}\left(\widehat{\mu \log \beta_g}, \widehat{\sigma \log \beta_g^2}\right)$$
$$P(\varphi xy_c) = \mathcal{N}\left([\widehat{\varphi x_c}, \widehat{\varphi y_c}], [1, 1]\right)$$

## Overview of VeloCycle latent variables

| Variable | Description | General name | Training step | Dimensions |
|---|---|---|---|---|
| $\varphi_{xy}$ | cell cycle phase | Manifold coordinates | manifold learning | (cell) |
| $v$ | Fourier coefficients for the genes | Manifold geometry | manifold learning | (gene, harmonics) |
| $\Delta v$ | batch-specific expression offset | Data-specific noise in manifold geometry | manifold learning | (batch, gene) |
| $shape\_inv$ | spliced NB noise | Measurement noise | manifold learning | (gene) |
| $\log \beta_g$ | log splicing rate | Velocity kinetics | velocity learning | (gene) |
| $\log \gamma_g$ | log degradation rate | Velocity kinetics | velocity learning | (gene) |
| $v\omega$ | Fourier coefficients for the angular speed | Velocity function | velocity learning | (condition, harmonics) |

**Variational distribution: LRMN.** The low rank multivariate normal (LRMN) model considers a variational distribution parametrized to mimic the correlative structure observed between the joint posteriors sampled by MCMC estimation. Specifically, we allow for a covariance and establish specific conditional relationships between the velocity, or angular speed $v\omega_t$, and the kinetic parameters $\beta_g$ and $\gamma_g$. The two main features are: (1) the joint posterior between $\gamma_g$ and $v\omega_t$ is parametrized as a LRMN; and (2) the marginal posterior of $\beta_g$ is expressed as conditioned on $\gamma_g$; namely, for each gene g, the marginal posterior of $\beta_g$, through an explicit parameter $\widehat{\rho_g}$, is allowed to correlate with the correspondent $\gamma_g$. The posterior factorizes as follows:

$$P(\{v\omega_t\},\{\varphi_c\},\{v_{gt}\},\{\beta_g\},\{\gamma_g\},\{\alpha_g\})$$
$$= P(\{\gamma_g\},\{v\omega_t\}) \prod_g P(\beta_g | \gamma_g) P(\alpha_g) \prod_t P(v\omega_t) P(v_{gt}) \prod_c P(\varphi_c)$$

The specific formulation that we used is:

$$\mathbf{x} \equiv \left[ \log(\gamma_1), \log(\gamma_2), \dots, \log(\gamma_{n_g}), v\omega_0, v\omega_1, \dots, v\omega_{n_{t-1}} \right]$$

$$\Sigma = \hat{\mathbf{F}}\hat{\mathbf{F}}^{\mathsf{T}} + \mathrm{diag}(\hat{\mathbf{d}}) \text{ where } \hat{\mathbf{F}} \in \mathbb{R}^{(n_g+n_t) \times k}, \text{ with } k = 5$$

$$P(\{\log(\gamma_g)\},\{v\omega_t\}) = P(\mathbf{x}) = \mathrm{MultivariateNormal}(\hat{m},\Sigma)$$

$$\mu \log \beta_g | \gamma = \widehat{\mu \log \beta_g} + \widehat{\rho_g} \cdot \widehat{\mu \log \beta_g} \cdot \frac{\left(\log(\gamma_g) - \widehat{\mu \log \gamma_g}\right)}{\widehat{\sigma \log \gamma_g}} \text{ with } \widehat{\rho_g} \in [0,1]$$

$$\sigma \log \beta_g | \gamma = \widehat{\mu \log \beta_g} \sqrt{1 - \widehat{\rho_g}^2}$$

$$P(\log(\beta_g) | \log(\gamma_g)) = \mathcal{N}(\mu \log \beta_g | \gamma, \sigma \log \beta_g | \gamma^2)$$

$$P(\varphi xy_c) = \mathcal{N}([\widehat{\varphi x_c}, \widehat{\varphi y_c}], [1,1])$$

$$P(v_{gt}) = \mathcal{N}\left(\widehat{\mu^v_{gt}}, \widehat{\sigma^v_{gt}}^2\right)$$

$$P(\alpha_g) = \mathrm{Delta}(\widehat{\alpha_g})$$

## Model implementation

To estimate an approximation of the joint posterior probability distribution for the angular cell cycle speed ($v\omega_t$) and the parameters of the $S^1$ manifold upon which $v\omega_t$ unwinds, we formulate a likelihood model for the data that we then solve using variational inference in Pyro. This implementation performs an estimation of the model latent variables in two steps: manifold learning and velocity learning.

For manifold learning, we estimate the position of each cell along the circular cell cycle manifold ($\varphi$) as well as the Fourier series coefficients for each gene ($v$) used to model the expectation of log spliced counts (ElogS), which are themselves modeled from the real data and an NB. We initialize all variables to the mean of the prior, which is determined using either the first two principal components ($\varphi$) or the per-gene mean and s.d. of the spliced expression ($v$). To allow for differences in average expression levels between different datasets or batches, we also define an offset term ($\Delta v$) for the first gene harmonic coefficient.

For velocity learning, we infer the Fourier coefficients of the angular speed ($v\omega$) as well as velocity kinetic parameters ($\gamma$ and $\beta$), conditioned on the mean of the posterior estimates for parameters obtained during manifold learning. These variables are used to model the expectation of log unspliced counts (ElogU), which are themselves modeled from the real data and an NB. We initialize all variables to the mean of the prior, which is zero for the angular speed (an assumption of zero cell cycle velocity). To enforce positive $\left(\omega(\varphi)\sum_f v_{gf}\partial_\varphi \zeta_f(\varphi) + \gamma_g\right)$ in equation (10) during learning, we use a relu function.

Given data, we solve the VeloCycle model using SVI and apply a ClippedAdam optimizer and ELBO loss function, with an evolving learning rate decaying from 0.03 to 0.005 from the first to last training iteration. Typically, we perform 5,000 training iterations for manifold learning and 10,000 training iterations for velocity learning; however, an option to terminate training early is made available, such that no further iterations are executed if the mean loss during the previous 100 iterations is fewer than five units different from the mean loss during the previous 10 iterations.

When performing MCMC, we use a No-U-Turn (NUTS) kernel beginning the mean posterior estimates obtained first with SVI. We typically use one chain, 2,000 warm-up sampling steps and 500 real sampling steps.

VeloCycle can be run using either a local CPU or GPU in a few minutes, with significantly improved runtime speeds on GPU, particularly when using a large number of cells (>30,000 cells) or genes (>300 genes). As there are many more parameters along the gene dimension, scaling up the number of genes reduces runtime more quickly than scaling up the number of cells.

**Biological constraints on parameters.** The velocity kinetic parameters $\beta$ and $\gamma$ are constrained by the biology. In particular,

$$\gamma_g^{-1} \in [0.5, 1.5]\mathrm{h}$$
$$T = 2\pi/\omega_o \in [6, 50]\mathrm{h}$$

Moreover, the priors for the gene harmonic coefficients are determined for each gene based on the mean level of expression and the variance across all the cells in the data. For the velocity harmonic coefficients, we assume as a prior mean no velocity (that is, 0) with a wide s.d. (3.0).

All priors can be easily modified using the 'velocycle.preprocessing' suite of functions and provided to a Pyro model object using the metaparameters ('mp') term.

**Approximate point estimate for constant cell cycle velocity.** To gain an initial insight into the relationship between cell cycle velocity and the expression profiles of the unspliced ($u$) and spliced ($s$) read counts, we used a simplified calculation based on solving the first-order differential equation $\frac{d}{dt} s_g(t) = \beta_g u_g - \gamma_g s_g$, where the degradation rate $\gamma_g$ is a gene-dependent constant. If we assume that $u_g(t)$ follows a periodic function with a single harmonic, which is $u_g(t) = u_{0g}(1 + \varepsilon \cos(\omega t - \varphi_{0g}))$ then $s_g(t)$ has the same functional form but with a scaled amplitude and shifted phase, depending on the half-life: $s_g(t) = s_{0g}(1 + \varepsilon' \cos(\omega t - \varphi_{1g}))$, with $\varepsilon' = \varepsilon \cos(\Delta\varphi_g)$, $\Delta\varphi_g = (\varphi_g - \varphi_{0g})$ and $\tan(\Delta\varphi_g) = \omega \gamma_g^{-1}$. Here, $\omega$ represents the cell cycle velocity.

Assuming now that we have multiple conditions (or replicates) $c$ and that the life times $\tau_g = \gamma_g^{-1}$ are condition-independent, we observe that the relation

$$\delta_{cg} = \tan(\Delta\varphi_{cg}) = \omega_c \tau_g$$

is a rank-1 decomposition of the matrix $\delta_{cg}$, which can be computed using the singular value decomposition, which is $\delta_{cg} = u_c d v_g +$ higher rank terms, using standard notation. This allows us to express the condition-specific cell cycle velocity $\omega_c$ in units of inverse mean half-lives (noted $\omega_c^\star$) as

$$\omega_c^\star = u_c d \bar{v}_g$$

where $\bar{v}_g$ stands for the mean over genes. The cycle-cycle period in units of mean half-lives is then $T_c^\star = \frac{2\pi}{\omega_c^\star}$.

**Gene sets and quality control filtering.** To select genes for velocity analysis that are expected to behave periodically with the cell cycle, we applied one of three differently sized, literature-based cycling gene sets: 'small' containing 97 genes[27], 'medium' containing 218 genes[32] and 'large' containing 1,918 genes[34]. VeloCycle uses the 'medium' gene set as a default, to minimize the influence of noisy or lowly expressed genes on manifold and velocity estimation; however, we also employed the 'large' gene set in contexts where the sequencing depth and dataset quality are particularly high. The function 'velocycle.utils.get_cycling_gene_set' can be used to access these human and mouse gene sets. Additional gene filtering based on mean detection of spliced and unspliced counts was also performed as described in the sections below.

**Categorical and continuous cell cycle phase assignment.** Categorical cell cycle phase assignment (G1, S and G2/M) was performed using the scanpy function 'sc.tl.score_genes_cell_cycle', as previously described[26,27]. Continuous cell cycle phase assignment using DeepCycle on both simulated and real datasets was achieved using the velocity information obtained from 'scvelo.pp.moments'[4] and standard parameters described in the original publication[33].

**Inference of the unspliced–spliced delay.** To compute the unspliced–spliced delay from the results of VeloCycle, we calculated the difference between phases of peak expression of unspliced and spliced UMIs on a per-gene basis (in radians) using the estimated expectations of unspliced (ElogU) and spliced (ElogS) counts.

**Posterior probability sampling.** Unless otherwise stated, the latent variables and associated estimate uncertainties were collected from 500 posterior samples after model training using 'pyro.infer.predictive' and credibility intervals were measured between the 5th and 95th percentiles.

Estimates for the cell cycle velocity obtained from the velocity function $\omega(\varphi)$ were scaled by the mean degradation half-life, that is, mean($\gamma_g$). To infer the cell cycle period over the entire cell cycle, we sampled from the velocity function on a grid of 20 phases (from 0 to $2\pi$) and took the area under the curve using 'scipy.integrate.trapz'.

The posterior mean, 5th percentile and 95th percentile were then computed using 'numpy.mean' and 'numpy.percentile'. The full uncertainty range of the posterior estimate was computed by taking the difference between the 95th and 5th percentile estimates.

**Extension to 1D nonperiodic intervals and 2D manifolds.** We explore the extensions of our model to other 1D or 2D manifolds. The first major change that needs to be made to accommodate for nonperiodicity is replacement of the Fourier series with some other smooth function. To ensure the applicability of our framework across different biologically relevant scenarios, we limit the assumptions on the form of that function to the continuity of its first and second derivatives. In particular, we use cubic B-spline basis ($B_3$) instead of the Fourier components:

$$\log(s_g(\varphi_c)) = \sum_f w_{gf} B_{3,f}(\varphi_c)$$

$$\log(u_g(\varphi)) = -\log(\beta_g)$$
$$+ \log\left(\sum_{i=1}^{2} \omega_{\varphi_i}(\varphi_c) \partial_{\varphi_i} \log(s_g(\varphi_c)) + \gamma_g\right) + \log(s_g(\varphi)) \,\forall g$$

Here, $\varphi_i, i \in \{1,2\}$ denote two dimensions of the manifold $\varphi$. We note that in the 1D case the B-spline basis of order $p$ ($p = 3$ in our case) and basis dimension $m$ are defined on a variable $t$ with a sequence of knots $t_i, i \in \{1, \ldots, m + p + 1\}$ by the following equations:

$$B_{p,i}(t) := \frac{t - t_i}{t_{i+p} - t_i} B_{p-1,i}(t) + \frac{t_{i+p+1} - t}{t_{i+p+1} - t_{i+1}} B_{p-1,i+1}(t),$$

$$B_{0,i}(t) := \begin{cases} 1 \text{ if } t_i \leq t \leq t_{i+1}, \\ 0 \quad \text{otherwise} \end{cases}$$

In the 2D case, both the spliced counts and the velocity are modeled using 2D splines, which we obtain from the Cartesian product of regular 1D B-splines. Though this vector field might not be fully representative of branching processes derived from potential landscapes, and which contain critical points, it serves the purpose of illustrating how a 2D can be tackled in principle.

Both 1D and 2D spline-based models are implemented in Pyro. The estimation is performed in two steps. In the first one, we fit the coefficients of the splines for the spliced counts, conditioned on the chosen $\varphi$. In the second one, we estimate the kinetic parameters and the velocity based on the unspliced counts. For the spline coefficients, we choose a broad normal prior with zero mean. In the 1D model, the B-spline basis for the spliced counts has five dimensions and the velocity $\omega(\varphi)$ is modeled as a scalar value, constant across the differentiation process. In the 2D model, the splines for both the spliced counts and velocity have six basis dimensions per axis. We use mean-field variational distribution and initialize all variables to the mean of the prior. The model is trained using SVI with either Adam or ClippedAdam optimizer.

$$\log(u_g(\varphi)) = -\log(\beta_g)$$
$$+ \log\left(\omega(\varphi) \sum_f w_{gf} \partial_\varphi B_{f,3}(\varphi) + \gamma_g\right) + \log(s_g(\varphi)) \,\forall g$$

**Structured data simulations and sensitivity analyses of VeloCycle**

To properly validate the performance of VeloCycle on datasets with a ground-truth for all latent variables of the manifold learning and velocity learning procedures, we employed a new structured simulation approach to preserve relationships among velocity kinetic parameters (splicing rate $\beta$ and degradation rate $\gamma$) and gene harmonics ($v_0$, $v_{1sin}$ and $v_{1cos}$). These relationships are expected in real data[2] and are necessary in

simulations to avoid improbable scenarios where the ratio of unspliced to spliced counts is unrealistically high or low. We expect that genes containing more velocity information should be those with a larger unspliced–spliced delay and slower splicing and degradation rates; genes with too fast kinetics will provide limited signal in scRNA-seq data. Thus, we formulated a generative VeloCycle model that imposes a correlation structure among the gene harmonic and velocity kinetic rate parameters for the sole purpose of sampling simulated data (and not for use during inference itself). We defined correlations as follows: a weak positive correlation among the gene harmonic coefficients ($r = 0.05$; assuming only one sine and cosine term per gene), a moderate positive correlation between the splicing rate and zeroth gene harmonic coefficient $v_0$ ($r = 0.30$), and a moderate positive correlation between splicing and degradation rates ($r = 0.30$).

Using this correlation matrix, simulated datasets were generated by randomly sampling for a user-defined number of genes and cells, from a 'pyro.dist.MultivariateNormal'. These variables, along with a user-defined ground-truth cell cycle speed ($\nu\omega$) and a cell-specific phase ($\varphi$) sampled from a random uniform distribution between 0 and $2\pi$, were plugged into the velocity equations to compute an expectation for unspliced (ElogU) and spliced (ElogS). Finally, raw data (S and U) was sampled from a 'pyro.dist.GammaPoisson' using the expectations and a noise parameter (shape_inv) sampled from 'pyro. dist.Gamma'. All simulated data generated for this study are available on Zenodo (see Data Availability). Additional datasets can be simulated using the 'velocycle.utils.simulate_data' function.

Evaluation of the manifold learning step was performed using 20 datasets, each containing 3,000 cells and 300 genes, independently simulated with a ground-truth velocity of 0.4. The same datasets were also used for validation of the velocity learning step. To perform sensitivity analysis on the number of cells and genes, four independently simulated datasets containing 10,000 cells and 1,000 genes were generated; data subsets were used to test the model's performance on varying numbers of cells (from 100 to 5,000 for manifold learning and from 50 to 10,000 for velocity learning) and genes (from 100 to 1,000 for manifold learning and from 50 to 1,000 for velocity learning). To assess velocity learning performance on datasets with different ground-truth velocities, we simulated four datasets with shared kinetics and gene harmonic parameters, but one of 16 different ground-truth velocities from 0.0 to 1.5.

Circular correlations between estimated and simulated ground-truth variables were computed using 'velocycle.utils. circular_corrcoef', which converts the input data into unit circle coordinates and computes a correlation by finding the mean of the product of estimated values and the complex conjugate of the ground-truth values. To compare phases obtained with VeloCycle to those from DeepCycle, the same simulated datasets were used to compute velocity moments with sc.pp.moments followed by running DeepCycle with default parameters described in the original publication[32].

### VeloCycle estimation across multiple standard scRNA-seq datasets

In this work, we performed manifold geometry and cell cycle velocity estimation with VeloCycle on a number of published datasets from different technologies, species and sampling contexts. For all datasets, the original raw data were reprocessed using VeloCycle[2] to obtain spliced and unspliced count matrices. A general procedure for running VeloCycle on these types of scRNA-seq data has been described above and is supported by tutorials on the corresponding GitHub page for these works. Here, we provide a summary of any specific filtering criteria and parameters used on a dataset-dependent basis.

**FACS-sorted mouse embryonic stem cells.** VeloCycle estimation of cell cycle phases and gene harmonics was performed on 279 single cells from a culture of Smart-seq2 mES cells using the standard parameters[33].

Genes used in manifold learning were those from the 'large' gene set (GO; 1,918 genes) available in 'velocycle.utils', after filtering out genes with ≤0.5 mean unspliced counts per cell or with ≤2 mean spliced counts per cell (1,358 genes remaining). Manifold learning was performed using 3,000 training steps.

To evaluate the predictive capacity of categorical cell cycle phase (G1, S or G2/M) using the VeloCycle phases, a 'DecisionTreeClassifier' from 'sklearn.tree' was trained with 65% of cells, reserving 35% of cells as a test set and for calculation of a confusion matrix. To compare with a model using the total gene expression matrix to predict categorical cell cycle phases, the linear 'LogisticRegressionCV' model from 'sklearn. linear_model' was trained (c.v. = 5) using the same train–test cell split as with the decision tree.

**Mouse embryonic stem cells and human fibroblasts.** VeloCycle was run separately on 5,191 single cells from a culture of mES cells and on 2,557 single cells from a culture of human fibroblasts using standard parameters[32]. Noncycling cells were filtered out before analysis according to the author's annotations. Genes used in manifold learning were those from the 'medium' gene set (DeepCycle; 218 genes) available in velocycle.utils, after filtering out genes with ≤0.1 mean unspliced UMIs per cell or with ≤0.3 mean spliced UMIs per cell (189 genes and 160 genes remaining for mES cell and fibroblasts, respectively). Manifold learning was performed using 5,000 training steps and velocity learning was performed using the 'normal' guide and the constant-velocity model for 10,000 training steps. Comparisons to DeepCycle phases were made using the published estimates described for these exact datasets in the original study.

**Human dermal fibroblasts.** VeloCycle was run on 1,222 single cells from a culture of untreated dHFs using the standard parameters; noncycling cells were excluded using the author's annotations[40]. Genes used in manifold learning were those from the 'large' gene set (GO; 1,918 genes) available in 'velocycle.utils', after filtering out genes with ≤0.1 mean unspliced UMIs per cell or with ≤0.3 mean spliced UMIs per cell (876 genes remaining). Manifold learning was performed using 5,000 training steps and velocity learning was performed with both the constant-velocity and periodic-velocity models for 10,000 training steps using the LRMN ('lrmn') guide.

Time-lapse microscopy data, including cell segmentation and tracking, for dHFs were obtained from the originally published study and are available on Zenodo at https://doi.org/10.5281/zenodo.6245943 (ref. 76). A cell was determined to be poorly tracked and excluded from analysis if it had a measured cell cycle length less than 8 h or greater than 32 h.

**PC9 lung adenocarcinoma cell line.** VeloCycle was run jointly on data from PC9 lung adenocarcinoma cell line before (D0, 7,927 cells) and after (D3, 3,743 cells) treatment with erlotinib using the standard parameters[46]. Genes used in manifold learning were those from the 'large' gene set (GO; 1,918 genes) available in 'velocycle.utils', after filtering out genes with ≤0.1 mean unspliced UMIs per cell or with ≤0.1 mean spliced UMIs per cell. After an initial manifold-learning step, only genes with a Pearson's correlation between the unspliced and spliced counts ≥0.8 and a predicted unspliced–spliced delay greater than ≥−0.25 were retained. Manifold learning was performed using 5,000 training steps and velocity learning was performed using the 'lrmn' guide and both the constant-velocity and periodic-velocity models for 10,000 training steps.

**Radial glial progenitors from the developing mouse brain.** VeloCycle was run jointly on all RG progenitor cells from the E10 time point, stratified by regional identity (FB, 3,293 cells; MB, 2,388 cells; HB, 2,012 cells) using the standard parameters[52]. Genes used in manifold learning were those from the 'large' gene set (GO; 1,918 genes) available in

'velocycle.utils', after filtering out genes with ≤0.05 mean unspliced UMIs per cell or with ≤0.1 mean spliced UMIs per cell. After an initial manifold learning step, only genes with a Pearson's correlation between the unspliced and spliced counts ≥0.8 and a predicted unspliced–spliced delay greater than ≥−0.10 were retained. Manifold learning was performed using 5,000 training steps and velocity learning was performed using the 'lrmn' guide and the constant-velocity model for 10,000 training steps.

Similarly, VeloCycle was run jointly on all RG progenitor cells from the E14 and E15 time points, stratified by regional identity (FB, 2,460 cells; MB, 307 cells; HB, 176 cells) using the standard parameters. With the same gene filtering steps as with the E10 analysis above, 239 genes were used. Manifold learning was performed using 5,000 training steps and velocity learning was performed using the 'lrmn' guide and the constant-velocity model for 10,000 training steps.

To spatially visualize VeloCycle speed estimates at the E10 time point, we ran the BoneFight algorithm to map scRNA-seq clusters to a corresponding spatial transcriptomics dataset of HybISS from the same study, then colored the corresponding clusters by their velocity estimate.

**Genome-wide Perturb-seq RPE1 cells data.** To ensure analysis was performed only on RPE1 cells with a complete knockdown of the individual gene target, we filtered out cells containing nonzero unspliced or spliced UMI reads for the targeted gene[62]. VeloCycle was run initially on a subset of data in two conditions: (1) the set of control, NT cells (11,485 cells) and a grouped set of cells where a gene from the 'small' cell cycle marker list were targeted for knockdown (CC-KO, 6,275 cells). Genes used in manifold learning were those from the 'medium' gene set (DeepCycle; 218 genes) available in 'velocycle.utils', after filtering out genes with ≤0.1 mean unspliced UMIs per cell or with ≤0.2 mean spliced UMIs per cell. After an initial manifold learning step, only genes with a Pearson's correlation between the unspliced and spliced counts ≥0.7 and a predicted unspliced–spliced delay greater than ≥−0.5 were retained (120 genes remaining). Manifold learning was performed using 5,000 training steps and velocity learning was performed using the 'lrmn' guide and the constant-velocity model for 10,000 training steps.

Condition-independent estimation of the periodic Fourier series components would be especially challenging on Perturb-seq knockdown conditions containing either (1) very few cells or (2) cells belonging to just one phase of the cell cycle. Therefore, to infer accurate cell cycle phases for these cells, we first performed manifold learning for 5,000 training steps to estimate the gene harmonic coefficients ($v_0$, $v_{1sin}$ and $v_{1cos}$) on a larger set of NT and CC-KO cells, which are more evenly distributed throughout the various phases of the cell cycle. After, we ran manifold learning again for 5,000 training steps, but on the entire Perturb-seq dataset of 167,119 cells and 986 gene knockdown conditions. This time, we conditioned VeloCycle on the gene harmonic coefficients learned in the first step. This allowed cells belonging to each stratified condition to be assigned to a position on the cell cycle manifold, but restricted those assignments such that they are based on gene expression patterns learned on a larger and more-informative dataset (with the term $\Delta v$ allowing for batch effect expression differences). Finally, we performed velocity learning for 10,000 training steps on the entire dataset, estimating an individual constant velocity for each gene knockdown condition.

**RPE1 cells (newly generated for this study).** To estimate the unspliced–spliced delay and cell cycle velocity between two identical replicates of FUCCI-RPE1 cells (replicate 1, 4,265 cells; replicate 2, 9,994 cells), manifold learning was run on the 'medium' gene set available in 'velocycle.utils', after filtering out genes with ≤0.1 mean unspliced UMIs per cell or with ≤0.3 mean spliced UMIs per cell (136 genes remaining). Manifold learning was performed using 3,000 training steps and

velocity learning was performed with both the constant-velocity and periodic-velocity models for 10,000 training steps using the LRMN ('lrmn') guide.

Likewise, for the third sample of wild-type RPE1 cells (3,354 cells), manifold learning was run on the 'medium' gene set available in 'velocycle.utils', after filtering out genes with ≤0.1 mean unspliced UMIs per cell or with ≤0.3 mean spliced UMIs per cell (128 genes remaining). Manifold learning was performed using 3,000 training steps and velocity learning was performed with both the constant-velocity and periodic-velocity models for 10,000 training steps using the LRMN ('lrmn') guide.

**RPE1 cells profiled by scEU-seq.** VeloCycle was run jointly on the RPE1 cells obtained from the EU-labeling pulse experiments (2,793 cells) as previously annotated by the original study[43] and further modified by Dynamo[8]. Genes used in manifold learning were those from the 'large' gene set (GO; 1,918 genes) available in 'velocycle.utils', after filtering out genes with ≤0.1 mean unspliced UMIs per cell or with ≤0.2 mean spliced UMIs per cell. Manifold learning was performed using 5,000 training steps and velocity learning was performed using the 'lrmn' guide and both the constant-velocity and periodic-velocity models for 8,000 training steps.

**A549 cells profiled by sci-fate.** VeloCycle was run on 7,404 A549 cells using standard parameters[44]. Genes used in manifold learning were those from the 'medium' gene set (GO; 1,218 genes) available in 'velocycle.utils', after filtering out genes with ≤0.1 mean unspliced UMIs per cell or with ≤0.3 mean spliced UMIs per cell. Manifold learning was performed using 3,000 training steps and velocity learning was performed using the 'normal' guide and both the constant-velocity and periodic-velocity models for 8,000 training steps.

## Experimental procedures

**Cell culture.** FUCCI-RPE1 cells (Fig. 4), a gift from the Tanenbaum laboratory and Battich et al.[43] were cultured at 37 °C and 5% $CO_2$ in DMEM/F12 medium (Gibco 11320033) supplemented with 1% NEAA (Gibco 11140-035), 1% penicillin and streptomycin (Sigma-Aldrich G6784) and 10% FBS (Gibco 10437-028).

Additional RPE1 cells (Fig. 5) were obtained from ATCC and cultured at 37 °C, 20% $O_2$ and 5% $CO_2$ in DMEM/F12 medium (Gibco 21331-020) supplemented with 1% MEM NEAA (Sartorius, 01-340-1B), 0.5% sodium pyruvate 1% penicillin/streptomycin/glutamine and 10% FBS. Medium was replaced daily and cells were passaged twice per week. RPE1 cells were maintained in culture for at least two passages and confirmed to be free of mycoplasma.

**scRNA-seq library preparations.** For the preparation of scRNA-seq libraries, an experimental setup was designed to mimic the conditions used for live-cell imaging. FUCCI-RPE1 cells (Fig. 4) were seeded (7,000 cells per cm²) in duplicate 2 days before collection. On the collection day, cells were detached with trypsin, washed with PBS, counted and diluted to a cell concentration of 1,000 cells per μl. Barcoded cDNA libraries were generated from single cell suspensions using the 10x Genomics Chromium v.3.1 dual-index system. The procedure was carried out in accordance with the manufacturer's instructions, with a goal of 4,000 cells per library. Samples were individually indexed and evenly pooled together. After quality control, libraries were sequenced on an Illumina HiSeq4000 platform, with a depth of approximately 300 million reads per sample, by the École Polytechnique Fédérale de Lausanne (EPFL) Gene Expression Core Facility (GECF).

Similarly, RPE1 cells (Fig. 5) were detached using trypsin–EDTA solution A 0.25% (Biological Industries; 030501B) for 5 min at 37 °C. Trypsin was neutralized with medium including 10% FBS and cells were centrifuged at 250 rcf for 5 min, followed by washing and resuspension in PBS with 0.04% BSA. The cell suspension was filtered with a 40-μm

cell strainer to remove cell clumps. A cell viability percentage higher than 90% was determined by Trypan blue staining. Cells were diluted to a final concentration of 700 cells per µl. scRNA-seq libraries were generated using the 10x Genomics Chromium v.3.1 dual-index system. The procedure was carried out in accordance with the manufacturer's instructions, with a goal of 3,000 cells per library. Samples were then indexed and sequenced on an Illumina NovaSeq 6000 platform by the EPFL GECF.

As with all publicly available datasets, raw fastq files were processed with Cell Ranger using the default human reference transcriptome to obtain count matrices. To obtain unspliced and spliced count matrices, we used velocyto v.0.17.17.

**Live-image microscopy and cell-tracking experiments with RPE1 cells.** RPE1 cells were seeded on glass bottom six-well chamber slides (IBIDI) to reach 30% confluence after one day. Cells were then imaged on a PerkinElmer Operetta microscope under controlled temperature and $CO_2$ every 10.25 min using brightfield and digital phase contrast with a ×10 (0.35 NA) air objective, binning of 2 and speckle scale set to 0 under nonsaturated conditions. Cell division tracking was achieved by stacking time-course images and manually tracing cell movement and division with napari[77]. Between 20–25 RPE1 cells were tracked from 15 different fields of view by three different individuals (A.R.L., A.H. and A.V.) for a total of 337 cells used to estimate a ground truth.

**Cumulative EdU and p21 staining experiments.** Cells were seeded on poly-L-lysine-coated 24-well plates to reach 30% confluence after one day. After a day, 10 µM EdU (Invitrogen, A10044) was added to the medium and cells were fixed at different time points after EdU addition: 30 min, 1 h, 2 h, 3 h, 5 h, 8 h, 12 h, 24 h, 32 h, 36 h, 49 h, 56 h and 72 h. For each time point, cells were fixed in 4% PFA for 10 min, washed twice with PBS and processed for EdU detection according to the manufacturer's instructions (Click-iT EdU Alexa Fluor 647 Imaging kit from Invitrogen, C10340). Additionally, cells were permeabilized with 0.2% Triton and stained overnight at 4 °C with p21 Waf1/Cip1 (12D1) rabbit monoclonal antibody (Cell Signaling Technology, 2947) and revealed with a secondary antibody conjugated to Alexa Fluor-488. After staining, cells were imaged on a Leica DMi8 (×20 NA 0.8).

To quantify the signal intensities of p21 and EdU, we segmented nuclei in the DAPI channel using stardist[78]. We obtained the average intensity for both signals per nucleus by subsetting the corresponding channel using segmentation masks. The intensity of p21 was normalized per image (percentile-based, p_min = 1, p_max = 99.8), as its intensity profile was expected to be approximately constant in time; conversely, the intensity of EdU was not normalized as it was expected to increase with time. Thresholds were selected observing the (bimodal) signal distribution across nuclei in all time points.

First, to compute the time it takes, on average, for a cell to traverse through two consecutive S phases ($t$EdU), we applied the Nowakowski method[79] on data collected at multiple time points for a total number of 678,204 cells. The Nowakowski method assumes a linear growth of EdU+ cells, until reaching a plateau where all cycling cells are positive for EdU. We obtained a linear fit of the growth and determined the $x$ value at the intersection between growth and plateau (A), the $y$ intercept of the linear fit (B) and the $y$ value of the plateau (GF). With these, we could compute $t$EdU as follows: $t$EdU = (B × A)/(GF − B) + A.

However, cells may on occasion exit the cycle to a G0 phase and then re-enter at a later time[80]. To correct for this, we plotted the fraction of EdU+/p21+ cells among the p21+ population to estimate the G0 duration ($t$G0). We determined the $t$G0 to be equal to the time point at which fraction of EdU+/p21+ cells plateaued, after which no statistically significant changes were detected (Tukey's multiple comparison test). The corrected estimate for the cell cycle duration was finally calculated as: $t$c = $t$EdU − (%p21 × $t$G0), where %p21 corresponded to the mean fraction of p21+ cells.

## Advanced computational analyses

**Noncycling cell contamination analyses.** To evaluate the robustness of VeloCycle estimates to the presence of noncycling (G0) contaminants, we performed sensitivity analysis constructed by spiking in different proportions of G0 cells into a pool of cycling cells. Across ten simulated datasets (Fig. 2), progressively added more simulated noncycling cells (with a simulated ground-truth velocity of 0), from 0–200 noncycling cells per 100 cycling cells and performed standard VeloCycle estimation. The phase assignments and velocity were then compared to the simulated ground truth using the circular correlation coefficient and percent error, respectively. Additionally, for the real dataset of human dermal fibroblasts[40], we spiked-in actual noncycling cells from the same dataset that were annotated in the original study as belonging to nonproliferative clusters (from 0–200 noncycling cells per 100 cycling cells). To evaluate performance of the phase and velocity estimates, we considered the results obtained without any noncycling cells present as the reference against which to compare.

**Benchmarking with other velocity methods.** We perform benchmarking between VeloCycle and four other methods (scvelo[4], cellDancer[45], Dynamo[8] and VeloVAE[18]) on the following datasets: simulated data (previously shown in our sensitivity analysis in Fig. 2), hDFs[40] (Fig. 5a), A549 cells from ref. 44 and RPE1 cells (generated for this study; Fig. 5h). For each dataset, we used the same set of gene features across all methods (simulated data, 300 genes; human dermal fibroblasts, 162 genes; A549 cells, 203 genes; RPE1 cells, 128 genes) but otherwise applied the standard parameters recommended by the authors of the original studies.

**Cross-boundary direction correctness scoring.** We calculated the cross-boundary direction correctness (CBDir) as originally proposed[6] using the functions provided by the UniTVelo package[5] in Python. CBDir scores measure the correctness of cell transitions from a source cluster to a target cluster by examining the vector field directionality in cells nearby the source-target cluster boundary[5] (−1, lowest CBDir correctness; +1, highest CBDir correctness). This metric requires user-defined information on the expected order of source and target clusters. First, we defined clusters on the low-dimensional embedding obtained with each velocity method benchmarked (VeloCycle, scvelo, Dynamo, cellDancer and VeloVAE). Then, we manually defined the pairs of source and target clusters using ground-truth knowledge from the categorical cell cycle phases. These clusters were used pairwise to obtain a mean CBDir for each combination of 'velocity method' and 'velocity method-derived embedding-defined clusters'. To account for differences in the embedding dimensions (one dimension for VeloCycle and two dimensions for UMAPs in the other methods), the number of neighbors used to select cells at the source-target cluster boundary was chosen to include all neighbors within 1 × s.d. of the mean intra-cluster cell distance.

**Velocity consistency scoring.** We calculated the velocity consistency scores for each cell as previously described in the work introducing the VeloVI package[21]. In summary, velocity consistency evaluates the correlation of the velocity vector field of a particular cell with the vector fields of its nearest neighbors (0, lowest consistency; 1, highest consistency). We computed the mean consistency score across all cells in each dataset using the 200 nearest neighbors, as determined using Euclidean distance in the low-dimensional embedding space (UMAP, $\varphi$) of each respective velocity method.

**Least action path analysis.** LAP analysis was performed in a dataset of RPE1 cells profiled by scEU-seq[43] using the standard pipeline and tutorials in Dynamo[8]. Specifically, the `dyn.pd.least_action` function was run with 20 init_cells taken from the earliest points of the cell cycle and 20 target_cells taken from the latest points of the cell cycle (as determined using the FUCCI-based 'Cell_cycle_possition'

attribute in the published dataset). LAP was run using both the velocity results obtained by Dynamo's metabolic modeling function (Dynamo-Metabolic; uses pulse time course and metabolically labeled information) and Dynamo's stochastic model (Dynamo; uses only spliced and unspliced counts). VeloCycle was run on the same dataset of cells using the same filtering criteria and gene set to effectively compare results with the LAP estimates.

**1D interval model analysis of pancreatic endocrinogenesis.** The 1D interval model of manifold-constrained velocity was run on 1,825 single cells belonging to the clusters 'Ngn3 high EP', 'Pre-endocrine' and 'Beta' from the E15.5 time point of pancreatic endocrinogenesis[63]. Gene feature selection was achieved by selecting the top 4,000 highly variable genes using the scanpy 'sc.pp.high_variable_genes' function, followed by filtering genes with '0.1 mean unspliced UMIs per cell or with ≤0.3 mean spliced UMIs per cell (263 genes remaining). Manifold learning was performed using 2,000 training steps to learn the B-spline coefficients describing gene expression patterns, with the model being conditioned on the pseudotime (phi) obtained by 'sc.tl.diffmap' and scaled between 0 and 10. Velocity learning was performed with a constant velocity for 6,000 training steps.

**1D interval model analysis of developing mouse dentate gyrus.** To evaluate velocity differences at the P0 and P5 time points within the same developmental lineage, the 1D interval model of manifold-constrained velocity was run on 6,844 single cells belonging to the clusters 'Nbl1', 'Nbl2', 'ImmGranule1', 'ImmGranule2' and 'Granule' from the P0 and P5 time points of the dentate gyrus[64]. Gene feature selection was achieved by selecting the top 8,000 highly variable genes using the scanpy 'sc.pp.high_variable_genes' function, followed by filtering genes with ≤0.1 mean unspliced UMIs per cell or with ≤0.2 mean spliced UMIs per cell (237 genes remaining). Manifold learning was performed using 2,000 training steps to learn the B-spline coefficients describing gene expression patterns, with the model being conditioned on the pseudotime (phi) obtained by 'sc.tl.diffmap' and scaled between 0 and 10. Velocity learning was performed with a constant velocity for 6,000 training steps.

To evaluate the CA2-3-4 lineage, the 1D interval model of manifold-constrained velocity was run on 18,213 single cells belonging to the clusters 'Nbl1', 'Nbl2', 'CA' and 'CA2-3-4' from the P0 and P5 time points of dentate gyrus[33]. Gene feature selection was achieved by selecting the top 8,000 highly variable genes using the 'sc.pp.high_variable_genes function', followed by filtering genes with ≤0.1 mean unspliced UMIs per cell or with ≤0.2 mean spliced UMIs per cell (408 genes remaining). Manifold learning was performed using 2,000 training steps to learn the B-spline coefficients describing gene expression patterns, with the model being conditioned on the pseudotime (phi) obtained by 'sc.tl.diffmap' and scaled between 0 and 10. Velocity learning was performed with a constant velocity for 6,000 training steps.

### Reporting summary
Further information on research design is available in the Nature Portfolio Reporting Summary linked to this article.

## Data availability
The raw and processed scRNA-seq data in the RPE1 cell line that was newly generated for this study are available at the Gene Expression Omnibus accession no. GSE250148. All other scRNA-seq data used in this study were collected from previously published works and relied on the cell-type annotations made by the original authors. These works are cited appropriately throughout the article. Jupyter notebooks and other affiliated files to reproduce the results shown in this study are provided via our GitHub page at https://github.com/lamanno-epfl/velocycle/. Processed versions of all published data (including spliced–unspliced counts matrices) are also available at the above link.

The simulated scRNA-seq datasets, processed scRNA-seq metadata for the new RPE1 samples, cell-tracking data from live-image microscopy and cumulative EdU staining experiments are also available at the above link.

## Code availability
VeloCycle is implemented in Python and available as an open-source package on GitHub at https://github.com/lamanno-epfl/velocycle. VeloCycle can be installed from PyPi using the command pip install VeloCycle or via direct installation from the GitHub page using the command pip install 'git+https://github.com/lamanno-epfl/velocycle.git@main'. Python v.3.8 or newer is required. Source code, installation instructions, tutorials and a file containing all required package dependencies are also available on GitHub. Additional code and notebooks to reproduce the results of this study are available on Zenodo at https://doi.org/10.5281/zenodo.12517650 (ref. 81).

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

## Acknowledgements
This project has been made possible in part thanks to Chan Zuckerberg Initiative grant nos. 2022-249212 and 2019-002427. G.L.M. received support from the Swiss National Science Foundation grant PZ00P3_193445. F.N. received support from the Swiss National Science Foundation grant 310030B_201267 and the EPFL. L.P. is partially supported by the National Human Genome Research Institute Genomic Innovator Award (R35HG010717). We thank members of the La Manno, Naef, Pinello, Williams and Castelo-Branco laboratories for their generous feedback and discussions on the project, particularly C. Smith, Q. Qin, E. Bingham, L. Seeker, N. Bestard, M. Kabbe and F. Baldivia Pohl. We also thank staff at the EPFL GECF for their assistance with scRNA-seq experiments.

## Author contributions
A.R.L. developed the idea, designed, implemented and refined the model framework, analyzed the scRNA-seq and time-lapse microscopy data, created the figures and wrote the paper. M.L., L.T. and C.D. participated in the development of the idea and model formulation. D.M.B. assisted with conceptualization, implementation, validation and figure design of model extensions for the 1D nonperiodic and 2D manifold cases. A.H., A.V., I.K. and H.J.F.C. performed scRNA-seq experiments. A.H. and A.V. also conducted time-lapse microscopy and cumulative EdU experiments, with image-processing analysis assistance from A.D.M. P.M.A. helped test phase-estimation approaches. L.P. helped refine the idea and related analyses. F.N. and G.L.M. developed the idea,

supervised the project and wrote the manuscript. All co-authors read and approved the paper.

## Competing interests

The authors declare no competing interests.

## Additional information

**Extended data** is available for this paper at https://doi.org/10.1038/s41592-024-02471-8.

**Correspondence and requests for materials** should be addressed to Felix Naef or Gioele La Manno.

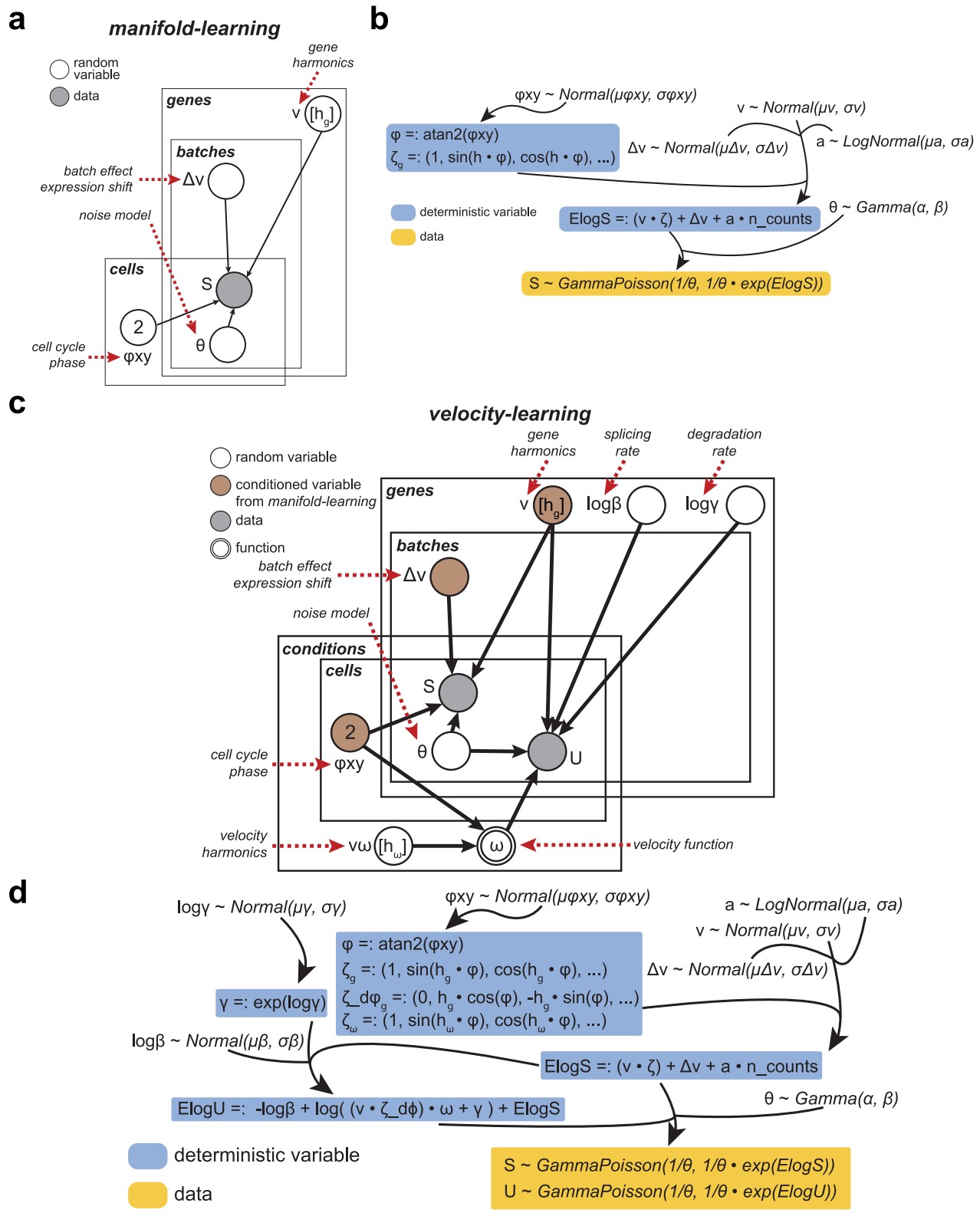

**Extended Data Fig. 1 | Plate notation diagram and mathematical formulation of VeloCycle. (a)** Plate diagram of the *manifold learning* procedure. The model assigns each cell to a phase along the cell cycle (φ) and fits a set of Fourier series coefficients (v) for each gene. **(b)** Mathematical representation of manifold learning shown in (a). Raw spliced counts (S) are defined as the expectation (ElogS) plus noise, modeled after a negative binomial distribution. **(c)** Plate notation diagram of the complete velocity learning procedure. **(d)** Mathematical representation of velocity learning shown in (c). In (a) and (c), nodes indicate a variable (white: random variable; gray: observed data; brown: conditioned variable from manifold learning) and arrows indicate dependency. Plates (genes, cells, conditions, and batches) signal independence and contain variables with the same dimensions. In (b) and (d), blue-boxed variables are deterministic and computed from latent variables; yellow-boxed variables are conditioned on observed data.

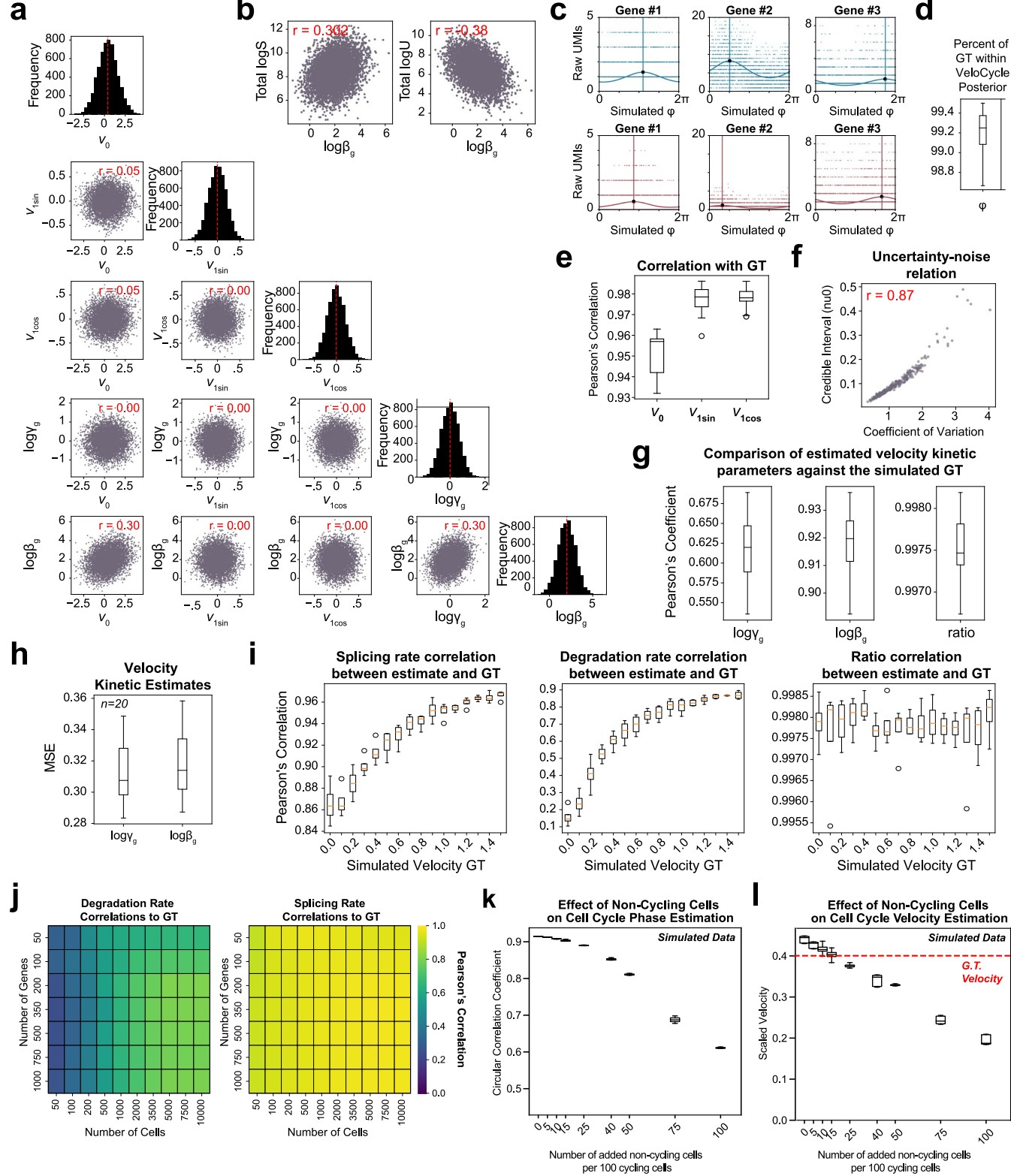

**Extended Data Fig. 2 | See next page for caption.**

**Extended Data Fig. 2 | Data generated with simulations assists in VeloCycle validation. (a)** Scatter-plots of correlation between gene harmonics coefficients ($v_0$, $v_{1sin}$, $v_{1cos}$) and kinetic parameters ($\log\beta_g$, $\log\gamma_g$) in the ground-truth (GT) from simulated data. Diagonal: histograms for each simulated latent variable. **(b)** Scatter-plots of simulated data correlations among splicing rate ($\log\beta_g$), spliced ($\log S$), and unspliced ($\log U$) counts. **(c)** Scatter-plots of simulated gene fits for spliced (blue) and unspliced (red) counts. Solid curved lines represent gene fits; vertical lines indicate peak expression. **(d)** Box plot of the percent of GT phases within the uncertainty interval estimated (min: 98.6%, max: 99.5%, median: 99.25%), across 20 simulated datasets. **(e)** Box plots of the mean circular correlation coefficient, across 300 genes, for $v_0$ (min: 0.93, max: 0.96, median: 0.96), $v_{1sin}$ (min: 0.96, max: 0.99, median: 0.98), and $v_{1cos}$ (min: 0,97, max 0.99, median: 0.98) estimated by VeloCycle compared to the GT. **(f)** Scatter-plot of gene-wise coefficient of variation (a measure of noise) and credible interval obtained for $v_0$. **(g)** Box plots of the mean Pearson's correlation coefficient between estimated and GT gene-wise values for degradation rate ($\log\gamma_g$; min: 0.54; max: 0.69; median: 0.62), splicing rate ($\log\beta_g$; min: 0.89; max: 0.94; median:

0.92), and kinetic ratio ($\log\gamma_g$-$\log\beta_g$; min: 0.996; max: 0.998; median: 0.997), across 20 simulated datasets. **(h)** Box plots of mean squared error (MSE) for $\log\gamma_g$ and $\log\beta_g$ against the GT for data in (g). **(i)** Box plots of mean Pearson's correlation coefficient between estimated and GT values for $\log\beta_g$, $\log\gamma_g$, and kinetic ratio, for all genes across four simulations with 16 different velocity GT between 0.0 and 1.5. For each box plot, the orange horizontal line represents the median across four datasets. **(j)** Heatmaps showing the correlation between estimated and GT values for the kinetic parameters using varying numbers of cells and genes. **(k)** Box plots of the circular correlation coefficient between estimated and GT phase across three simulated datasets with varying proportions of non-cycling cells (from 0 to 100 non-cycling cells per 100 cycling cells). **(l)** Box plots of scaled velocity posterior estimates compared to the GT (red dashed line) across three simulated datasets with varying proportions of non-cycling cells. Pearson's correlation coefficients (red) are indicated in each scatter-plot of (a), (b), and (f). Each purple dot represents a single gene in (a), (b), and (f). For box plots in (d-e), (g), (i), and (k-l), boundaries are defined by the interquartile range (IQR), and whiskers extend each box by 1.5x the IQR.

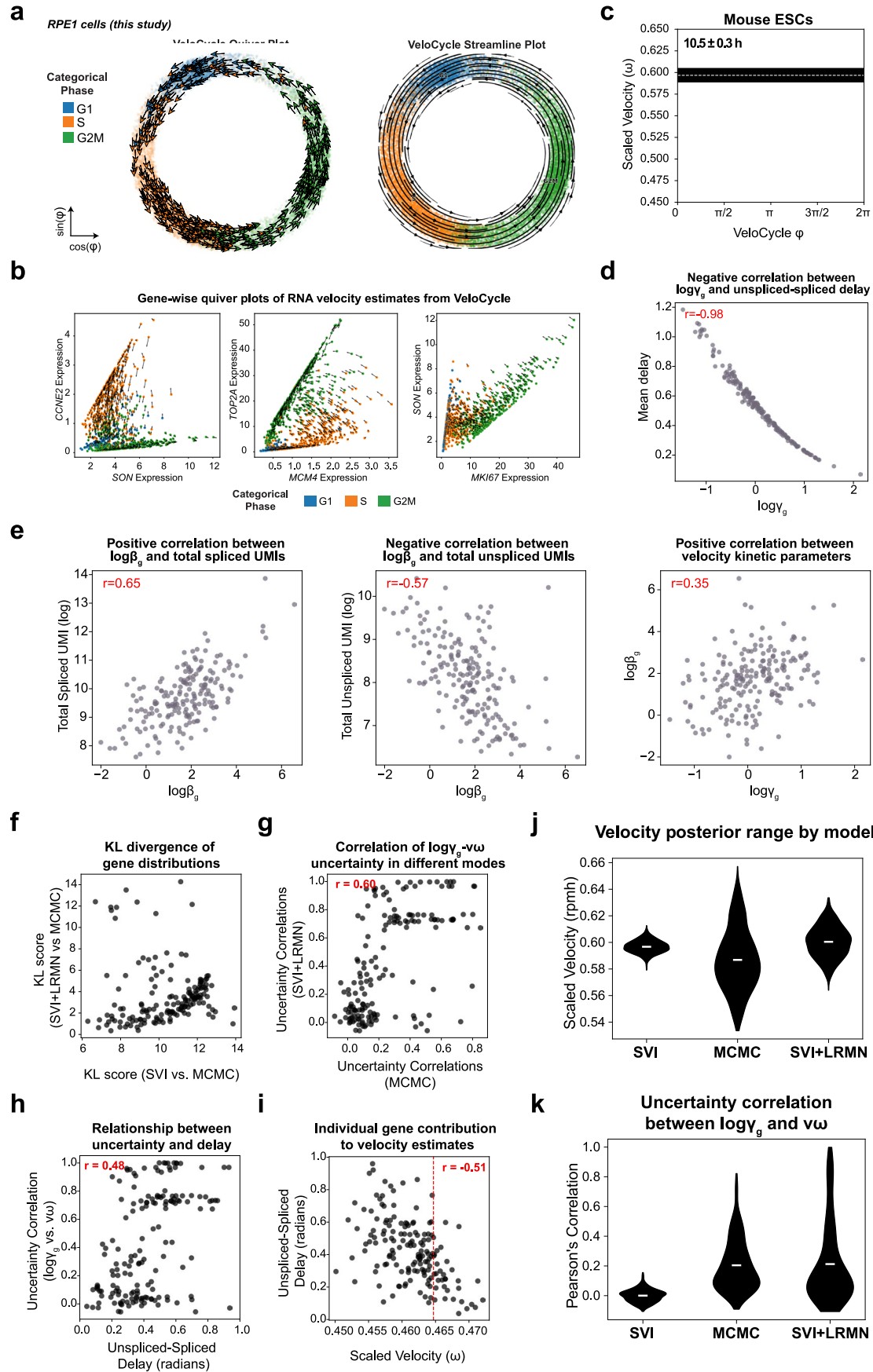

**Extended Data Fig. 3 | See next page for caption.**

**Extended Data Fig. 3 | VeloCycle reveals relationships among kinetic parameters and a structured variational distribution yields better uncertainty estimates. (a)** Velocity quiver plot (left) and streamline plot (right) for 14,259 RPE1 cells from Fig. 4a-e, colored by categorical phase assignments. **(b)** Gene-wise velocity quiver plots for three marker gene pairs corresponding to distinct categorical phases in RPE1 cells: G1 marker *SON* and S marker *CCNE2*; S marker *MCM4* and G2M marker *TOP2A*; G2M marker *MKI67* and G1 marker *SON*. **(c)** Posterior estimate plot of cell cycle velocity inferred on mouse embryonic stem cells (mESC)[37]. White dashed lines represent the mean of 500 posterior samples; black bars indicate the full posterior interval. **(d)** Scatter-plot of the relationship between degradation rate ($\log\gamma_g$) and average un/spliced delay in mESC. **(e)** Scatter-plots of the relationships among splicing rate ($\log\beta_g$), $\log\gamma_g$, and total UMI counts (un/spliced) in mESC. **(f)** Scatter-plot of gene-wise Kullback–Leibler (KL) divergence comparing uncertainty distributions between SVI and MCMC (x-axis) and SVI + LRMN and MCMC (y-axis) for dermal human fibroblasts (dHF)[37] from Fig. 4g-k. A lower KL divergence indicates a greater overlap between the two distributions. **(g)** Scatter-plot between the gene-wise $\log\gamma_g$-$v\omega$ uncertainties computed from the posterior of MCMC or SVI + LRMN for dHF. **(h)** Scatter-plot between un/spliced peak expression delay (radians) and $\log\gamma_g$-$v\omega$ uncertainty correlation, both obtained using the SVI + LRMN velocity model. **(i)** Scatter-plot between scaled velocity and un/spliced delay during a leave-one-out estimation approach. Each dot is positioned on the x-axis at the velocity estimate obtained when removing one particular gene (n = 160) from the gene set. Each dot is located on the y-axis at the position of the un/spiced delay (in radians) for that removed gene. **(j)** Violin plots of the scaled velocity for mESC, comparable to Fig. 4h. **(k)** Violin plots of the Pearson's correlations between $\log\gamma_g$ and angular speed ($v\omega$) posteriors across all 189 genes for mESC, comparable to Fig. 4i. Pearson's correlation coefficients (red) are indicated in the top right of plots in (d-e), and (g-i).

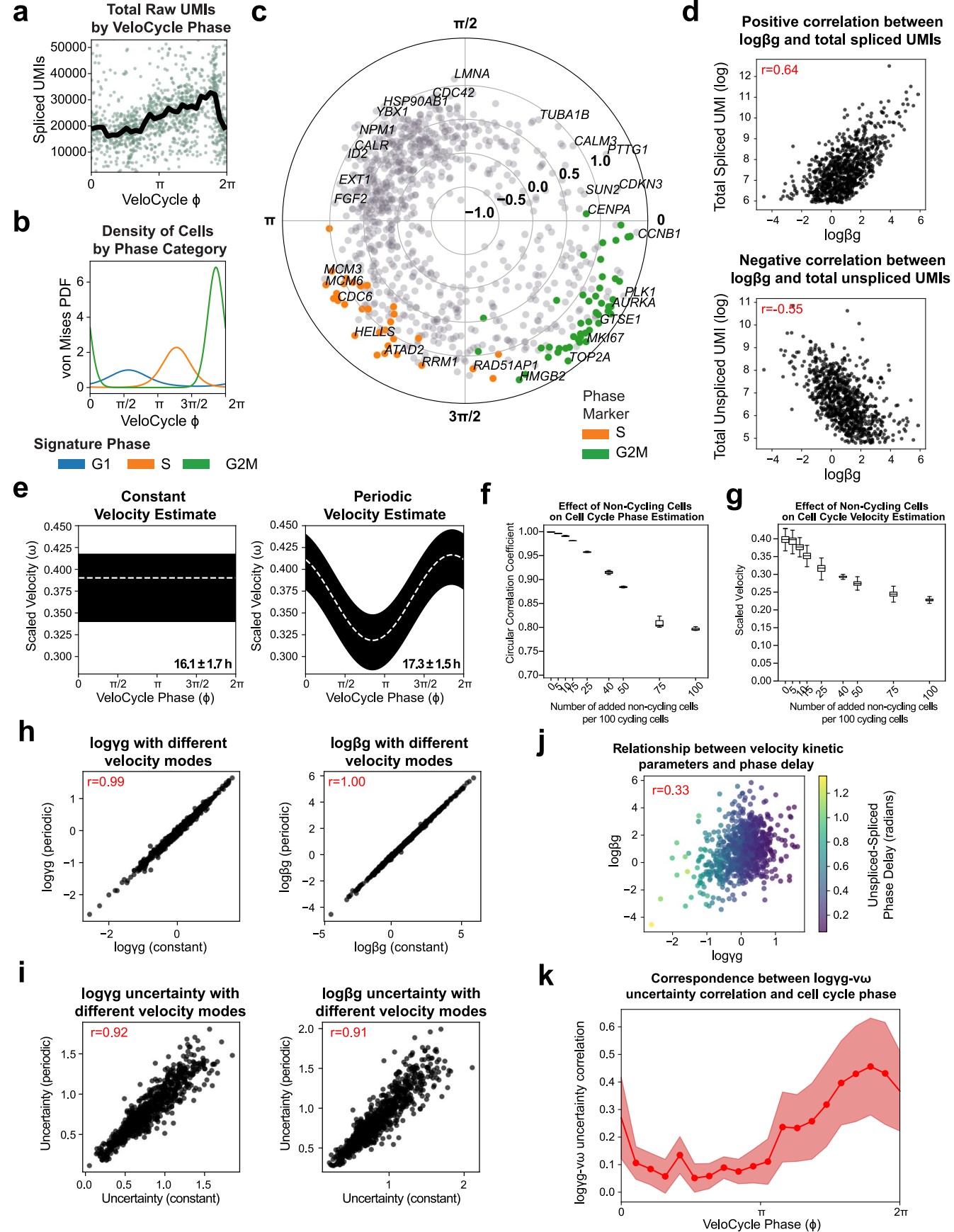

**Extended Data Fig. 4 | See next page for caption.**

**Extended Data Fig. 4 | VeloCycle coupled with live-cell imaging of human fibroblasts enables experimental validation of cell cycle speed.** (**a**) Scatter-plot of total raw spliced counts by cell cycle phase estimated with VeloCycle for dHF[40]. Black line indicates the binned mean. (**b**) Probability density plot along the VeloCycle phase estimate for cells in (a), stratified by categorical assignment. (**c**) Polar plot indicating phase of peak expression and amplitude for 876 cycling genes. Each dot represents a gene; genes colored orange (S) or green (G2M) represent marker genes used in categorical assignment[26]. (**d**) Scatter-plots of the gene-wise relationship among splicing rate and un/spliced counts. (**e**) Posterior estimate plot of constant (left) and periodic (right) velocity estimates obtained for data in (a) using a medium-sized gene set[37]. (**f**) Box plots of circular correlation coefficients between phase with varying proportions of non-cycling cells and phase with only cycling cells (from a-c). Contaminant cells were taken from non-cycling cell types, as annotated in the original study[40]. (**g**) Box plots of scaled velocity posterior estimates with varying proportions

of non-cycling cells. The box plots in (f) and (g) indicate the results across three independently sampled subsets of non-cycling cells. (**h**) Scatter-plots of degradation rates (left) and splicing rates (right) obtained using either constant (x-axis) or periodic (y-axis) models of velocity estimation. (**i**) Scatter-plots of degradation (left) and splicing (right) rate posterior uncertainties obtained from 500 posterior samples using either constant (x-axis) or periodic (y-axis) models. (**j**) Scatter-plot of the degradation and splicing rates obtained with the SVI + LRMN model. Gene-wise dots are colored by the un/spliced phase delay. (**k**) Binned plot of Pearson's correlation coefficients between gene-wise degradation rate and velocity posterior uncertainties on dHF using the SVI + LRMN model. The solid red line indicates binned mean delay; the red bands indicate one standard deviation. Pearson's correlations coefficients are indicated in red text in (d-e) and (h-i). Boundaries in (f) and (g) are defined by the interquartile range (IQR), and whiskers extend each box by 1.5x the IQR.

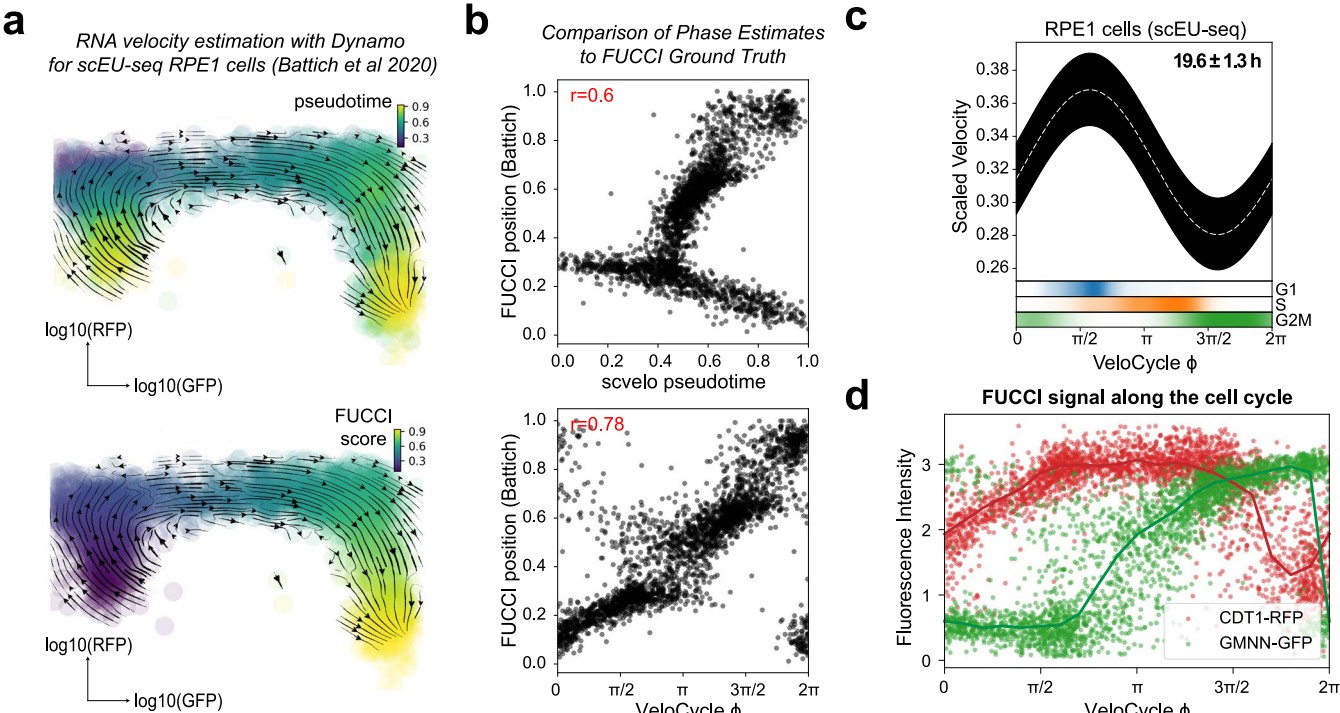

**Extended Data Fig. 5 | Least action path analysis of FUCCI-RPE1 cells profiled by scEU-seq.** (**a**) RNA velocity streamline plots obtained with Dynamo[8] using metabolically-labeled data (scEU-seq) for 2,793 RPE1 cells[43], represented on the FUCCI-defined embedding space and colored by pseudotime (top) or FUCCI score (bottom). (**b**) Scatter-plot of cell cycle phase pseudotime from scvelo[4] (top) and VeloCycle (bottom) compared to the ground truth FUCCI phase assignment acquired for RPE1 cells in (**a**). Pearson's circular correlation coefficient is indicated in red. (**c**) Posterior estimate plot of velocity estimate for RPE1 cells obtained with VeloCycle. (**d**) Scatter-plot of per cell FUCCI signal (GMNN-GRP in green; CDT1-RFP in red) along the VeloCycle phase.

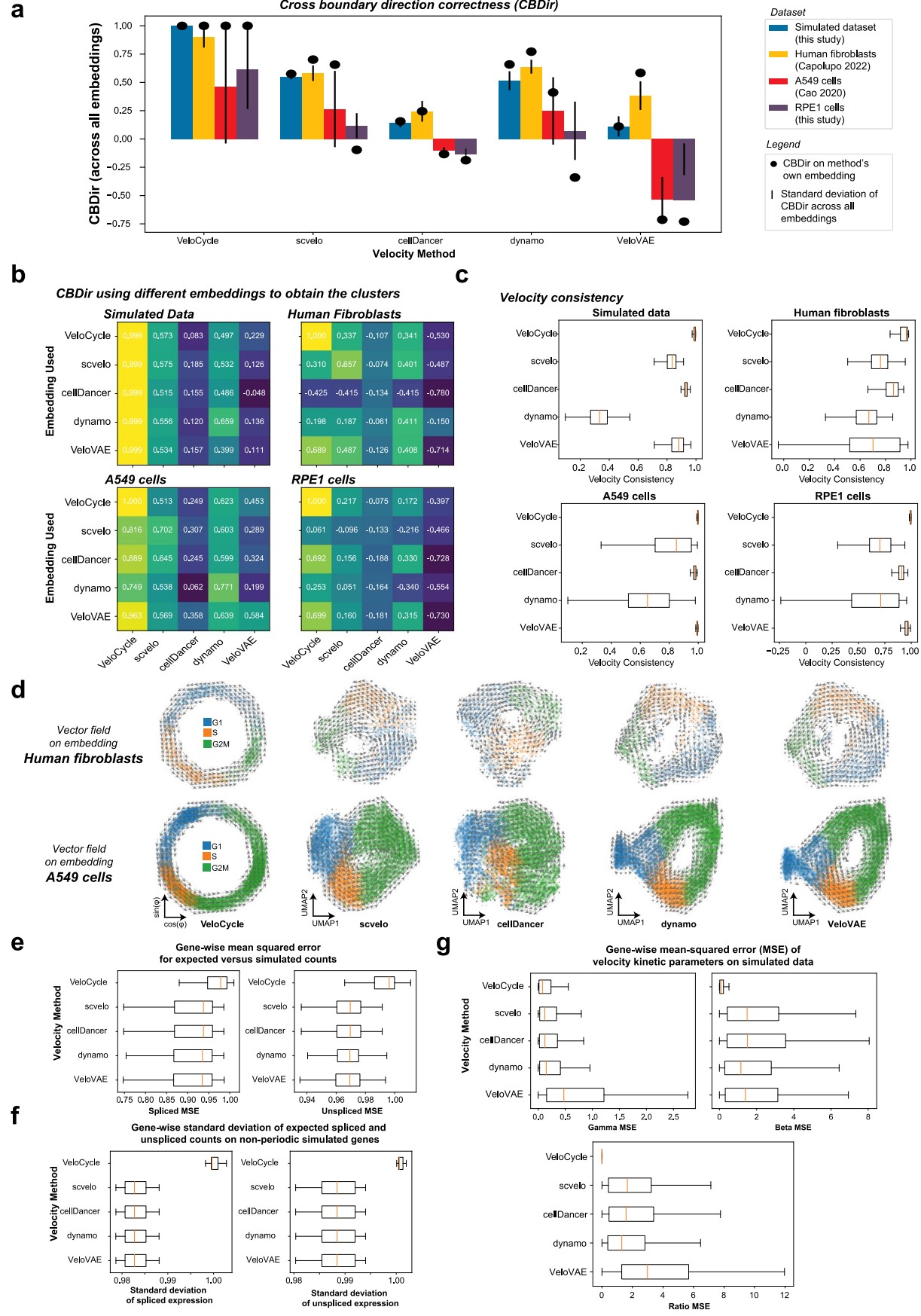

**Extended Data Fig. 6 | See next page for caption.**

**Extended Data Fig. 6 | Benchmarking *VeloCycle* against numerous methods, metrics, and datasets.** (**a**) Bar plots of mean cross-boundary direction correctness (CBDir)[6] computed on four datasets with five velocity methods. The CBDir score was computed for clusters obtained on each method's embedding (**see Methods 8.3**). The black dot represents the score calculated using that specific method's embedding, whereas the black line indicates the standard deviation across scores computed across all five embeddings and cluster annotations. (**b**) Heatmaps of the CBDir scores computed pairwise for each method using cluster relationships defined on each embedding, for all four datasets. (**c**) Box plot of cell-wise velocity consistency score[21] (**see Methods 8.4**) computed on the same datasets and methods as in (a-b). (**d**) Low-dimensional embedding plots with grid-wise velocity vector fields computed for on fibroblasts (top) and A549 cells (bottom). Cells are colored by categorical phase to enable visual inspection of vector field direction correctness. (**e**) Box plots of the spliced (left) and unspliced (right) gene-wise mean squared error (MSE) obtained between expected un/spliced counts and simulated GT. (**f**) Box plots of gene-wise standard deviation of expected spliced (left) and unspliced (right) expression estimated. (**g**) Box plots of gene-wise MSE for the velocity kinetic parameters ($\log\gamma_g$, $\log\beta_g$, kinetic ratio) compared to the simulated GT. For box plots in (c), (e), (f), and (g), boundaries are defined by the interquartile range (IQR), and whiskers extend each box by 1.5x the IQR.

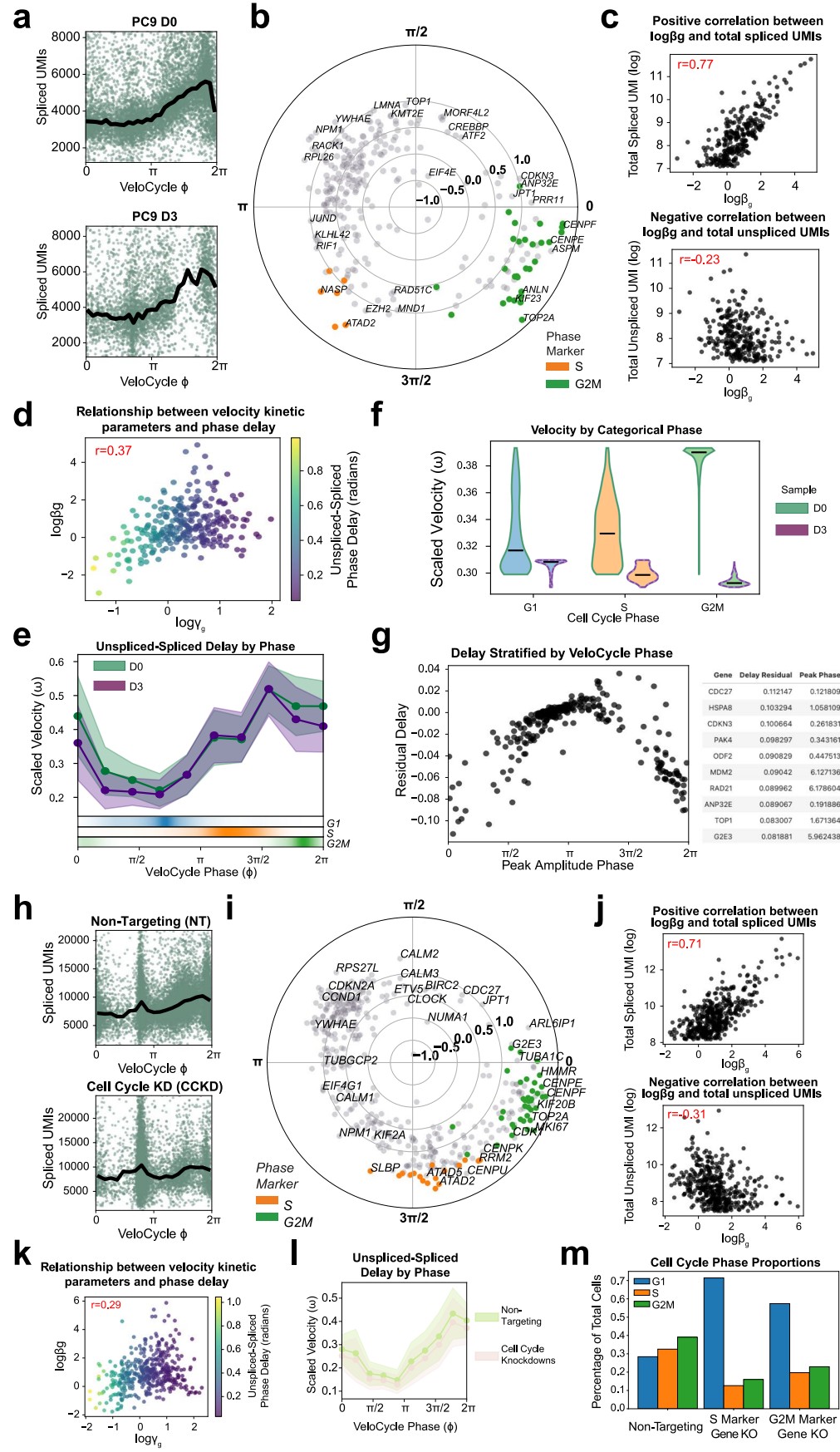

**Extended Data Fig. 7 | See next page for caption.**

**Extended Data Fig. 7 | Velocity credibility testing enables characterization of erlotinib treatment and cell cycle knockdowns with Perturb-seq. (a)** Scatter-plot of total raw spliced counts by phase estimated for PC9 cells populations before (D0; 9,927 cells) and after (D3; 3,943 cells)[46] erlotinib treatment. Black line indicates the binned mean. **(b)** Polar plot indicating peak expression and amplitude for cycling genes used in (a). Each dot represents a gene; genes colored orange (S) or green (G2M) represent marker genes used in categorical phase assignment. **(c)** Scatter-plots of the gene-wise relationship among splicing rate and un/spliced counts for data in (a-b). **(d)** Scatter-plot of degradation and splicing rates obtained with the SVI + LRMN model. Gene-wise dots are colored by un/spliced phase delay. **(e)** Gene-binned delay in D0 and D3 samples. Solid green (D0) and purple (D3) lines indicate the binned mean delay; bands indicate one standard deviation. **(f)** Violin plots of scaled velocity estimates for D0 and D3, stratified by phase (G1: 2,738 cells at D0 and 1,954 cells at D3; S: 2,287 cells at D0 and 900 cells at D3; G2M: 2,902 cells at D0 and 889 cells at D3). Black horizontal lines indicate the mean by categorical phase at D0 (G1: 0.33, S: 0.34, G2M: 0.39) and D3 (G1: 0.31, S: 0.30, G2M: 0.30). **(g)** Left: scatter-plot of peak gene amplitude and residual un/spliced delay (D3-D0) for 273 genes. Right: top ten differentially delayed genes in D0 versus D3. **(h)** Scatter-plots of total UMIs along cell cycle phase for non-targeting (NT; top) and knockdown (CC-KD) strata of genome-wide Perturb-seq data from Fig. 6. **(i)** Polar plot indicating peak expression and amplitude for 426 cycling genes used in (h). **(j)** Scatter-plots of the gene-wise relationship among splicing rate and un/spliced counts for data in (h-i). **(k)** Scatter-plot of degradation and splicing rates; gene-wise dots are colored by mean un/spliced phase delay. **(l)** Gene-binned delay between maximum expression (in radians) for NT and CC-KD samples. Solid green (non-targeting) and beige (cell cycle knockdowns) lines indicate binned mean delay; bands indicate one standard deviation. **(m)** Bar plots of categorical phase proportions as percentage of the total number of cells, stratified by non-targeting, S phase, and G2M marker conditions.

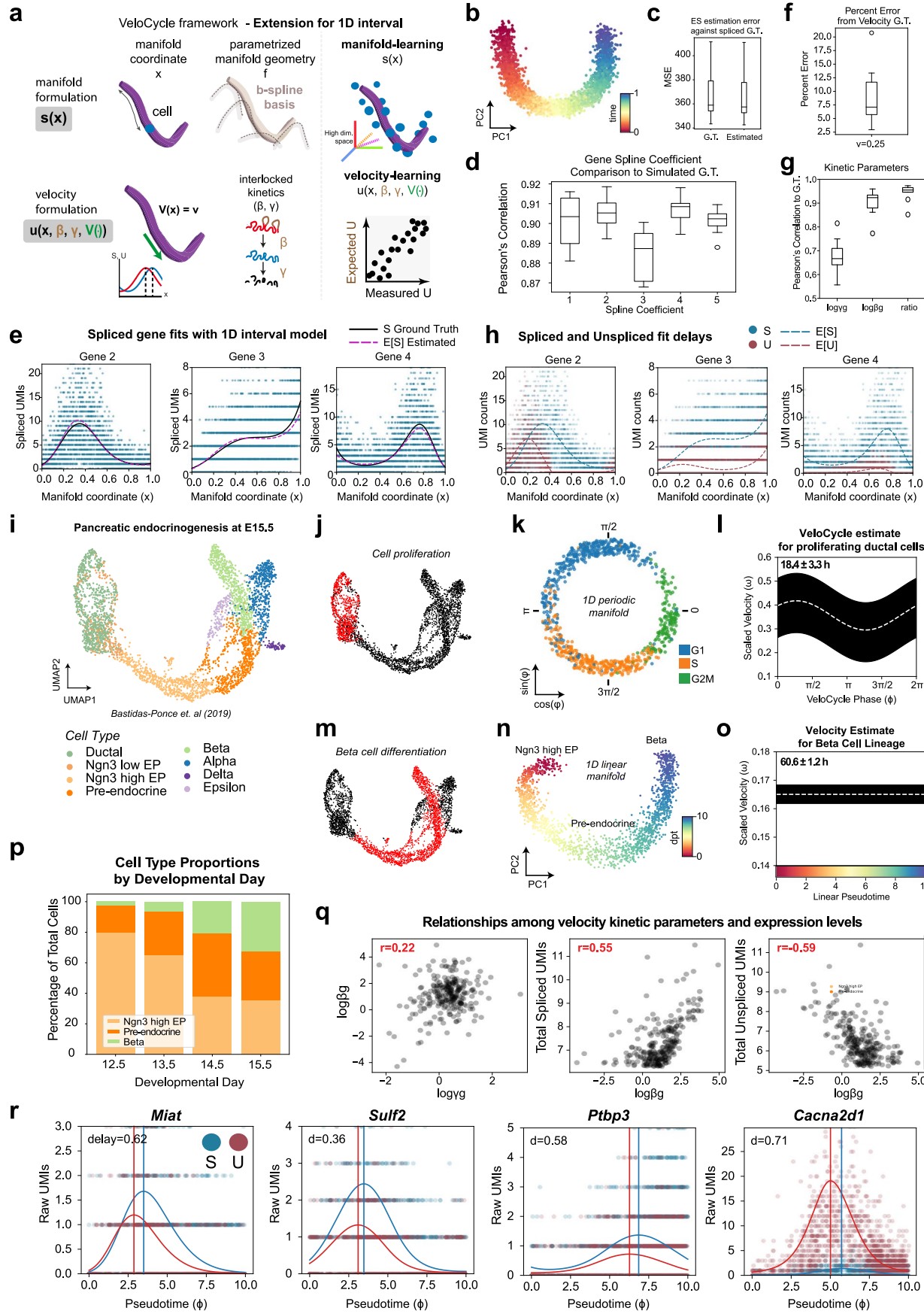

**Extended Data Fig. 8 | See next page for caption.**

**Extended Data Fig. 8 | Evaluation of a 1D interval model for manifold-constrained RNA velocity in simulated data and during pancreatic endocrinogenesis. (a)** Schematic of the 1D interval manifold model, where variation in gene expression along the manifold is estimated using B-splines instead of a Fourier series (as in *VeloCycle*). **(b)** PCA plot of two principal components colored by the manifold coordinate used (time). **(c)** Box plots of log mean squared error (MSE) for expected spliced counts (ES) compared to simulated raw data (S). MSE was calculated using either simulated ground truth (GT) or estimates recovered by the non-periodic manifold-constrained model (Estimated). **(d)** Box plots of the Pearson's correlation coefficient between estimated B-spline coefficients and GT. **(e)** Scatter-plots of spliced gene expression fits along the 1D interval manifold. The solid black line indicates GT; the red dashed line indicates the estimate obtained during manifold learning. **(f)** Box plot of percent error for the velocity estimate compared to GT (0.25; min: 0.7%, max: 20.0%, median: 4.7%). **(g)** Box plot of Pearson's correlation coefficients between estimated and GT for kinetic parameters. **(h)** Scatter-plots of spliced (blue) and unspliced (red) gene expression fits obtained by the model. **(i)** UMAP of mouse E15 pancreas[63], colored by published cell types. **(j)** UMAP of dataset in (i), colored by selected cell subsets (red) extracted to estimate cell cycle velocity (Ductal). **(k)** Low-dimensional plot of the cell cycle

manifold estimated with VeloCycle. **(l)** Posterior estimate plot of cell cycle speed from VeloCycle. **(m)** UMAP of dataset in (i), colored by selected cell subsets (red) along the beta cell differentiation lineage (Ngn3 high EP, Pre-endocrine, Beta). **(n)** PCA plot of beta differentiation manifold obtained with diffusion pseudotime on the principal components. **(o)** Velocity posterior estimate plot obtained for beta differentiation using the 1D interval model. **(p)** Stacked bar plot of cell type proportions along the differentiation axis in four datasets from the original study. **(q)** Scatter-plots demonstrating the relationship between the kinetic parameters ($\log\gamma_g$, $\log\beta_g$) and total (spliced, unspliced) counts. Pearson's correlation coefficients are indicated in red. **(r)** Scatter-plots of selected genes, illustrating the estimated expected spliced (blue dashed line; ES) and unspliced (red dashed line; EU) levels along the cell cycle manifold, compared to the measured spliced (blue; S) and unspliced (red; U) counts. In (a-h), all analyses were performed across ten simulated datasets with 3,000 cells and 300 genes (see Fig. 2). In (l) and (o), the white line indicates the mean over 200 posterior samples; the black line indicates the full posterior interval. The cell cycle period (l) and beta cell differentiation process time (o) are indicated at the top left of the respective plots. For each box plot in (c-d) and (f-g), the black horizontal line represents the median; boundaries are defined by the interquartile range (IQR), and whiskers extend each box by 1.5x the IQR.

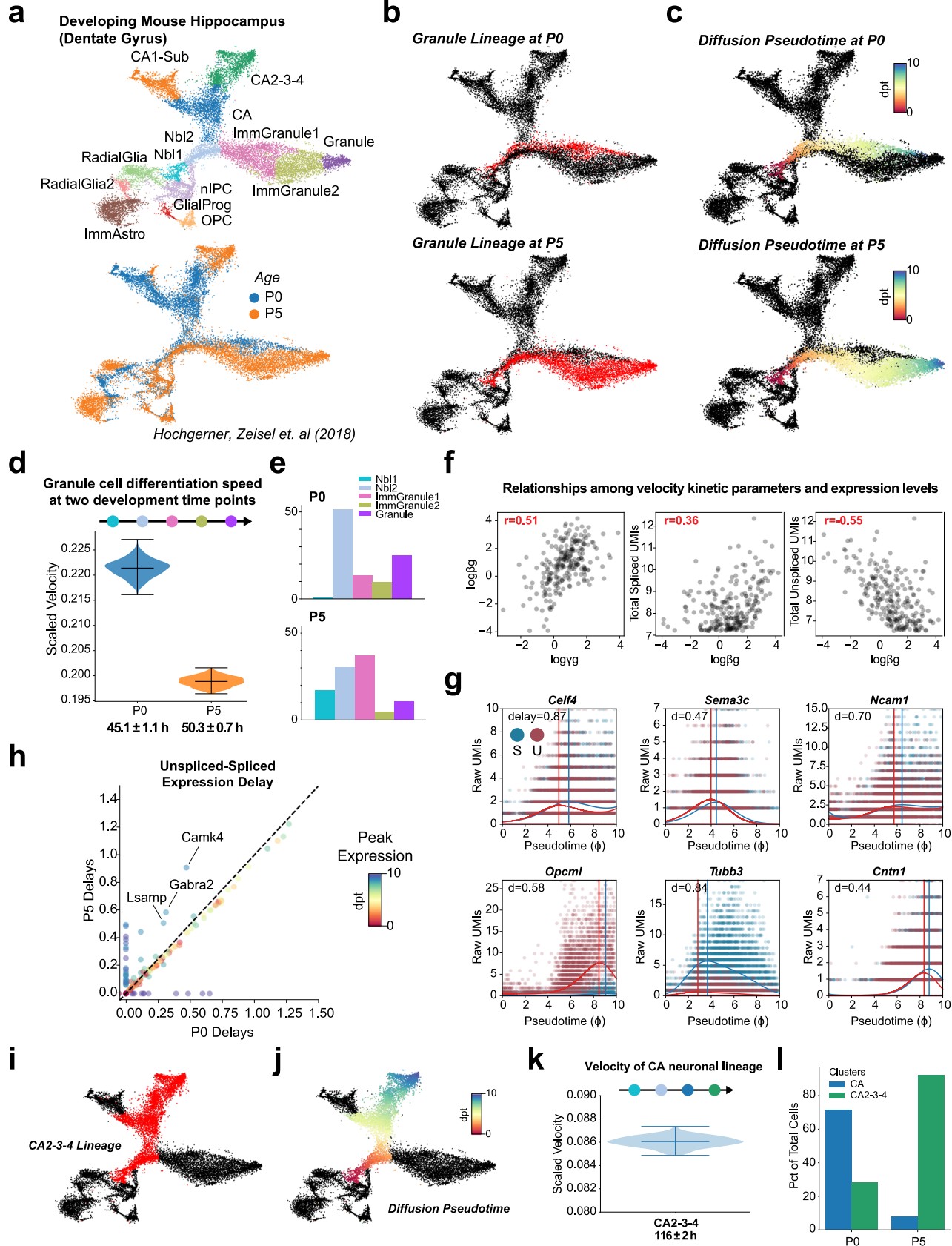

**Extended Data Fig. 9 | Manifold-constrained velocity analysis across cell types and developmental stages in the dentate gyrus. (a)** UMAP of mouse dentate gyrus[64], colored by published cell types (top) and postnatal time point (bottom). **(b)** UMAP in (a), colored by selected data subsets (red) used to estimate 1D interval velocity of the granule cell differentiation lineage (Nbl1, Nbl2, ImmGranule1, ImmGranule2, Granule) at P0 and P5 time points. **(c)** UMAP of data subsets in (b), colored by the diffusion pseudotime applied as the low-dimensional manifold during velocity inference. **(d)** Violin plots of the posterior estimates obtained on the entire granule cell differentiation lineage. **(e)** Bar plots of cell type proportions in the granule lineage at P0 and P5 relative to the total number of cells. **(f)** Scatter-plot of the relationships between kinetic parameters and total un/spliced counts in cells from (d). Pearson's correlation coefficients are indicated in red. **(g)** Scatter-plots of three selected genes, illustrating the estimated expected spliced (blue dashed line; ES) and unspliced (red dashed line;

EU) levels along granule cell differentiation (d) compared to measured spliced (blue; S) and unspliced (red; U) counts. Peak un/spliced delay is indicated by vertical lines and the value at the top left (delay). **(h)** Scatter-plot of mean un/spliced expression delay for 237 shared genes during granule cell differentiation between P0 and P5. Gene dots are colored by manifold time (pseudotime) at peak expression. **(i)** UMAP colored by selected data subsets (red) used to estimate RNA velocity of the CA differentiation lineage (Nbl1, Nbl2, CA, CA2-3-4). **(j)** UMAP of lineage in (i), colored by diffusion pseudotime. **(k)** Violin plot of the posterior estimate obtained on the CA2-3-4 cell differentiation lineage using both P0 and P5 cells. **(l)** Bar plots of cell type proportions in the CA and CA2-3-4 clusters relative to the total number of cells at each time point. 200 posterior samples were used in (d) and (k); the black horizontal lines indicate the 5th, 50th, and 95th percentiles. An estimate of total differentiation process duration is indicated in black at the bottom of the x-axis.

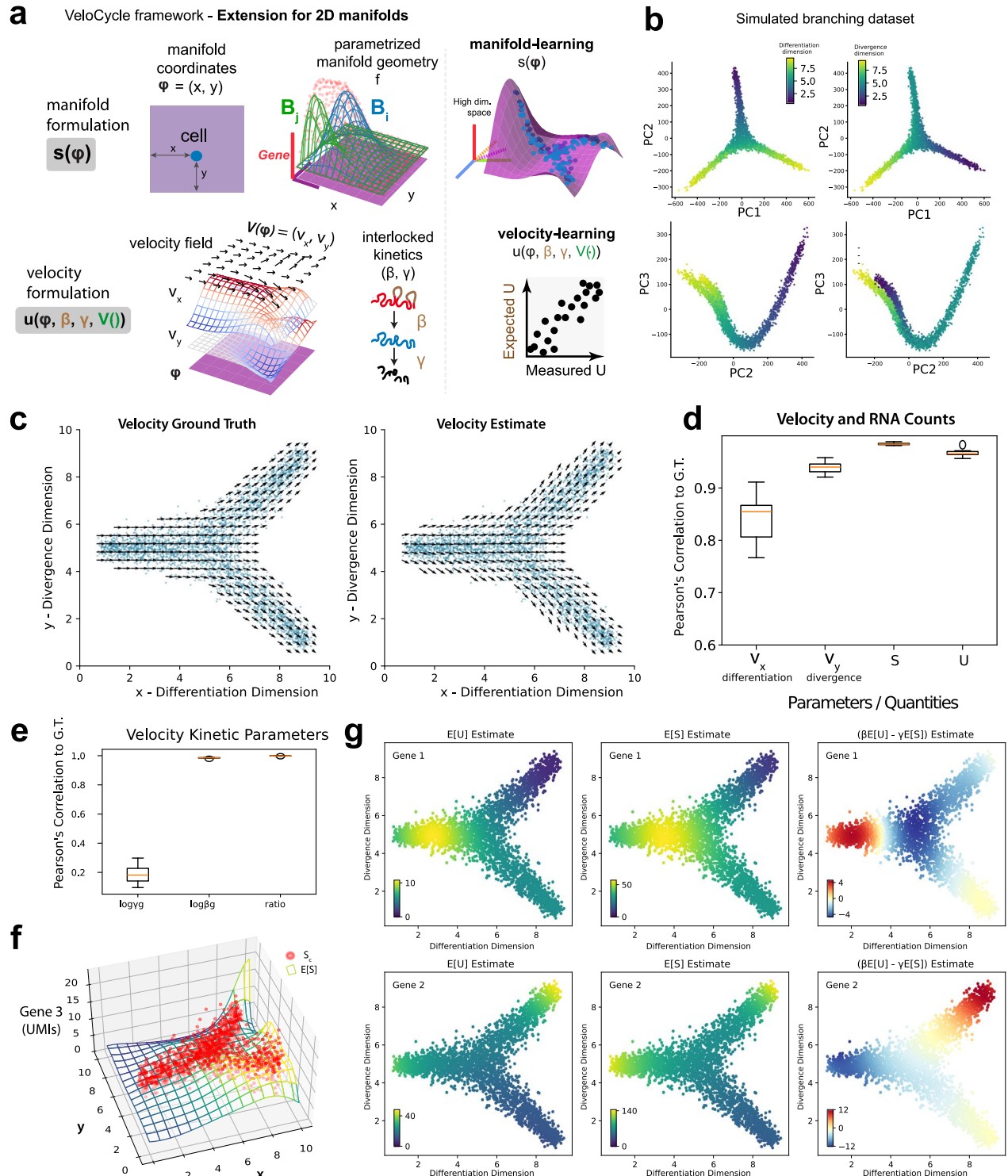

**Extended Data Fig. 10 | Formulation of manifold-constrained velocity analysis along a 2D axis. (a)** Schematic of the 2D manifold model, where variation in gene expression along two dimensions (defined, for example, by two principal components) is estimated using B-splines instead of a Fourier series (as in VeloCycle). **(b)** Simulated 2D branching dataset with estimated and ground truth (GT) velocity. Cell positions from one of the ten simulated datasets are shown in blue. Velocity and spliced mRNA counts were parameterized as 2D B-splines conditioned on the coordinates of the cells; GT velocity spline coefficients were set manually. **(c)** Box plots of Pearson's correlation coefficients between the estimated and GT velocities and mean un/spliced counts. Velocities were evaluated separately along two dimensions, one corresponding to the overall differentiation process and one representing the divergence of the branches. **(d)** Box plots of Pearson's correlation coefficients between the estimated and GT kinetic parameters across ten simulated datasets. Boundaries in (d) and (e) are defined by the interquartile range (IQR); whiskers extend each box by 1.5x the IQR. **(e)** Estimated expected un/spliced counts of two selected genes and the estimated spliced counts derivative. **(f)** Scatter and surface plot representing an example of a gene fit as a function of manifold location using splines. Red dots are simulated data and the mesh surface is the expectation that was fit by the manifold learning step. **(g)** Scatter-plots of representative genes colored by the expected S and U obtained by manifold learning and velocity learning steps. Plots on the right make the un/spliced delay easier to appreciate by coloring the scatter by a proxy for gene-wise velocity βE[U] - γE[S].

| | |
|---|---|

# Reporting Summary

## Statistics

For all statistical analyses, confirm that the following items are present in the figure legend, table legend, main text, or Methods section.

| n/a | Confirmed | |
|---|---|---|
| ☐ | ☒ | The exact sample size (*n*) for each experimental group/condition, given as a discrete number and unit of measurement |
| ☐ | ☒ | A statement on whether measurements were taken from distinct samples or whether the same sample was measured repeatedly |
| ☐ | ☒ | The statistical test(s) used AND whether they are one- or two-sided<br>*Only common tests should be described solely by name; describe more complex techniques in the Methods section.* |
| ☒ | ☐ | A description of all covariates tested |
| ☒ | ☐ | A description of any assumptions or corrections, such as tests of normality and adjustment for multiple comparisons |
| ☐ | ☒ | A full description of the statistical parameters including central tendency (e.g. means) or other basic estimates (e.g. regression coefficient) AND variation (e.g. standard deviation) or associated estimates of uncertainty (e.g. confidence intervals) |
| ☐ | ☒ | For null hypothesis testing, the test statistic (e.g. $F$, $t$, $r$) with confidence intervals, effect sizes, degrees of freedom and $P$ value noted<br>*Give P values as exact values whenever suitable.* |
| ☐ | ☒ | For Bayesian analysis, information on the choice of priors and Markov chain Monte Carlo settings |
| ☒ | ☐ | For hierarchical and complex designs, identification of the appropriate level for tests and full reporting of outcomes |
| ☐ | ☒ | Estimates of effect sizes (e.g. Cohen's *d*, Pearson's *r*), indicating how they were calculated |

*Our web collection on statistics for biologists contains articles on many of the points above.*

## Software and code

Policy information about availability of computer code

| | |
|---|---|
| Data collection | VeloCycle is implemented in Python and available as an open-source package on GitHub at https://github.com/lamanno-lab/velocycle. VeloCycle can be installed from PyPi using the command pip install velocycle or via direct installation from the GitHub page using the command pip install git+https://github.com/lamanno-epfl/velocycle.git@main. Source code, installation instructions, tutorials, and a requirements.txt file containing all necessary package version dependencies are also available on GitHub. Tutorials can be found at the link: https://github.com/lamanno-epfl/velocycle/tree/main/tutorials |
| Data analysis | Analyses performed in this study were completed using VeloCycle v0.1.0.5. Additional code and notebooks to reproduce the results of this study are available on Zenodo with the following registered DOI: 10.5281/zenodo.12517650.<br><br>Python version 3.8 or newer is required. Analyses were performed using the following version of major packages: numpy (v1.24.4), pandas (v2.0.3), scanpy (v1.9.6), matplotlib (v3.7.4), pyro-ppl (v1.8.6), pyro-api (v.0.1.2), scikit-learn (v1.3.2), torch (v2.0.1), and pycircstat (v0.0.2).<br><br>To process raw fastq files into spliced and unspliced count matrices, we either used processed files available from previous studies or ran CellRanger (v6.0.2) and velocyto (v0.17.17). For phase estimation with DeepCycle, we used the latest version of the software, installed via GitHub from commit a33701a. |

For manuscripts utilizing custom algorithms or software that are central to the research but not yet described in published literature, software must be made available to editors and reviewers. We strongly encourage code deposition in a community repository (e.g. GitHub). See the Nature Portfolio guidelines for submitting code & software for further information.

# Data

Policy information about availability of data

All manuscripts must include a data availability statement. This statement should provide the following information, where applicable:

- Accession codes, unique identifiers, or web links for publicly available datasets
- A description of any restrictions on data availability
- For clinical datasets or third party data, please ensure that the statement adheres to our policy

The raw and processed scRNA-seq data in the RPE1 cell line that was newly generated for this study are available at GEO accession number GSE250148. All other scRNA-seq data used in this study were collected from previously published works (see publication for references) and relied on the cell type annotations made by the original authors.

Jupyter notebooks and other affiliated files to reproduce the results shown in this study are provided on Zenodo at the following DOI: 10.5281/zenodo.12517650. Processed versions of all published data (including spliced-unspliced counts matrices), simulated scRNA-seq datasets, processed scRNA-seq metadata for the new RPE1 samples, cell tracking data from live-image microscopy and cumulative EdU staining experiments are also on Zenodo.

# Human research participants

Policy information about studies involving human research participants and Sex and Gender in Research.

| | |
|---|---|
| Reporting on sex and gender | n/a |
| Population characteristics | n/a |
| Recruitment | n/a |
| Ethics oversight | n/a |

Note that full information on the approval of the study protocol must also be provided in the manuscript.

# Field-specific reporting

Please select the one below that is the best fit for your research. If you are not sure, read the appropriate sections before making your selection.

☒ Life sciences ☐ Behavioural & social sciences ☐ Ecological, evolutionary & environmental sciences

For a reference copy of the document with all sections, see nature.com/documents/nr-reporting-summary-flat.pdf

# Life sciences study design

All studies must disclose on these points even when the disclosure is negative.

| | |
|---|---|
| Sample size | No sample size calculation was performed, the study is about development and validation of a method. |
| Data exclusions | For scRNA-seq samples containing a mixture of cell types (i.e., human fibroblasts, developing mouse brain), cycling cells were isolated using the categorical cell type annotations provided by the authors of the studies in which the datasets were originally published. For the Perturb-seq dataset, all cells belonging to a knockdown condition that was represented by a total of at least 75 cells was retained. For all newly-generated scRNA-seq datasets, bad quality cells were filtered based on low UMIs and doubled were filtered based on high UMIs using standard practices of the community; genes were generally filtered based on an average spliced expression > 0.3 and an average unspliced expression > 0.1. More information on dataset specific filtering criteria are available in the Methods section and in the Jupyter notebooks used to perform the analysis, which are available at the Zenodo page. |
| Replication | For cell tracking experiments with RPE1 cells, between 20-25 cells were tracked from 15 different fields of view by three different individuals (A.R.L., A.H., A.V.) for a total of 337 cells used to estimate a ground truth cell cycle period.<br><br>For cumulative EdU and p21 experiments, 3 different experimental replicates (with 10, 10, and 9 time points each) were used to estimate the %p21 positive cells and the overall cell cycle time. |
| Randomization | For most analyses that were dataset-specific, randomization was not relevant to the analyses performed. For cell tracking experiments, three different individuals manually tracked between 20-25 randomly selected cells from 5 randomly chosen (but non-overlapping) fields of view. |
| Blinding | Blinding is not relevant to the experimental study presented in this work. |

# Reporting for specific materials, systems and methods

We require information from authors about some types of materials, experimental systems and methods used in many studies. Here, indicate whether each material, system or method listed is relevant to your study. If you are not sure if a list item applies to your research, read the appropriate section before selecting a response.

## Materials & experimental systems

| n/a | Involved in the study |
|-----|-----------------------|
| ☒ | ☐ Antibodies |
| ☐ | ☒ Eukaryotic cell lines |
| ☒ | ☐ Palaeontology and archaeology |
| ☒ | ☐ Animals and other organisms |
| ☒ | ☐ Clinical data |
| ☒ | ☐ Dual use research of concern |

## Methods

| n/a | Involved in the study |
|-----|-----------------------|
| ☒ | ☐ ChIP-seq |
| ☒ | ☐ Flow cytometry |
| ☒ | ☐ MRI-based neuroimaging |

## Eukaryotic cell lines

Policy information about cell lines and Sex and Gender in Research

| | |
|---|---|
| Cell line source(s) | FUCCI-RPE1 cells (Battich et al. Science 2020) were obtained from the Tanenbaum Lab (Hubrecht Institute) via Oncode. To generate these cells, hTERT RPE-1 cells (ATCC; CRL-4000) were transduced with lentivirus expressing mkO2-hCdh1 (30/120) (FUCCI-G1) and mAG-hGem (1/110) (FUCCI-G2) (Sakaue-Sawano et al., 2008). |
| Authentication | None of the cell lines were authenticated by genome sequencing prior to experiments. However, RPE1 cells were maintained in culture at least for two passages prior to experiments to ensure culture stability. |
| Mycoplasma contamination | RPE1 cells were confirmed to be free of mycoplasma before proceeding with experiments. |
| Commonly misidentified lines (See ICLAC register) | No commonly misidentified cell lines were used in this study. |

