## [Peer Review File · Nature Methods]

Statistical inference with a manifold-constrained RNA velocity model uncovers cell cycle speed modulations

Corresponding Author: Professor Gioele La Manno

Version 0:

Decision Letter:

13th Feb 2024

Dear Gioele,

Your Article, "Statistical inference with a manifold-constrained RNA velocity model uncovers cell cycle speed modulations", has now been seen by 3 reviewers. As you will see from their comments below, although the reviewers find your work of considerable potential interest, they have raised a number of concerns. We are interested in the possibility of publishing your paper in Nature Methods, but would like to consider your response to these concerns before we reach a final decision on publication.

We therefore invite you to revise your manuscript to fully address these concerns. In particular, please make sure to add comprehensive benchmarking, and demonstration of generalizability.

Link Redacted

We hope to receive your revised paper within 12 weeks. If you cannot send it within this time, please let us know. In this event, we will still be happy to reconsider your paper at a later date so long as nothing similar has been accepted for publication at Nature Methods or published elsewhere.

OPEN SCIENCE REQUIREMENTS

REPORTING SUMMARY AND EDITORIAL POLICY CHECKLISTS

Reporting summary: <https://www.nature.com/documents/nr-reporting-summary.zip>
Editorial policy checklist: <https://www.nature.com/documents/nr-editorial-policy-checklist.zip>

DATA AVAILABILITY

All novel DNA and RNA sequencing data, protein sequences, genetic polymorphisms, linked genotype and phenotype data, gene expression data, macromolecular structures, and proteomics data must be deposited in a publicly accessible database, and accession codes and associated hyperlinks must be provided in the "Data Availability" section.

CODE AVAILABILITY

Please include a "Code Availability" subsection in the Online Methods which details how your custom code is made available. Only in rare cases (where code is not central to the main conclusions of the paper) is the statement "available upon request" allowed (and reasons should be specified).

For more information on our code sharing policy and requirements, please see: <https://www.nature.com/nature-research/editorial-policies/reporting-standards#availability-of-computer-code>

MATERIALS AVAILABILITY

SUPPLEMENTARY PROTOCOL

To help facilitate reproducibility and uptake of your method, we ask you to prepare a step-by-step Supplementary Protocol for the method described in this paper. We [encourage authors to share their step-by-step experimental protocols](https://www.nature.com/nature-research/editorial-policies/reporting-standards#protocols) on a protocol sharing platform of their choice and report the protocol DOI in the reference list. Nature Portfolio 's Protocol Exchange is a free-to-use and open resource for protocols; protocols deposited in Protocol Exchange are citable and can be linked from the published article. More details can found at www.nature.com/protocolexchange/about.

ORCID

Sincerely,
Madhura

Madhura Mukhopadhyay, PhD
Senior Editor
Nature Methods

Reviewers' Comments:

Reviewer #1:

Remarks to the Author:

The authors developed a Bayesian model VeloCycle to study cell cycle dynamics. Based on the assumption that cell cycle dynamics can be described on a one-dimensional periodic manifold, VeloCycle can infer RNA velocity for the cell cycle. Remarkably, VeloCycle can estimate cell-cycle periods on a real-time scale, verified by experiments. The source code is well-written. The estimation provided in the Jupyter notebooks can reproduce the respective results. The major concern is how the proposed manifold-constrained framework can be extended to other biological processes rather than the cell cycle. Or is this framework specifically designed to solve the cell cycle estimation?

1. In the introduction, the authors claimed "However, a seldom discussed, yet central, limitation of most RNA velocity models is that velocity estimation is not performed jointly on all genes". However, several algorithms such as VeloVAE, UniTVelo, and DeepVelo have joined all genes using auto-encoder methods and constructed the low-dimension space.

2. One key component of the work is to learn a good manifold x which is a function of time t . The manifold for the cell cycle is intuitive and straightforward. However, it is still elusive to me how to perform manifold learning for dynamics other than cell cycle e.g. developmental process.

3. Another following question is the Extended Data Fig. 4 in the manuscript of 'RNA velocity of single cells' (Nature, 2018) showed different PCs can reflect different biological processes. Can manifold learning infer different low-dimensional spaces related to a specific biological function?

4. I'm wondering whether VeloCycle could be applied to scRNA-seq datasets comprising of heterogeneous cell types, some of which are not cycling (e.g. cells in the resting stage). Moreover, one of the key changes in the RNA velocity field is its capability to infer velocity in heterogeneity cell populations which may have the transcriptional boost or branching events. Authors should also evaluate and discuss their models by comparing their tools with cellDancer and VeloVAE.

5. One important application of RNA velocity is to study the development and differentiation. To take into account the heterogeneity of cell dynamics, RNA frameworks such as CellDancer use deep learning to estimate RNA velocity through cell-dependent rates. The authors briefly discussed cell dynamics which can be described by multiple manifolds with varying topologies, spanned by cells in different subspaces. Could the authors discuss the viability of extending VeloCycle to study those dynamics?

6. Line 225-227, "Specifically, we incorporated positive correlations among the splicing and degradation rates ($r=0.30$) and baseline expression levels ($r=0.30$)". What are baseline expression levels? It implies to be v_0 as the zero-order coefficient in the Fourier series. What is the biological meaning?

The method part needs to be improved. I have a few comments as follows.

1. Use of notations appears to be quite arbitrary and is at times confusing. For example, in Eqn. 5, $v(x)$ should be $V(x)$. In Eqn. 6, a partial derivative to ϕ is introduced while s has only one explicit variable ϕ . Eqn. 9 are actually two equations and need to be split to avoid confusion. In Eqn. 10, partial derivative to t , etc. I suggest the authors carefully check the consistency and accuracy of the notations thoroughly.
2. Above Eqn. 8. The statement "Moreover, since s is positive, we will use the notation $\log s$ " is confusing as well. In various figures (such Fig 2), the authors have shown that s fit nicely as harmonic functions of ϕ ; however, here there is no evidence that $\log(s)$ behaves in the same fashion, especially in the case the authors consider up to the first order.
3. (critical) Eqn. (12) is wrong. ω should not multiple γ . Since the same wrong equation also appears in Figure 1, I'm concerned whether this is a typo, or the implementation is wrong.

Reviewer #2:

Remarks to the Author:

In this very nicely drafted manuscript, Alex, et. al. proposed, VeloCycle, a generative model for RNA velocity and a Bayesian inference approach aimed at addressing the fragility and heuristic reliance of existing RNA velocity algorithms. The results from VeloCycle are thoroughly validated on both the simulation, in vivo dataset of cell cycle and even the live imaging imaging and EdU labeling experiments. This study also reveals interesting regionally-defined proliferation patterns in neural progenitors and how genetic perturbations affect cell cycle speed in a stage dependent fashion. In the following, I will list a few key comments that should be addressed to improve the presentation of this study.

About the manifold-constrained RNA velocity. The idea of constraining the RNA velocity on the latent manifold of the single cell gene expression to avoid RNA velocity rapidly escaping the cell manifold is interesting. In this study the authors explicitly deal with a specific type of manifold – a circular manifold of the cell cycle progress and leverages Bayesian framework and variational inference based method to estimate the kinetic parameters and RNA velocity vectors (Eq. 4 in their paper). However, an alternative and seemingly more flexible approach is to first learn RNA velocity using existing methods, including velocity, scVelo, other deep learning based methods, such as LatentVelo, UniVelo, DeepVelo, VeloVI, etc., but then project the velocity to the tangent space of the underlying manifold. This idea was previously attempted in the graph-dynamo paper (<https://www.biorxiv.org/content/10.1101/2023.09.24.559170v1>). In this study, the manifold is defined by the latent dimension that can be returned from PCA, autoencoders or other non-linear dimension reduction methods. The complex geometry of the manifold is then approximated by the nearest neighbor cells. The new tangent space projection (TSP) method demonstrates it can maintain the RNA velocity magnitude and thus time scale better than previous cosine correlation based kernels in both simulation and real datasets. It will be interesting to discuss graph-dynamo in this work to explain the rationale behind the VeloCycle approach. It will be also interesting to see some extensive benchmarks between graph-dynamo and the VeloCycle method.

In this study, the author focuses on their own cell cycle dataset and claims that it is the first example of direct validation of RNA velocity estimation in units of real time. There are in fact several other cell cycle datasets with time resolved information based RNA metabolic labeling, including the scEU-seq (<https://www.science.org/doi/10.1126/science.aax3072>) and the sci-fate (<https://www.nature.com/articles/s41587-020-0480-9>). The time resolved scRNA-seq will naturally offer RNA velocity with real units (molecules per hour). It will be interesting to see how VeloCycle performs on these two datasets in comparison to the Dynamo approach (<https://www.sciencedirect.com/science/article/pii/S0092867421015774>) that explicitly leverages the metabolic labeling information.

In Eq. 5 of their paper, the author defines the integral to calculate the time needed between two states S_0, S_1 . Since there are an infinite number of paths connecting two states in the state space, how do the authors define the path taken for the integral? I wonder whether the least action path (LAP) principle proposed in the Dynamo paper can be used here? In fact, the LAP method can be easily used to calculate the cell cycle time by specifying the starting point and end point on the cell cycle manifold, followed by running the LAP prediction. It will be interesting to compare the estimates of cell cycle time from VeloCycle with those from Dynamo.

The paper only demonstrates a single case of cell cycle, another simple manifold is a linear trajectory. Can we also include an analysis for such a manifold structure? I suggest running analyses on either the sci-fate dataset (which include both the cell cycle process and a linear response after DEX treatment) and the scNT-seq dataset (the neural polarization experiment, <https://www.nature.com/articles/s41592-020-0935-4>). Extensive benchmarks with Dynamo and other tools should also be performed on these datasets.

Minor comments:

Fig 1. This figure includes several equations that should be better explained in the figure caption. For example, in panel d, you may want to explain how to calculate the RNA velocity by assuming it is defined as the inner product (or the projection) of the velocity $V(x(t))$ to the tangent space via the gradient of S_g on the latent space x . In addition, this section discusses one unique feature of VeloCycle is to perform statistical velocity significance test. This can be an important point and it will be interesting to see more application of this method in performing the significant test across different samples or batches for meaningful biological discovery related to RNA velocity shifts.

Fig 2. Minor comments on the figure caption. In panel I, the bottom plot is not the boxplot instead it is the point plot with the standard deviation.

Fig 3. It will be interesting to see the quiver plot and the streamline plot of the RNA velocity vector overlaying the manifold in figure 3a. It may be also worth to see the RNA velocity plot for just two cell cycle genes (a plot where the x, y axis are gene expression of two genes and the quivers are the RNA velocity for this two genes in each single cell)

Fig 4. Minor point. The text "Pearson's correlation is indicated in red" should be for Panel C but it is currently written for panel b.

Fig 7. The author uses HybISS approach and the BoneFight algorithm to reveal the region specific cell cycle pattern along the forebrain-midbrain-hindbrain axis. It may be interesting to directly leverage the single cell data from Stereo-seq

(cell.com/cell/pdf/S0092-8674(22)00399-3.pdf) to perform such analyses because it gives the real spatial information and at the same has the intron and exon information. The slices used for the Cover figure in that issue may be a good idea to try. Method sections: more explanation and clarification are needed for several equations. For example, $M: x \mapsto s(x) \in M$, it is confusing to me to understand why latent space coordinate x is mapped to the manifold through the function s . What is the function s ? Is this some function for the spliced RNA?

Reviewer #3:

Remarks to the Author:

Lederer et al. developed a framework, VeloCycle, involving a generative model of RNA velocity and Bayesian inference, to estimate RNA velocity in a statistically-robust way that is consistent with the gene expression manifold which cells traverse. Specifically, they propose to parameterize RNA velocity as a field defined on the corresponding gene expression manifold. This is done, in turn by simultaneous estimation of RNA velocity inference and manifold inference.

VeloCycle specifically is focused on the inference of RNA velocity along the cell cycle. In principle, it could be extended to additional topologies, but the parameterization could be a challenge. The authors applied VeloCycle to both in vitro perturb-seq data and in vivo samples, and uncovered proliferation modes and changes in cell cycle speed across samples, conditions and space in different samples.

Indeed, estimation of RNA velocity based on scRNA-seq data is known to suffer from limitations related to inherent noise and lack of robustness, which could be mitigated by the statistically robust method suggested in this paper.

The authors first validated VeloCycle on simulated data, showing that they can reconstruct both the manifold and velocity on top of it - the right phases of circular processes, rate parameters and fourier series coefficients. Next they showed good reconstruction of cell cycle phases for mESC smart-seq2 dataset (for which they also suggest new cell cycle markers), and human fibroblasts 10X chromium data.

Using estimates of cell cycle speed from live-microscopy imaging, the authors were able to, for example, estimate real-time progression, and reason about changing speed along the cycle.

The authors also statistically compared, and found differences in cell cycle progression and characteristics of a adenocarcinoma cancer cell line before and after drug treatment. They were additionally able to quantify differences in cell cycle speed across spatial zones of radial glial cells, which stabilized at later developmental stages. Finally, they propose a method for transfer learning that enables to infer velocity consistent with the cell cycle manifold using small numbers of cells, and demonstrate it in the context of perturb-seq dataset.

The paper is thorough and well written.

The code is at <https://github.com/lamanno-epfl/velocycle>, and not at <https://github.com/lamanno-lab/velocycle> as is currently noted in the data and code availability section. The code itself is available, organized, documented, and includes multiple tutorials.

It would be useful to give more intuition in the main text about the underlying assumptions and mathematical application of transfer learning in the context of VeloCycle.

The authors take the learned gene expression manifold as a natural constraint for velocity estimation. It would be interesting to discuss additional types of potential constraints, in the specific context of the cell cycle or potentially broader than that, that could aid in the reliable inference of RNA velocity.

It would be useful to add visualized comparisons of low-d velocity estimates by VeloCycle relative to baselines which do not constrain velocity to lie on the low-d gene expression manifold.

Should we expect VeloCycle to work better/worse on particular types of data/experiments?

Author Rebuttal letter:

Point-by-Point Response Letter

Date: 17th May 2024

Manuscript Number: NMETH-A54818

Title of Article: Statistical inference with a manifold-constrained RNA velocity model uncovers cell cycle speed modulations

Name of the Corresponding Authors: Gioele La Manno & Felix Naef

Email Address of the Corresponding Author: gioele.lamanno@epfl.ch, felix.naef@epfl.ch

We are grateful for the positive comments and pleased that all reviewers recognized the value of our manifold-constrained framework for RNA velocity estimation and inference, from which VeloCycle is derived. We are also thankful for the appreciation of how rigorously we validated our method using a combination of simulated and real data as well as experimental validations. The reviewers have offered numerous points of constructive feedback. In response, our revised submission incorporates new results, models, and comparisons that we

think greatly strengthen the overall manuscript.

We have diligently and to the best of our ability sought to address all comments. For comments that recurred multiple times across all reviewers, we typically went beyond what the reviewers requested. Specifically, we identified three main recurring motifs in the feedback:

1. There was a consensus that clarifying or demonstrating how our manifold-constrained framework can be extended beyond VeloCycle, which was specifically designed for periodic manifolds, would be beneficial to showcase our advances and encourage further manifold-constrained extensions from the community.
2. We were encouraged to thoroughly benchmark the outputs of our framework that have analogues in other RNA velocity methods.
3. Clarification and tests were requested on the performance differences between our models and those based on metabolically-labeled single-cell data.

In response to these comments and the detailed feedback, the update to our manuscript includes 9 new figures with a total of 56 new display items, a further metabolic labeling-based validation, and additional models, code, and data on Github and the affiliated Google Drive. The key additions are summarized as follows:

- We demonstrate that our manifold-constrained RNA velocity framework can be adapted to formulate other models beyond the 1D periodic case. Specifically, we formulate, implement, and perform key tests on two new models: a 1D interval (non-periodic) model designed to study differentiation speed, and a two-dimensional model suited for examining more complex settings. The models are fit and showcased on simulated data and real datasets. We believe that we managed to incorporate these additional models into the manuscript without diverging from our primary

1

narrative and focus on the cell cycle, which remains the most finely statistically refined method in our study, and the one carefully validated experimentally.

- We have now introduced careful and extensive benchmarking of all features and outputs of VeloCycle that are also present in other RNA velocity methods. We acknowledge that this aspect was underdeveloped in our previous manuscript version; our intention was not to diminish the value of these established high-quality methods in the literature, but to highlight the unique attributes of our model: specifically, the direct estimation of low-dimensional velocity as an explicit function of a manifold and the use of the joint posterior to perform inference. These novel contributions of unique features cannot yet be benchmarked, but for now just validated, which is what we had focused on.

- Furthermore, we have clarified, through detailed experiments and discussion, how our model compares with those based on metabolic labeling. It was never our intention to suggest that unspliced/spliced RNA velocity could outperform methods that utilize metabolically labeled data, which have access to superior targeted data, including typically at least four modalities worth of data (labeled and unlabeled, spliced and unspliced), across up to six pulse intervals. These targeted methods, due to their data richness, are unsurpassable by approaches using untargeted snapshots. However, metabolic labeling methods are considerably more labor-intensive, less standard, costly, and limited in application to in vitro cell cultures. Thus, our demonstration that our velocity estimation outputs are comparable, despite having access to much less data, to these methods is a significant achievement.

With the revisions implemented, we believe our work has significantly improved and we are grateful to the reviewers for their truly constructive feedback. We hope that the reviewers will recognize the value of how we have integrated the new display items into our manuscript. These additions have been made without compromising the original narrative and focus, which, according to the reviews, were already compelling.

2

Reviewer #1:

The authors developed a Bayesian model VeloCycle to study cell cycle dynamics. Based on the assumption that cell cycle dynamics can be described on a one-dimensional periodic manifold, VeloCycle can infer RNA velocity for the cell cycle. Remarkably, VeloCycle can

estimate cell-cycle periods on a real-time scale, verified by experiments. The source code is well-written. The estimation provided in the Jupyter notebooks can reproduce the respective results.

We thank the reviewer for their positive comments on our manuscript and for recognition of the experimental work we performed to validate VeloCycle estimates for cell cycle periods on a real-time scale. We are happy to hear the reviewer found the code and notebooks to be well-organized and reproducible.

The major concern is how the proposed manifold-constrained framework can be extended to other biological processes besides the cell cycle. Or is this framework specifically designed to solve the cell cycle estimation problem?

VeloCycle is a method that emerges from a new manifold-constrained theoretical framework we propose here for the first time. We chose the cell cycle as the first formulation to explore both for the biological importance of the process and because it offers a unique opportunity to conduct multiple orthogonal validations. Other interesting model formulations derived from this framework for different applications are possible, and we now showcase this in the revised manuscript (Fig. S8-S11). Indeed, we anticipate the community will rapidly follow up with more application-specific derivations from this manifold-constrained framework in the future.

Despite these very promising results, we anticipate that a dedicated effort will be required in order to elevate other formulations to stable and validated methods of proven use for the community. This effort will involve careful model selection, study of the specific model's behavior, sensitivity analyses, benchmarking, and an experimental validation tour de force as extensive as the one we present in our current manuscript for VeloCycle. We think the following sentences in the revised text clearly communicate both the opportunities and non-triviality of this process:

In the results section:

“In the rest of this work, we study and characterize VeloCycle, a method derived from this manifold-constrained framework for one-dimensional periodic manifolds. However, the framework can also be leveraged to generate formulations of higher dimensionality and various geometries, including a one-dimensional interval (Fig. S8, S9, and S10; see Methods 4.7, 8.6, 8.7) and two-dimensional case (Fig. S11; see Methods 4.7) (Hochgerner et al. 2018; Bastidas-Ponce et al. 2019).”

In the discussion section:

3

“While our model for RNA velocity estimation offers clear benefits, there remain open avenues for further development. VeloCycle focuses on the case of one-dimensional periodic manifolds, yet extensions to latent spaces of different dimensionalities and topologies can be naturally pursued. Here, we showcased a formulation for processes unfolding on one-dimensional intervals, with similar timescale prediction and inference capabilities as VeloCycle on simulated and real datasets (Fig. S8, S9, and S10; see Methods 4.7); we also demonstrate applicability to a two-dimensional case (Fig. S11; see Methods 4.7). However, further experimental validation and characterization of these models, as we show for VeloCycle, will be needed.”

In the revised manuscript and methods, we mathematically describe, implement, and test two new formulations from the manifold-consistent framework targeting different applications. First, a 1D interval (i.e., a 1D non-periodic manifold) can be used to study simple cell differentiation. This requires using a different basis function, namely B-spline bases instead of Fourier harmonics, to model the variation of genes over the manifold. First, we test this method on simulated data (Fig. S8, see Methods 4.7), confirming the model correctly identifies the parameters and recovers both the global velocity and simulated gene-wise parameters.

After evaluating the 1D interval model on simulated data, we moved to studying two real datasets widely used by the RNA velocity community: the pancreatic endocrinogenesis dataset (Fig. S9, see Methods 8.6) and the dentate gyrus postnatal development dataset (Fig. S10, see Methods 8.7). For both, we show the capability to identify plausible velocity time scales compatible with the biological process. We also showcase how the 1D interval (i.e., non-periodic) and periodic analysis can be combined to study co-occurring processes within the same dataset (Fig. S9b, d, and g).

Importantly, we go beyond merely applying the 1D interval model to infer a single trajectory (Fig. S9g and S10j), and we are able to perform statistical inference to: (1) compare the speed at two developmental timepoints (Fig. S10d) and (2) compare the speed of a differentiation trajectory to a real time scale (Fig. S10j and k). These capabilities extend beyond what is currently possible with other velocity methods.

Finally, we showcase a model formulated around a two-dimensional bounded manifold that can be used to study multiple processes that affect each other, or alternatively complex trajectories which are not easily tackled by a divide-and-conquer approach to separate them into linear segments. (see Methods 4.7). Despite this model being more complex and with more degrees of freedom than the others, we obtained encouraging results on simulated data (Fig. S11).

Overall these completely new models and results show the feasibility of deriving more complex models from our statistical, manifold-constrained RNA velocity framework.

4

Figure S8. Evaluation of a 1D interval model of manifold-constrained RNA velocity using simulated data. (a) Schematic of the 1D interval manifold model, where variation in gene expression along the manifold is estimated using B-splines instead of a Fourier series (as in VeloCycle). (b) PCA plot of the two first principal components colored by the manifold coordinate used (time). (c) Box plots of the log mean squared error (MSE) for the expected spliced counts (ES) compared to the simulated raw spliced data (S). MSE was calculated using either the simulated ground truth (G.T.) or the estimates recovered by the non-periodic manifold-constrained model (Estimated). (d) Box plots of the Pearson's correlation coefficient between estimated B-spline coefficients and simulated ground truth. (e) Example scatter plots of spliced gene expression fits along the one-dimensional interval manifold. The solid black line indicates the ground truth and the red dashed line indicates the estimate obtained during manifold-learning. (f) Box plot of the percent error for the velocity estimate compared to a ground truth of 0.25. (g) Scatter plots of example spliced (blue) and unspliced (red) gene expression fits obtained by the model. (h) Box plot of Pearson's correlation coefficients between estimated and ground truth values for the velocity kinetic parameters. All analyses were performed across ten simulated datasets, each containing 3,000 cells and 300 genes (see Fig. 2).

5

Figure S9. Modular and manifold-constrained velocity analysis of the cell cycle and beta cell differentiation during mouse pancreatic endocrinogenesis. (a) UMAP of pancreas dataset (3,696 cells; Bastidas-Ponce et. al 2019) from mouse E15.5, colored by published cell types. (b) UMAP of dataset in (a), colored by the selected cell subsets (in red) extracted to estimate RNA velocity of the cell cycle (left; Ductal; 916 cells) and beta cell differentiation (right; Ngn3 high EP, Pre-endocrine, Beta; 1,825 cells). (c) Low-dimensional plots of the gene expression manifolds estimated for the cell cycle

6

(left) and beta cell differentiation (right). The cell cycle manifold was obtained using manifold-learning in VeloCycle, whereas the differentiation manifold was obtained with diffusion pseudotime performed on the first two principal components. (d) Posterior estimate plot of the cell cycle speed from VeloCycle for cycling ductal cells. (e) Violin plots of the posterior distributions for cell cycle velocity from (d), stratified by categorical phase. (f) Scatter plots of selected genes, illustrating the estimated expected spliced (blue dashed line; ES) and unspliced (red dashed line; EU) along the cell cycle manifold as compared to measured spliced (blue; S) and unspliced (red; U). The peak unspliced-spliced delay is indicated by vertical lines and the value at the top left of each plot. (g) Velocity posterior estimate plot obtained for beta cell differentiation using the 1D interval model, extended upon the manifold-constrained framework. (h) Stacked bar plot of the cell type proportions along the differentiation axis in four different scRNA-seq datasets (E12.5, E13.5, E14.5, and E15.5) from the original study. (i) Scatter plots demonstrating the relationship between the velocity kinetic parameters ($\log\gamma_g$, $\log\beta_g$) and total (spliced, unspliced) counts. Pearson's correlation coefficients are indicated in red. (j) Scatter plots of selected genes, illustrating the estimated expected spliced (blue dashed line; ES) and unspliced (red dashed line; EU) along the cell cycle manifold compared to measured spliced (blue; S) and unspliced (red; U). In panel (d) and (g), the white line indicates the mean over 200 posterior samples, and the black line indicates the full posterior interval. The cell cycle period (d) and the beta cell differentiation process time (g) is indicated at the top left of the respective plots.

7

Figure S10. Manifold-constrained velocity analysis across cell types and developmental stages in the mouse dentate gyrus. (a) UMAP of mouse dentate gyrus dataset (18,213 cells; Hochberger, Zeisel, et. al 2018), colored by published cell types (top) and postnatal time point (bottom). (b) UMAP

8

of dentate gyrus dataset, colored by the selected data subsets (in red) extracted to estimate RNA velocity of the granule cell differentiation lineage (Nbl1, Nbl2, ImmGranule1, ImmGranule2, Granule) at the P0 (left; 1,876 cells) and P5 (right; 4,968 cells) time points. (c) UMAP of data subsets in (b), colored by the diffusion pseudotime applied as the low-dimensional manifold during velocity inference. (d) Violin plots of the posterior estimates obtained on the entire granule cell differentiation lineage at the two time points. (e) Bar plots of cell type proportions in the granule lineage at P0 (top) and P5 (bottom) relative to the total number of cells at each time point. (f) Scatter plot of the relationships between the velocity kinetic parameters, total spliced, and total unspliced counts in the data from (d). Pearson's correlation coefficients are indicated in red. (g) Scatter plots of four selected genes, illustrating the estimated expected spliced (blue dashed line; ES) and unspliced (red dashed line; EU) along granule cell differentiation (d) compared to measured spliced (blue; S) and unspliced (red; U). The peak unspliced-spliced delay is indicated by the vertical lines and the value at the top left of each plot. (h) Scatter plot of the mean unspliced-spliced expression delay for 237 shared genes during granule cell differentiation between P0 and P5 time points. Gene dots are colored by peak expression phase. (i) UMAP colored by the selected data subsets (in red) to estimate RNA velocity of the CA differentiation lineage (Nbl1, Nbl2, CA, CA2-3-4). (j) UMAP of lineage in (i), colored by the diffusion pseudotime applied as the low-dimensional manifold during velocity inference. (k) Violin plot of the posterior estimate obtained on the CA2-3-4 cell differentiation lineage using both P0 and P5 cells. (l) Bar plots of cell type proportions in the CA and CA2-3-4 clusters at P0 and P5 relative to the total number of cells at each time point. 200 posterior samples were generated in (d) and (k), and the estimation of the total differentiation process duration is indicated in black at the bottom of the x-axis.

9

Figure S11. Formulation of manifold-constrained velocity analysis along a 2D manifold. (a) Schematic of the 2D manifold model, where variation in gene expression along two dimensions (defined, for example, by two principal components) is estimated using B-splines instead of a Fourier series (as in VeloCycle). (b) Simulated 2D branching dataset with ground truth and estimated velocity. Cell positions from one of the ten simulated datasets are shown in blue. Velocity and spliced mRNA counts were parameterized as 2D B-splines conditioned on the coordinates of the cells; ground truth velocity spline coefficients were set manually. (c) Box plots of Pearson's correlation coefficients

10

between the estimated and ground truth velocities and mean spliced or unspliced counts. Velocities were evaluated separately along two dimensions, one corresponding to the overall differentiation process and one representing the divergence of the branches. (d) Box plots of Pearson's correlation coefficients between the estimated and ground truth velocity kinetic parameters across ten simulated datasets. (e) Estimated expected spliced and unspliced counts of two selected genes and the estimated spliced counts derivative.

1. In the introduction, the authors claimed "However, a seldom discussed, yet central, limitation of most RNA velocity models is that velocity estimation is not performed jointly on all genes". However, several algorithms such as VeloVAE, UniTVelo, and DeepVelo have joined all genes using auto-encoder methods and constructed the low-dimension space.

We thank the reviewer for pointing out our inaccurate phrasing. We are aware of the powerful methods mentioned and that they fit all genes simultaneously. Indeed, in an attempt at being brief, we oversimplified. Now, the message should be clearer from the following paragraph in the revised manuscript:

"However, a seldom discussed limitation of some of the earliest and still commonly used RNA velocity models (La Manno et al. 2018; Bergen et al. 2020) is their reliance on the gene-wise fit kinetic parameters and velocities. In this setting, even when global reconciliation is sought post hoc, the estimated kinetic parameters remain independent; this leads to a physically and geometrically inconsistent velocity vector, whose gene-specific

components are on different timescales and whose resulting direction is not necessarily tangent to the low dimensional manifold cells traverse.

Therefore, it is desirable to perform a joint gene fit to regularize the estimates, a strategy introduced by recent methods (Gu, Blaauw, and Welch 2022; Cui et al. 2024; Liang et al. 2020) where the manifold, non-constant kinetic parameters and velocities are all the output of a nonlinear function (e.g. a neural network) of a shared latent representation. This unstructured interdependence does not fully control the information flow from data to estimates and makes it difficult to understand in which way regularization is applied.”

2. One key component of the work is to learn a good manifold x which is a function of time t . The manifold for the cell cycle is intuitive and straightforward. However, it is still elusive to me how to perform manifold learning for dynamics other than cell cycle e.g. developmental process.

We appreciate the reviewer’s observation and acknowledge we did not explain how manifold learning in dynamics other than the cell cycle might be performed. As explained in detail in response to the reviewer’s general comment above, we have included additional reasoning, examples, and results in the revised manuscript from two new formulations (Fig. S8-S11).

11

Since there are different interpretations of the term “manifold”, we want to clarify that we do not use it in the data-centric sense commonly found in machine learning, which refers to “a nonlinear subspace construction summarizing as much as possible the variability of the data.” Rather, we define a manifold as a “a surface of points existing in the high dimensional space with an explicit parametric mapping to a low dimensional latent space, along which the dynamical process of interest unfolds” (see Methods 1). Therefore, the manifold might not be constructed to summarize all the information in the data, but instead to capture the axis of a particular process along which one aims to perform velocity inference.

To be able to find such a manifold, our framework requires specifying (i) the expected topology of a manifold and (ii) a corresponding set of basis functions that reflect how (smoothly) gene expression can vary along the manifold. Given a coherent set of cells undergoing dynamics of interest, these two specifications are not difficult to formulate. For example, during differentiation processes, a 1D interval (i.e., non-periodic) manifold and a polynomial or spline basis is often adequate (as now implemented in a new formulation detailed in Fig. S8-10 and described in Methods 4.7).

In a setting where two dynamic processes interact in the same cells, more complex manifolds are needed. For an illustration of this scenario, we refer to our recent work by Droin et al. Nature Physics 2019, where the circadian clock accelerates the cell cycle (Droin, Paquet, and Naef 2019), and the underlying manifold has the topology of a torus. A 2D plane topology is also adequate to approximate some branching processes, for example, between multiple cell fates or lineages (Rand et al. 2021). In such a 2D case, basis functions can be chosen as the packet functions obtained from the cartesian product of 1D splines (Fig. S11a and Methods 4.7). Another more parsimonious way to approach branching is to stratify cells into two trajectories and model them as separate 1D manifolds (as we now demonstrate in Fig. S9 with proliferating ductal cells and beta cell differentiation). Velocity across these two paths is more easily biologically interpretable than the results of inference on a vector field.

Finally, we agree with the reviewer that for more complex datasets containing a mixture of many different cell types (e.g., a large atlas), specifying an accurate manifold is very difficult, although some theoretically grounded proposals on how to attempt it have been described (Rand et al. 2021). Importantly we argue that in such a complex setting, performing global inference on velocity will probably be beyond reach, but what is desired is rather inference which is “local” (on related cells) or “contrastive” (on groups of cells). Thus, a divide-and-conquer approach is likely a more technically robust way to proceed for statistical inference on RNA velocity with optimal biological interpretability, as we now demonstrate during pancreatic endocrinogenesis (Fig. S9d and g).

3. Another following question is the Extended Data Fig. 4 in the manuscript of ‘RNA velocity of single cells’ (Nature, 2018) showed different PCs can reflect different biological processes. Can manifold learning infer different low-dimensional spaces related to a specific biological function?

We are particularly excited that the reviewer is interested in this direction. We believe it is a desired capability and it is indeed where we would like to bring the framework in the future. We comment on this aspect in the Discussion twice:

“...These could involve using either unbiased or gene-ontology informed data factorization models to extract and study the speed along these modules (Piran et al. 2024; Mages et al. 2023; Piran and Nitzan 2024; Levitin et al. 2019). This type of gene-modular modeling was successfully demonstrated by the method cell2fate (Aivazidis et al. 2023).

“The promising outcomes of tailoring RNA velocity to single processes advocates for the development of new models that dissect the high-dimensionality of single-cell data into individual biological axes with corresponding and interpretable RNA velocity fields.”

In line with this direction, the revised manuscript includes new evidence that we can consider different low dimensional spaces related to different biological functions. Specifically, we now present a new 1D interval and 2D formulation, and in analysis of Fig. S9, both the cell cycle and differentiation from the same dataset, which is the equivalent of Extended Data 4 from La Manno 2018 mentioned by the reviewer. In this case, the analysis was achieved in a semi-supervised fashion (with access to relevant covariates and informed gene feature selection); however, we believe in the future this decomposition could be part of the model (i.e. considering with latent Bernoulli allocations or probabilistic non-negative factorizations) so to behave in a fully automatic manner.

4. I'm wondering whether VeloCycle could be applied to scRNA-seq datasets comprising of heterogeneous cell types, some of which are not cycling (e.g. cells in the resting stage). Moreover, one of the key changes in the RNA velocity field is its capability to infer velocity in heterogeneity cell populations, which may have the transcriptional boost or branching events.

We thank the reviewer for bringing up this important point. While a full treatment in terms of suitable mixture models is beyond the scope of the present work, we found that the current estimator exhibits some interesting robustness with respect to cell heterogeneity. Indeed, in the paper we already showed that VeloCycle can be applied to a cell pool where a fraction of cells is not cycling (i.e., in G0; see Fig. 6m). Specifically, there is a fraction of cells that are not cycling during the validation experiment shown Figure 6, as confirmed by p21 stainings (Fig. 6n). Remarkably, this setting did not significantly impact velocity and cell cycle period estimation, which aligned with the experimentally measured values.

In the revised manuscript, we add further evidence of this robustness using both simulated and real data. Fig. S12 shows a sensitivity analysis constructed by spiking in different proportions of G0 cells into a pool of cycling cells. Across ten simulated datasets, we find that manifold-learning is stable (Pearson's circular correlation > 0.70) until we add 50% as many G0 cells (i.e., 50 non-cycling cells per 100 cycling cells), and that velocity-learning gets

compromised (percent error from the simulated ground truth $> 25\%$) once we add 50% as many G0 cells.

For the real dataset of human dermal fibroblasts (Capolupo et al. 2022), we spiked-in actual non-cycling cells from the same dataset that were annotated in the original study as belonging to non-proliferative clusters. Here, we show that manifold-learning is stable (Pearson's circular correlation > 0.70) even up to 200% as many G0 cells (i.e., 200 non-cycling cells per 100 cycling cells) and that velocity-learning is only compromised once we add 50% as many G0 cells. Considering one can typically detect and filter out non-cycling cells with modest accuracy, these results can be interpreted as proof or robustness to realistic use cases.

Furthermore, VeloCycle can be used to analyze and perform inference on different cell populations within the same dataset to test the hypothesis that they might harbor different velocities. This is done by considering a covariate of interest, performing inference and comparing the posteriors of the populations. We note this can be achieved not only with experimentally-defined covariates (e.g. age, conditions) but also, in an exploratory fashion by stratifying on an arbitrarily selected gene, enrichment score or principal component. We show an example of this comparative setup of cells of the same organ in Fig. 7D, where their velocities are compared.

We have now updated the text in multiple locations to indicate the results of these new sensitivity analysis on simulated and real data as follows:

“Finally, we assessed the impact of increasing fractions of simulated non-cycling cells (i.e., between 0 and 200 non-cycling cells per 100 cycling cells) on manifold-learning, determining that a circular correlation greater than 0.70 obtained in mixed populations containing up to 50 non-cycling cells per 100 cycling cells (Fig. S12a). Velocity estimates also remained within 25% of the ground truth with up to 50 non-cycling cells per 100 cycling cells (Fig. S12b).”

[section talking about the human fibroblast data] “... and results of sensitivity analyses incorporating different fractions of non-cycling G0 cells aligned with the moderate robustness observed in simulations (Fig. S12c-d).”

14

Figure S12. Sensitivity analysis of manifold-learning and velocity-learning on the introduction of non-cycling cells in simulated data and human fibroblasts. (a) Box plots of the circular correlation coefficient between the estimated phase and simulated ground truth (G.T.) across three simulated datasets with varying proportions of non-cycling cells (from 0 to 100 non-cycling cells per 100 cycling cells). (b) Box plots of scaled velocity posterior estimates compared to the simulated G.T. (red dashed line) across three simulated datasets with varying proportions of non-cycling cells. (c) Box plots of the circular correlation coefficient between the estimated phase with varying proportions of non-cycling cells and the phase without any non-cycling cell contaminants taken from the same dataset of human dermal fibroblasts. Contaminant cells were taken from non-cycling cell types as annotated in the original study (Capolupo et al. 2022). (d) Box plots of scaled velocity posterior estimates compared in human dermal fibroblasts with varying proportions of non-cycling cells. The box plots in (c) and (d) indicate the results across three independently sampled subsets of non-cycling cells.

Authors should also evaluate and discuss their models by comparing their tools with cellDancer and VeloVAE.

15

We recognize that comparing with related published methods is a good practice, and we now have followed the suggestion of the reviewer. However, we would like to explain the reason we did not previously attempt comparisons between VeloCycle and other methods such as cellDancer and VeloVAE: due to the design of VeloCycle, the methods perform significantly different tasks and application cases are thus incomparable in many of their characterizing aspects. Specifically, our method is focused on inferential possibilities and estimations of the true time scales of the biological processes (VeloCycle velocities are provided in units of inverse physical time), achieved through constraints, whereas cellDancer and VeloVAE focus on adapting to different cases with a flexible parametrization, not making explicit attempts to capture the biological time scales.

Nonetheless, we fully understand the constructive intention of the request, and thus we crafted the best comparison we could come up with. To respond to this request and the other reviewers, we perform benchmarking between VeloCycle and four other methods – scvelo (Bergen et al. 2020), cellDancer (Li et al. 2023), VeloVAE (Gu, Blaauw, and Welch 2022), and dynamo (Qiu et al. 2022) – on the following datasets:

1. Simulated data (previously show in our sensitivity analysis in Fig. 2)
2. Cycling human dermal fibroblasts (Fig. 6a and Capolupo et al. 2022)
3. A549 cells newly added from (Cao et al. 2020)
4. Cycling RPE1 cells (generated for this study, Fig. 6h)

Specifically, we scrutinize all shared output between the methods with the following means:

- Cross boundary direction correctness (Fig. S13a-b) established by (Qiao and Huang 2021), which is further evaluated across different manifolds for maximal transparency.
- Velocity consistency (Fig. S13c) established by (Gayoso et al. 2024).
- Mean squared error of the spliced and unspliced expression (Fig. S14a-b)
- Mean squared error of the velocity kinetic parameters (Fig. S14c).
- Comparison of the vector fields on the low-dimensional embedding (Fig. S13d).

VeloCycle compares positively to the other methods, with the cross-boundary direction

correctness and velocity consistency metrics performing better with VeloCycle on most of the datasets. Specifically, VeloCycle achieves the best overall in cross boundary direction correctness, one of the most widely-used metrics in the field, even when the clusters are defined by other method's embeddings. An interesting exception is the MSE for the spliced and unspliced, which is higher in our method and lower for VeloVAE and cellDancer. This is in line with what was already mentioned in the paper: we expect other methods to have a higher degree of freedom and thus increased flexibility (lines 112-116). However, this is accompanied with a higher tendency to overfit as shown by the comparably higher variance in the spliced/unspliced expectation when even ground truth is flat (Fig. S14b).

16

Figure S13. Velocity method benchmarking with cross-boundary direction correctness and velocity consistency. (a) Bar plots of mean cross boundary direction correctness (CBDir; Qiao et al. 2021) computed on four datasets (simulated data, human fibroblasts, A549 cells, and RPE1 cells) and with five different velocity methods (VeloCycle, scvelo, cellDancer, dynamo, and VeloVAE). CBDir correctness requires a ground truth direction between clusters; therefore, the score was computed for clusters obtained on each method's embedding (see Methods 8.3). The black dot represents the

17

CBDir score calculated using that specific velocity method's embedding, whereas the black line indicates the standard deviation across scores computed using all five embeddings and cluster annotations. (b) Heatmaps of the CBDir scores computed pairwise for each method using cluster relationships defined on each different embedding, for all four datasets benchmarked. (c) Box plot of cell-wise velocity consistency score (Gayoso, Weiler et al 2023; see Methods 8.4) computed on the same datasets and methods as in (a-b). (d) Low dimensional embedding plots with grid-wise velocity vector fields computed for on human fibroblasts (top) and A549 cells (bottom). Cells are colored by categorical phase to enable the visual inspection of vector field direction correctness.

Figure S14. Velocity method benchmarking against ground truth values with simulated data. (a) Box plots of the spliced (left) and unspliced (right) gene-wise mean squared error (MSE) obtained between the expected un/spliced counts and the simulated ground truth for five benchmarked velocity methods (VeloCycle, scvelo, cellDancer, dynamo, and VeloVAE). (b) Box plots of the gene-wise standard deviation of expected spliced (left) and unspliced (right) expression estimated by five different velocity methods. (c) Box plots of the gene-wise MSE for the velocity kinetic parameters (γ , β , and γ/β ratio) compared to the simulated ground truth for five velocity methods. Simulated data was generated as presented in Fig. 2 for 3,000 cells and 300 genes.

Overall, while metrics show our method has consistently superior performance, we also do not mean to claim that our method is generally better than the others, some of which are more general in application or possess extra features (e.g. they leverage metabolic labeling). Each method is intended for different purposes, and our stance remains that users should focus on scope and proven robustness and validation. Given this, we have commented in the following way in the text:

In the new results section "VeloCycle achieves a competitive cross boundary direction correctness and consistency scores across multiple datasets":

18

"Despite the conceptual challenge of comparing VeloCycle to methods of different scope and assumptions, we benchmarked the performance of VeloCycle on four independent datasets – simulated data (Fig. 2), dermal human fibroblasts (Fig. 6a), metabolically-labeled A549 cells (Cao et al. 2020), and RPE1 cells (Fig. 6h) against four diverse RNA velocity estimation methods: scvelo (Bergen et al. 2020), cellDancer (Li et al. 2023), dynamo (Qiu et al. 2022), and VeloVAE (Gu, Blaauw, and Welch 2022). VeloCycle achieved noticeably improved cross boundary direction correctness (CBDir; Fig. S13a-b) and velocity consistency scores across multiple datasets as compared to pre-existing methods (Fig. S13c), even when their embeddings were used to generate the ground-truth clusters for evaluation (Fig. S13b). This benchmarking analysis also revealed a higher MSE on the spliced and unspliced fits for VeloCycle, which we expected, as it reflects our choice to prioritize regularization over error minimization (Fig. S14)."

In the Discussion section:

“Regarding comparisons with other methods, which are overall quite favorable towards VeloCycle, such benchmarks should not be overinterpreted. For example, the optimal velocity consistency and cross-boundary direction correctness (CBDir) scores obtained by VeloCycle are expected given our self-consistent formulation (Fig. S13). Analogously, we interpret higher mean squared errors on gene expression fits as the necessary cost to attain inferential capabilities (Fig. S14). Overall, we advocate for selection of methods based on their intended application, proven validity, and underlying goals rather than crude metrics.”

5. One important application of RNA velocity is to study the development and differentiation. To take into account the heterogeneity of cell dynamics, RNA frameworks such as CellDancer use deep learning to estimate RNA velocity through cell-dependent rates. The authors briefly discussed cell dynamics which can be described by multiple manifolds with varying topologies, spanned by cells in different subspaces. Could the authors discuss the viability of extending VeloCycle to study those dynamics?

We thank the reviewer for elaborating more deeply on this aspect. As we described above, the revised manuscript offers two examples of how to derive new methods considering different topologies and functional parametrizations. Both cases we considered are related to the use cases the reviewer hints at here. We show the case of a simple A-B differentiation that can be represented with a 1D non-periodic manifold (Fig. S8, S9, and S10) and the case of a branching represented by a 2D manifold (Fig. S11). As reasoned above, we imagine a scalable divide-and-conquer strategy to tackle complex datasets where different processes can be thought as one-dimensional axes involving a subset of the cells in the dataset. Ways to perform this decomposition automatically is by using unsupervised or semi-supervised factorization (Piran et al. 2024; Mages et al. 2023; Piran and Nitzan 2024; Levitin et al. 2019); an alternate approach is with models from the family of Latent Bernoulli/Dirichlet allocations.

19

As requested by the reviewer, we now comment about these aspects in two parts of the manuscript: at the end of the result section “Manifold-constrained RNA velocity addresses shortcomings of other approaches” (lines 229-237) and in the discussion (lines 989-1002), as previously mentioned in the first general comment for this reviewer.

We also comment later in the discussion (lines 1015-1020):

“[...] frameworks that consider multiple manifolds with varying topologies, spanned by cells in different subspaces, while also assigning specific cells and genes to these features, will notably enhance the general applicability and utility of manifold-consistent RNA velocity estimation. These could involve using either unbiased or gene-ontology informed data factorization models to extract and study the speed along these modules (Piran et al. 2024; Mages et al. 2023; Piran and Nitzan 2024; Levitin et al. 2019). This type of gene-modular modeling was successfully demonstrated by the method cell2fate (Aivazidis et al. 2023)”

6. Line 225-227, “Specifically, we incorporated positive correlations among the splicing and degradation rates ($r=0.30$) and baseline expression levels ($r=0.30$)”. What are baseline expression levels? It implies to be v_0 as the zero-order coefficient in the Fourier series. What is the biological meaning?

The reviewer is correct, v_0 is the zero-order coefficient in the Fourier series, which represents the average expression level of each gene across the cell cycle. We simulate some correlation between splicing and degradation rate and the splicing and baseline expression levels to avoid simulating configurations that are implausible and non-biological. For example, a naive simulation can generate a low splicing rate and a high baseline expression for a gene, leading to an extremely high concentration of unspliced RNA. This is unrealistic, and even if it were biological, in practice we do not observe it in our data.

We now clarified this in the text by stating:

“[...] which helps ensure biological plausibility and avoid unrealistic parameter configurations that diverge from empirical observations.”

The method part needs to be improved. I have a few comments as follows.

1. Use of notations appears to be quite arbitrary and is at times confusing. For example, in

Eqn. 5, $v(x)$ should be $V(x)$. In Eqn. 6, a partial derivative to ϕ is introduced while s has only one explicit variable ϕ . Eqn. 9 are actually two equations and need to be split to avoid confusion. In Eqn. 10, partial derivative to t , etc. I suggest the authors carefully check the consistency and accuracy of the notations thoroughly.

We thank the reviewer for this comment and for spotting a few typos. While the use of partial derivatives as we have done is not uncommon in certain areas of mathematics and physics,

20

we have now streamlined the notation as suggested and avoided use of partial derivative notation (which is not wrong) for univariate functions.

2. Above Eqn. 8. The statement “Moreover, since s is positive, we will use the notation $\log s$ ” is confusing as well. In various figures (such Fig 2), the authors have shown that s fit nicely as harmonic functions of ϕ ; however, here there is no evidence that $\log(s)$ behaves in the same fashion, especially in the case the authors consider up to the first order.

We have fixed the confusing statement mentioned. Regarding the periodicity of s and $\log(s)$, while both can be represented as Fourier series, we opted to parameterize $\log(s)$ with a truncated Fourier. This was motivated as $\log(s)$ (due to the \log) is in general a smoother function that requires less harmonics to fit typical gene profiles. This is now emphasized in the methods (see Methods 4.1):

“For manifold-learning, we estimate the position of each cell along the circular cell cycle manifold (ϕ) as well as the Fourier series coefficients for each gene (v) used to model the expectation of \log spliced counts (ElogS), which are themselves modeled from the real data and a Negative Binomial.”

Models using truncated (even with only one harmonic) Fourier series in the log spaces have been frequent in the literature and can be found at (Liang et al. 2020; Auerbach, FitzGerald, and Li 2022; Riba et al. 2022; Talamanca, Gobet, and Naef 2023).

To further clarify this point to the reviewer, we provide supplemental panels that demonstrate in a side-by-side manner the gene harmonic fits in both exponential and logarithmic space (Fig. R1.1). These new figure panels are for the simulated data (Fig. 2), the FACS-sorted mouse embryonic stem cells (Buettner et al 2015; Fig. 3d), and the human fibroblasts (Riba et al 2022; Fig. 3k).

21

Figure R1.1. Comparison of gene harmonic fits illustrated in exponential and logarithmic space across multiple datasets. (a) Left: Spliced (blue) and unspliced (red) gene fits obtained with VeloCycle on simulated data in exponential space (from Fig. S2c of first submission). Right: Spliced (blue) and unspliced (red) gene fits obtained with VeloCycle on simulated data in logarithmic space. (b) Top: spliced gene fits obtained with manifold-learning on FACS-sorted mouse embryonic stem cells (Buettner et al. 2015) in exponential space (from Fig. 3d). Bottom: spliced gene fits in logarithmic space. (c) Left: Spliced gene fits obtained with manifold-learning on human fibroblasts (Riba et al.

22

2022) in exponential space (from Fig. 3k). Right: Spliced gene fits obtained with manifold-learning on human fibroblasts in logarithmic space.

3. (critical) Eqn. (12) is wrong. ω should not multiple γ . Since the same wrong equation also appears in Figure 1, I'm concerned whether this is a typo, or the implementation is wrong.

The reviewer is absolutely correct that we incorrectly reported equation 12 in both the methods section and Figure 1. This is a typo and we thank the reviewer for catching it. We have corrected this error in both Figure 1 and the method description.

To clarify, ω should be inside the parenthesis, and this was an error at the level of transposing the equations into the text editor. The implementation and code is not at all affected (see GitHub) because it is anyways implemented as the logarithmic form presented in equation 11. Equation 12 was only reported in the methods to clarify the relationships, and the equation in Figure 1 was adopted from the methods, transferring this incorrect formulation.

23

Reviewer #2:

In this very nicely drafted manuscript, Alex, et. al. proposed, VeloCycle, a generative model for RNA velocity and a Bayesian inference approach aimed at addressing the fragility and heuristic reliance of existing RNA velocity algorithms. The results from VeloCycle are thoroughly validated on both the simulation, in vivo dataset of cell cycle and even the live imaging and EdU labeling experiments. This study also reveals interesting regionally-defined proliferation patterns in neural progenitors and how genetic perturbations affect cell cycle speed in a stage dependent fashion.

We thank the reviewer for their positive comments on our manuscript and for recognizing the extensive efforts made to validate the performance of our model in multiple biological settings.

In the following, I will list a few key comments that should be addressed to improve the presentation of this study.

About the manifold-constrained RNA velocity. The idea of constraining the RNA velocity on the latent manifold of the single cell gene expression to avoid RNA velocity rapidly escaping the cell manifold is interesting. In this study the authors explicitly deal with a specific type of manifold – a circular manifold of the cell cycle progress and leverages Bayesian framework and variational inference based method to estimate the kinetic parameters and RNA velocity vectors (Eq. 4 in their paper). However, an alternative and seemingly more flexible approach is to first learn RNA velocity using existing methods, including velocity, scVelo, other deep learning based methods, such as LatentVelo, UniVelo, DeepVelo, VeloVI, etc., but then project the velocity to the tangent space of the underlying manifold.

We appreciate the reviewer's insightful comment on the projection approach for RNA velocity estimation, highlighting an alternative strategy that involves projecting velocity vectors onto a low dimensional embedding obtained by UMAP or another ML-based embedding method. We believe the approach mentioned is valid but has a significantly different goal and effects.

We mention this possibility now explicitly in the discussion section of the main text:

“We note that projections and smoothing methods of the velocity field on low dimensional embeddings have been proposed and applied post-hoc to achieve a smoother, less overfit-prone vector field, which are particularly relevant for visualization purposes and data exploration (Farrell, Mani, and Goyal 2023; Gao, Qiao, and Huang 2022; Cui et al. 2024; Gu, Blaauw, and Welch 2022). Here, we propose, test and experimentally validate an explicit parametrization of RNA velocity as a vector field defined on the manifold coordinates that, from the beginning, considers tangency among its core assumptions (see Methods 1-4).”

24

This comment allows us to clarify the difference between these approaches and advantages of our strategy, which integrates the manifold structure directly into the RNA velocity estimation process. Unlike post hoc projection methods, our approach offers several key benefits.

Before detailing the benefits below, we would like to stress that in our manuscript, we do not define the manifold casually as “the output of a ML non-linear embedding method that approximates in low dimensional space distances in high dimensional space.” Rather, we define the manifold more technically as “a surface of points existing in the high dimensional space with an explicit parametric mapping to a low dimensional latent space, along which the dynamical process of interest unfolds (see Methods 1).” We believe clarifying this difference is crucial to understanding the unique advantages to our approach.

The benefits over a post hoc approach are:

- Obtaining a structured regularization: our method integrates the manifold's structure directly into the RNA velocity estimation, allowing for a more cohesive and informed fitting procedure. Unlike post hoc projection, which applies after all individual gene-wise velocity estimates are generated, our approach enables the simultaneous

correction and refinement of each gene kinetic parameter according to the manifold's constraints. This integration allows for a synergistic optimization, where the manifold influences gene fits, enhancing the overall coherence and accuracy of the RNA velocity estimates.

- Incorporating biological inductive bias: a key advantage of our method is its ability to incorporate different biological priors directly into RNA velocity estimation, including the topology and the type of gene expression variations we expect over it (through the basis functions). This is a capability that projection methods lack. By integrating the manifold structure, our approach considers variations that follow these priors and constraints. This direct integration facilitates the marginalization of confounders and orthogonal variations in the selected features, potentially enhancing the biological relevance and accuracy of the velocity estimates.
- Unified modeling framework: our methodology distinguishes itself by employing a single, end-to-end cohesive model that integrates RNA velocity estimation within the defined manifold structure. In contrast, post hoc projection methods sequentially apply procedures and fits, each with their own set of possibly inconsistent assumptions. By maintaining a unified end-to-end model, we ensure every aspect of our approach is aligned and consistent.
- Rigorous inference capability: a cornerstone of our approach is the ability to conduct statistically sound inference, which projection methods do not. This possibility is linked to our modeling choices, which see the manifold and velocity as the two top-level latent variables of a probabilistic generative model. The velocity latent variable is low dimensional (simplifying inference) and is statistically independent

25

from kinetic parameters before conditioning on data. This is in stark contrast to what happens estimating gene-wise velocities, which are expected to be highly correlated. Thus, naive bootstrap or other procedures considering estimated parameters uncertainties independently are at risk of being extreme and overconfident. In the paper we showed that studying posterior probability distribution of our model and adapting variational formulation makes it possible to control for these correlations on the latent parameters when conditioned on the data (Fig. 5 and S4).

We now add this paragraph to the main text:

“VeloCycle enriches velocity analysis by incorporating the manifold's constraints directly into the RNA velocity estimation process. This facilitates a structured regularization by unifying kinetic parameter estimation and the manifold's intrinsic geometry, ensuring coherence of the estimated quantities (see Methods 1). Additionally, our approach stands out for offering a unified end-to-end model, which is fit on raw data, avoids heuristics, promotes interpretability and has rigorous inference capabilities. Inference on the primary latent variables of this model – manifold geometry, velocity, and kinetic parameters – avoids the pitfalls associated with multiple gene-wise velocities (Gorin et al. 2022; Bergen et al. 2021). Specifically, it does not lead to the overconfidence stemming from considering each gene as independent and neglecting the correlation of their uncertainties (see Methods 3.2).”

This idea was previously attempted in the graph-dynamo paper (<https://www.biorxiv.org/content/10.1101/2023.09.24.559170v1>). In this study, the manifold is defined by the latent dimension that can be returned from PCA, autoencoders or other non-linear dimension reduction methods. The complex geometry of the manifold is then approximated by the nearest neighbor cells. The new tangent space projection (TSP) method demonstrates it can maintain the RNA velocity magnitude and thus time scale better than previous cosine correlation based kernels in both simulation and real datasets. It will be interesting to discuss graph-dynamo in this work to explain the rationale behind the VeloCycle approach. It will be also interesting to see some extensive benchmarks between graph-dynamo and the VeloCycle method.

We thank the reviewer for alerting us to the TSP in the recent graph-dynamo preprint. Indeed, it is exciting to think of VeloCycle as building a parallel with graph-dynamo. We think that these two methods diverge significantly in their implementation, specific aims and biological applications, and theoretical underpinnings, yet there are some common conceptual aspects. Specifically, the highlighted tangent space projection method tackles the same inconsistency of the manifold consistency constraints as in our framework. Overall, we believe this parallelism is reflective of a relevant limitation of previous methods that

starts to be recognized by the community and to which leaders of the field, including the authors of graph-dynamo, are currently proposing solutions.

26

VeloCycle differentiates itself primarily through its direct integration of the manifold structure within the RNA velocity estimation process, defining a model around tangency from the beginning (*ab initio*). This is in contrast with graph-dynamo's tangent space projection, which is a post hoc procedure similar to a convolutional filter. The substantial aspect VeloCycle gains out of this important difference is its inferential value as well as the other aspects mentioned in the response to the previous point.

This being said, from our study of the formulation in graph-dynamo preprint, it appears that TSP meaningfully addresses the tangency problem for visualization and there is no doubt that TSP is a more elegantly formulated projection than we had originally proposed in 2018 in the first RNA velocity paper (La Manno et al. 2018).

In conclusion, different approaches enrich the field's methodological repertoire, the two methods each offering distinct advantages and perspectives. What they have in common is the recognition of the manifold's tangency significance, but exploit it for different aims. We summarize the above points in the following statement, now added to the revised manuscript:

"The pivotal role of having a velocity estimate tangent to manifold structure has also been recognized by another recently-proposed method, graph-dynamo (Zhang et al. 2023), whose tangent space projection corrects post hoc the vector field for visualization and interpretation. VeloCycle accentuates the centrality of velocity tangency further by integrating manifold constraints directly into the estimation process, opening the possibility to exploit them for parameter identification and inference."

Unfortunately, it was not possible to perform benchmarking between graph-dynamo and the VeloCycle method due to challenges running the graph-dynamo method. The graph-dynamo documentation and tutorials on GitHub appear to still be under construction: there is a "WARNING: Under construction" note on the GitHub repo page (version May 2024), the Documentation Link leads to a dead-end page, and the two functions proposed in the "Quick Start" do not appear in the latest code base (`graph_dynamo.tangent_space_projection` and `graph_dynamo.graph_fokker_planck_equation`). Likewise, the sample notebooks present on GitHub appear incomplete and appear to use the dynamo/Aristotle framework rather than graph-dynamo.

This is totally understandable for a recently released preprint. Admittedly, we were able to identify in the code base the definition "tangent_correcting_velocity", an 8-lines routine, that analogous to the convolution described in the paper; however, finding out the right way to apply it to produce an output reflecting the use intended by its authors could not be determined and thus, a fair benchmark was impossible.

Overall, we have referenced graph-dynamo in our revised manuscript and mentioned the theoretical underpinnings including commonalities and differences. We remain open to

27

performance evaluations of graph-dynamo if they are deemed essential by the reviewer. Yet we note that the revised manuscript includes visualizations of (see answer below) the vector field and stream plots corresponding to our methods and these appear perfectly smooth and tangent, which we believe aligns with the aspect the requested benchmark would have scrutinized.

In this study, the author focuses on their own cell cycle dataset and claims that it is the first example of direct validation of RNA velocity estimation in units of real time. There are in fact several other cell cycle datasets with time resolved information based RNA metabolic labeling, including the scEU-seq (<https://www.science.org/doi/10.1126/science.aax3072>) and the sci-fate (<https://www.nature.com/articles/s41587-020-0480-9>). The time resolved scRNA-seq will naturally offer RNA velocity with real units (molecules per hour). It will be interesting to see how VeloCycle performs on these two datasets in comparison to the Dynamo approach (<https://www.sciencedirect.com/science/article/pii/S0092867421015774>) that explicitly leverages the metabolic labeling information.

RNA velocity methods based on metabolic labeling are indeed extremely relevant when thinking of estimating the velocity in real time units, and we thank the reviewer for bringing them into discussion. However, we think it would be misleading to equate classical RNA velocity methods such as velocity, scVelo and VeloCycle with metabolic modeling velocity estimation methods (such as the method dynamo) (Qiu et al. 2022) based on metabolic labeling measurements (scEU-seq, sci-fate). These two families of methods are based on significantly different data, with very different experimental designs, information content, limitations and costs.

Metabolic labeling vs classical scRNA-seq: single-cell metabolic labeling-based velocity estimates are generally more powerful and reliable than classical scRNA-seq based estimates. However, producing metabolic labeling data is not trivial:

1. The experimental design is significantly more constrained and almost exclusively limited to in vitro samples.
2. The techniques require specific experimental methods, none of which are commercially available.
3. The experimental procedure is more intensive and more expensive (pulse and chase data needs to be collected, often this means 6 times more data needs to be obtained)
4. Computational methods for velocity estimation on metabolically-labeled data are not retroactively applicable to normal scRNA-seq data (i.e., it can't be run on atlases generated with classical scRNA-seq).

Taken together, these are enough motivations to justify the development of both types of velocity methods, recognizing the significant gap between the data they are based on. To be convinced of this fact, it should be enough to keep in mind the number of papers featuring data produced by the two methods: traditional single cells surpassing in frequency of adoption metabolic labeling by at least two orders of magnitude.

28

A well-crafted RNA velocity method which uses metabolic labeling information is bound to be more accurate, since it has access to 24 data tables: Spliced, Unspliced, Labeled Spliced, and Labeled Unspliced, each for 6 different timepoints (Battich et al. 2020).

Thus, we believe that Dynamo, which uses metabolic modeling, should provide a state-of-the-art gold standard for the evaluation of VeloCycle, which only uses two matrices, Spliced and Unspliced. Therefore, we perform this analysis as part of the response to the next point (least action path analysis on scEU-seq data) and as a validation of the estimated velocities and cell cycle periods (Fig. S15 and Fig. 6o-p). We also include dynamo and the sci-fate dataset in our broader benchmarking analyses (Fig. S13). We want to make sure these points are clear, so we bring it up twice in the revised manuscript.

In the introduction:

“Single-cell metabolic labeling measurements methods can solve some of these problems (Qiu et al. 2022), but their applicability is much more limited to specific design and in vitro settings.”

In the discussion:

“Although we advocate the use of metabolic labeling techniques, which with their more informative experimental design, targeted chemistry, and structured data are likely to allow better estimation of velocity kinetic parameters, we recognize this practice is limited in application. Thus, the design of experimentally-validated or manifold-constrained RNA velocity methods such as VeloCycle is an important effort with general purpose applications by the community.”

In Eq. 5 of their paper, the author defines the integral to calculate the time needed between two states S_0 , S_1 . Since there are an infinite number of paths connecting two states in the state space, how do the authors define the path taken for the integral? I wonder whether the least action path (LAP) principle proposed in the Dynamo paper can be used here? In fact, the LAP method can be easily used to calculate the cell cycle time by specifying the starting point and end point on the cell cycle manifold, followed by running the LAP prediction. It will be interesting to compare the estimates of cell cycle time from VeloCycle with those from Dynamo.

We thank the reviewer for bringing up the problem of multiple paths. This is a very relevant

question and the exact one we were thinking about years ago when we started this project. Considering it deeply is what led us to develop this framework, which actually is meant to avoid the conundrum of finding the optimal path among an infinite number of possible paths connecting two states.

29

Our cell cycle model is “inverted” in the sense that the manifold exists at the top of the hierarchy of a generative probabilistic model. Consequently, there are actually not an infinite number of paths connecting two states in VeloCycle but only one. There is only one process in our model (e.g., the cell cycle), the manifold is the expectation path of that process, and measurements from cells are considered to be realizations from it. The reviewer’s point will, however, be valid for higher dimensional manifolds where indeed a continuum of paths is possible. In this case it is both relevant to calculate the time taken for a specific path, or to find the path of shortest/longest time, depending on the specific question at hand.

Taking the reviewer’s comment into account, we now perform LAP estimation with dynamo and compare it to our cell cycle period estimate 1D manifold. However, because of the reasons above, we do not see a canonical way to use the LAP method on top of our method; rather, it is interesting to compare the path from dynamo+LAP and VeloCycle across the cell cycle phase ϕ .

Using the scEU-seq dataset recommended to us by the reviewer, which contains cycling Fucci-labeled RPE1 cells (Battich et al. 2020) (Fig. 6p), we successfully obtained cell cycle velocity estimates with both dynamo (Fig. S15a) and VeloCycle (Fig. S15c). Since we have Fucci measurements for each cell, we can compare the phase estimate obtained by both approaches to a ground truth phase estimation; while the estimate from VeloCycle largely correlated to the Fucci ground truth (Pearson’s $r=0.78$), the scvelo velocity pseudotime estimate correlated less (Pearson’s $r=0.60$) and shows a qualitative error: a “split” in the phase estimation (Fig. S15b and S15d). Note that because of this dynamo needs to rely on the Fucci ground truth phases as underlying embedding for all the estimations indicated below, whereas VeloCycle does not need to.

Furthermore, we were able to reconstruct a cell cycle period using three different approaches (Fig. 6p-q; Fig. S15):

- (1) VeloCycle: the standard approach described in our manuscript using only spliced and unspliced tables.
- (2) Dynamo and LAP using only unspliced and spliced counts, this is to consider a fair contender of VeloCycle as it uses the same data, and it has the only advantage of having access to the Fucci ground-truth.
- (3) Dynamo and LAP using metabolically labeled data. This includes the following 24 tables unspliced unlabeled, unspliced labeled, spliced labeled, spliced unlabeled each at six different pulse time points that are combined to yield the final results. We consider the comparison to this case akin to a gold standard of estimation from sequencing data.

30

Figure S15. VeloCycle, dynamo, and least action path analysis of Fucci-RPE1 cells profiled scEU-seq. (a) RNA velocity streamline plots obtained with Dynamo (Qiu et al. 2022) using metabolically-labeled (scEU-seq) data for 2,793 single RPE1 cells (Battich et al. 2020), represented on the Fucci-defined embedding space and colored by pseudotime (top) or Fucci score (bottom). (b) Scatter plot comparison of cell cycle phase pseudotime from scvelo (top) and VeloCycle (bottom) compared to the ground truth Fucci phase assignment acquired for dataset in (a). Pearson’s circular correlation coefficient is indicated in red. (c) Posterior estimate plot of cell cycle velocity for RPE1 cells obtained with VeloCycle. (d) Scatter plot of per cell Fucci signal (GMNN-GRP in green; CDT1-RFP in red) along the VeloCycle phase estimate obtained with manifold-learning.

Figure 6. Validation of computationally inferred velocities by cell tracking and labeling experiments. [...] (p) Illustration of the experiment designed of the scEU-seq (Battich et al. 2020) generating the 24 tables used together with Dynamo (Qiu et al. 2022) to produce a gold standard cell cycle period estimate from sequencing data. (q) Left: a schematic of the different experimental measurements, manifold, and cell path inference approaches taken by VeloCycle, Dynamo (without

metabolically labeled information) and Dynamo-metabolic (with metabolically labeled information) models. Dynamo uses the FUCCI signal as the manifold, whereas VeloCycle learns the manifold independently. Dynamo uses the least action path (LAP) approach and multiple time points of

31

metabolically labeled data to extrapolate a cell cycle period, whereas VeloCycle uses the manifold-constraint settings to recover a cell cycle period. Right: plot showing the estimated cell cycle period as obtained by VeloCycle, dynamo, and dynamo-metabolic models. The violin plot displays the posterior distribution output by VeloCycle, and the circles are individual evaluations of the LAP starting from different start and end cells; red stars indicate the means.

We found that the VeloCycle posterior probability aligned overall with Dynamo using metabolically labeled information (Dynamo Metabolic), corresponding to the biologically-reasonable cell cycle period of around 20 hours (Fig. 6q, right, “VeloCycle” and “Dynamo Metabolic”). Remarkably, VeloCycle is able to achieve this feature (1) using only the unspliced/spliced counts (no metabolically labeled time course) and (2) without the FUCCI embedding as a ground truth manifold. Dynamo in its U-S table-limited-version was unsuccessful at recovering an accurate or stable estimate for the cell cycle period, as opposed to Dynamo Metabolic and VeloCycle (Figure 6q, right, “Dynamo”).

This exciting new result suggests that VeloCycle can obtain a biologically-accurate estimate of the cell cycle period similar to that obtained by Dynamo, despite the latter achieves it with a much more complex experimental effort and with one order of magnitude (i.e. 24 tables vs 2) more informative metabolically labeled data. Taken together, we have included this information in the results section and main Figure 6, directly after our experimental validation by time-lapse microscopy and cumulative EdU labeling (Fig. 6p-q). We thank the reviewer again for suggesting this path of action.

The paper only demonstrates a single case of cell cycle, another simple manifold is a linear trajectory. Can we also include an analysis for such a manifold structure? I suggest running analyses on either the sci-fate dataset (which include both the cell cycle process and a linear response after DEX treatment) and the scNT-seq dataset (the neural polarization experiment, <https://www.nature.com/articles/s41592-020-0935-4>). Extensive benchmarks with Dynamo and other tools should also be performed on these datasets.

We thank the reviewer for pointing us towards these additional cell cycle datasets, as many of the standard datasets used by the RNA velocity community to benchmark methods contain limited or no proliferating cells. As demonstrated with the previous comment, we have now performed complete analysis of the recommended scEU-seq dataset (Battich et al. 2020) using VeloCycle and dynamo (Fig. 6p-q and S15). Furthermore, the sci-fate dataset of A549 cells (Cao et al. 2020) is included in our extensive benchmarking of five different velocity methods: VeloCycle (this study), scvelo (Bergen et al. 2020), cellDancer (Li et al. 2023), VeloVAE (Gu, Blaauw, and Welch 2022), and Dynamo (Qiu et al. 2022) on the following datasets:

5. Simulated data (previously show in our sensitivity analysis in Fig. 2)
6. Cycling human dermal fibroblasts (Fig. 6a and Capolupo et al. 2022)
7. A549 cells newly added from (Cao et al. 2020)
8. Cycling RPE1 cells (generated for this study, Fig. 6h)

Specifically, we scrutinize all shared output between the methods with the following metrics:

32

- Cross boundary direction correctness (Fig. S13a-b) established by (Qiao and Huang 2021), which is further evaluated across different manifolds for maximal transparency
- Velocity consistency (Fig. S13c) established by (Gayoso et al. 2024)
- Mean squared error of the spliced and unspliced expression (Fig. S14a-b)
- Mean squared error of the velocity kinetic parameters (Fig. S14c)
- Visual comparison of the vector fields on the low-dimensional embedding (Fig. S13d)

VeloCycle compares positively to the other methods, with the cross-boundary direction correctness and velocity consistency metrics performing better with VeloCycle on most of the datasets. Specifically, VeloCycle achieves the best overall in cross boundary direction correctness, one of the most widely-used metrics in the field, even when the clusters are defined by other method’s embeddings. An interesting exception is the MSE for the spliced and unspliced, which is higher in our method and lower for VeloVAE and cellDancer. This is in line with what was already mentioned in the paper: we expect other methods to have a higher degree of freedom and thus increased flexibility (lines 112-116). However, this is accompanied with a higher tendency to overfit as shown by the comparably higher variance in

the spliced/unspliced expectation when even ground truth is flat (Fig. S14b).

33

Figure S13. Velocity method benchmarking with cross-boundary direction correctness and velocity consistency. (a) Bar plots of mean cross boundary direction correctness (CBDir; Qiao et al. 2021) computed on four datasets (simulated data, human fibroblasts, A549 cells, and RPE1 cells) and with five different velocity methods (VeloCycle, scvelo, cellDancer, dynamo, and VeloVAE). CBDir correctness requires a ground truth direction between clusters; therefore, the score was computed for clusters obtained on each method's embedding (see Methods 8.3). The black dot represents the

34

CBDir score calculated using that specific velocity method's embedding, whereas the black line indicates the standard deviation across scores computed using all five embeddings and cluster annotations. (b) Heatmaps of the CBDir scores computed pairwise for each method using cluster relationships defined on each different embedding, for all four datasets benchmarked. (c) Box plot of cell-wise velocity consistency score (Gayoso, Weiler et al 2023; see Methods 8.4) computed on the same datasets and methods as in (a-b). (d) Low dimensional embedding plots with grid-wise velocity vector fields computed for on human fibroblasts (top) and A549 cells (bottom). Cells are colored by categorical phase to enable the visual inspection of vector field direction correctness.

Figure S14. Velocity method benchmarking against ground truth values with simulated data. (a) Box plots of the spliced (left) and unspliced (right) gene-wise mean squared error (MSE) obtained between the expected un/spliced counts and the simulated ground truth for five benchmarked velocity methods (VeloCycle, scvelo, cellDancer, dynamo, and VeloVAE). (b) Box plots of the gene-wise standard deviation of expected spliced (left) and unspliced (right) expression estimated by five different velocity methods. (c) Box plots of the gene-wise MSE for the velocity kinetic parameters (γ , β , and γ/β ratio) compared to the simulated ground truth for five velocity methods. Simulated data was generated as presented in Fig. 2 for 3,000 cells and 300 genes.

Minor comments:

Fig 1. This figure includes several equations that should be better explained in the figure caption. For example, in panel d, you may want to explain how to calculate the RNA velocity by assuming it is defined as the inner product (or the projection) of the velocity ($V(x(t))$) to the tangent space via the gradient of S_g on the latent space x .

We thank the reviewer for the comment. We have revised the figure caption accordingly to provide more detail and clarity.

35

In addition, this section discusses one unique feature of VeloCycle is to perform statistical velocity significance test. This can be an important point and it will be interesting to see more application of this method in performing the significant test across different samples or batches for meaningful biological discovery related to RNA velocity shifts.

We thank the reviewer for highlighting what we consider one of the key advancements of VeloCycle and our manifold-constrained framework of RNA velocity formulation: its inference value to ask and statistically test scientific hypotheses.

We demonstrated on numerous datasets and in multiple biological contexts the ability to identify statistically significant differences between RNA velocity estimations across different samples or batches; we list them below:

- Comparison of biological replicates using RPE1 cells: this is the first example that set the stage for the following demonstrations. By examining the overlap between the posterior distributions, we are able to correctly identify that there is no difference in cell cycle speed between two biological replicates of the same RPE1 cultures run by the same experimenter (Fig. 4).
- Lung adenocarcinoma PC9 cell line in response to erlotinib: we found significant velocity differences between the untreated and day 3 of treatment with a subtoxic concentration of erlotinib (Aissa et al. 2021) (Fig. 7b). The results of VeloCycle highlight that the significant velocity shifts occur exclusively in the G2/M phase and

not in G1 or S. This finding was further validated by the identifications of individual genes with reduced phase delay, among which were anaphase-promoting complex member CDC27, cyclin-dependent kinase inhibitor CDKN3, and centrosome scaffolding factor ODF2, all of which important roles in M phase.

- Proliferation differences of neural progenitors at different regions and timepoints: VeloCycle estimation revealed differences between the proliferation speeds of progenitors (La Manno et al. 2021) at different regions: at embryonic day 10, radial glial progenitors in the hindbrain divided more quickly than those in the forebrain (Fig. 7d). A more nuanced visualization of the gradient was obtained using in situ hybridization spatial transcriptomics (HybISS) data and the BoneFight algorithm (Fig. 7e). We also found that at embryonic days 14 and 15 (E14 and E15), progenitors across all brain regions exhibited similar proliferation speeds with no significant velocity differences (Fig. 7f). This later uniformity corresponded with the transition of midbrain and hindbrain progenitors to a less proliferative state, while forebrain progenitors, which developed more slowly at E10, continued to proliferate (Fig. 7g-i).
- Re-analyzed Perturb-seq data identifies genes knockdowns affecting cell cycle velocity: we applied VeloCycle to a genome-wide Perturb-seq dataset (Replogle et al. 2022) where hundreds of gene knockouts were introduced into RPE1 cells using CRISPR, followed by scRNA-seq. VeloCycle revealed a significant increase in cell cycle period from 25.6 ± 1.3 hours in non-targeted control cells (NT) to 30.9 ± 1.3 hours in cells with cell cycle-related gene knockouts (CC-KO) (Fig. 8a). Knockouts of

36

specific genes, such as MCM3 and MCM6, significantly impaired cell cycle speed, while knockouts of certain splicing and mRNA processing genes, such as DBR1, PRPF3, and PRPF31, either decreased or increased cell cycle speed (Fig. 8d-e). These findings demonstrate that VeloCycle can effectively assess the impact of gene knockdowns on cell dynamics in large-scale perturbation screens using transfer learning (Fig. 8f-g)

Furthermore, with an extended 1D interval model to study differentiation in the revised manuscript, we have performed additional demonstrations with biological relevance on new datasets for the proof-of-concept:

- Simultaneous analysis of cell cycle and pancreatic beta cell differentiation: we studied a mouse pancreatic endocrinogenesis dataset at E15.5 (Bastidas-Ponce et al. 2019) to analyze both cell cycle and beta cell differentiation velocities (Fig. S9). For beta cell differentiation, we used the 1D interval model, confirming that the estimated velocities aligned with the expected timeline of differentiation from Ngn3 high EP to pre-endocrine and mature beta cells (Fig. S9g-j). The 1D non-periodic model estimated differentiation times aligned with the cell proportions distribution exposed by single-cell sampling at different timepoints (Fig. S9h) and highlighted gene-specific unspliced-spliced delays (Fig. S9j). At the same time and in the same dataset, VeloCycle could be applied to infer a cell cycle speed of proliferating ductal cells (Fig. S9d-e). These results demonstrate potential use cases for frameworks providing timeline-compatible velocity estimates across different biological processes.
- Granule cell differentiation slows down over time: we analyzed a mouse dentate gyrus dataset (Hochgerner et al. 2018) to estimate RNA velocity during granule cell differentiation across postnatal time points (P0 and P5) (Fig. S10). VeloCycle's velocity estimates showed differentiation speeds compatible with the proportions of cell types sampled and significantly reduced at P5 (Fig. S10d-e). Gene-specific unspliced-spliced delays were visualized, revealing distinct kinetic parameters and expression dynamics across time points (Fig. S10f-h).
- CA2-3-4 pyramidal neurons take five days to differentiate: we detected a slower differentiation trajectory to be one for CA2-3-4 pyramidal neurons (Fig S10i-j), where velocity estimation yields a period of 116h (Fig. S10k). This is internally validated by the fact that at P0, mostly non differentiated CA neurons are present, whereas five days later at P5, nearly all cells are in the differentiated state (Fig. S10l).

Overall, we think this is an extensive showcase of examples both in terms of variety of analyses types, number of datasets, and types of findings.

37

Figure S9. Modular and manifold-constrained velocity analysis of the cell cycle and beta cell differentiation during mouse pancreatic endocrinogenesis. (a) UMAP of pancreas dataset (3,696 cells; Bastidas-Ponce et. al 2019) from mouse E15.5, colored by published cell types. (b) UMAP of

dataset in (a), colored by the selected cell subsets (in red) extracted to estimate RNA velocity of the cell cycle (left; Ductal; 916 cells) and beta cell differentiation (right; Ngn3 high EP, Pre-endocrine, Beta; 1,825 cells). (c) Low-dimensional plots of the gene expression manifolds estimated for the cell cycle

38

(left) and beta cell differentiation (right). The cell cycle manifold was obtained using manifold-learning in VeloCycle, whereas the differentiation manifold was obtained with diffusion pseudotime performed on the first two principal components. (d) Posterior estimate plot of the cell cycle speed from VeloCycle for cycling ductal cells. (e) Violin plots of the posterior distributions for cell cycle velocity from (d), stratified by categorical phase. (f) Scatter plots of selected genes, illustrating the estimated expected spliced (blue dashed line; ES) and unspliced (red dashed line; EU) along the cell cycle manifold as compared to measured spliced (blue; S) and unspliced (red; U). The peak unspliced-spliced delay is indicated by vertical lines and the value at the top left of each plot. (g) Velocity posterior estimate plot obtained for beta cell differentiation using the 1D interval model, extended upon the manifold-constrained framework. (h) Stacked bar plot of the cell type proportions along the differentiation axis in four different scRNA-seq datasets (E12.5, E13.5, E14.5, and E15.5) from the original study. (i) Scatter plots demonstrating the relationship between the velocity kinetic parameters ($\log\gamma$, $\log\beta$) and total (spliced, unspliced) counts. Pearson's correlation coefficients are indicated in red. (j) Scatter plots of selected genes, illustrating the estimated expected spliced (blue dashed line; ES) and unspliced (red dashed line; EU) along the cell cycle manifold compared to measured spliced (blue; S) and unspliced (red; U). In panel (d) and (g), the white line indicates the mean over 200 posterior samples, and the black line indicates the full posterior interval. The cell cycle period (d) and the beta cell differentiation process time (g) is indicated at the top left of the respective plots.

39

Figure S10. Manifold-constrained velocity analysis across cell types and developmental stages in the mouse dentate gyrus. (a) UMAP of mouse dentate gyrus dataset (18,213 cells; Hochberger, Zeisel, et. al 2018), colored by published cell types (top) and postnatal time point (bottom). (b) UMAP

40

of dentate gyrus dataset, colored by the selected data subsets (in red) extracted to estimate RNA velocity of the granule cell differentiation lineage (Nbl1, Nbl2, ImmGranule1, ImmGranule2, Granule) at the P0 (left; 1,876 cells) and P5 (right; 4,968 cells) time points. (c) UMAP of data subsets in (b), colored by the diffusion pseudotime applied as the low-dimensional manifold during velocity inference. (d) Violin plots of the posterior estimates obtained on the entire granule cell differentiation lineage at the two time points. (e) Bar plots of cell type proportions in the granule lineage at P0 (top) and P5 (bottom) relative to the total number of cells at each time point. (f) Scatter plot of the relationships between the velocity kinetic parameters, total spliced, and total unspliced counts in the data from (d). Pearson's correlation coefficients are indicated in red. (g) Scatter plots of four selected genes, illustrating the estimated expected spliced (blue dashed line; ES) and unspliced (red dashed line; EU) along granule cell differentiation (d) compared to measured spliced (blue; S) and unspliced (red; U). The peak unspliced-spliced delay is indicated by the vertical lines and the value at the top left of each plot. (h) Scatter plot of the mean unspliced-spliced expression delay for 237 shared genes during granule cell differentiation between P0 and P5 time points. Gene dots are colored by peak expression phase. (i) UMAP colored by the selected data subsets (in red) to estimate RNA velocity of the CA differentiation lineage (Nbl1, Nbl2, CA, CA2-3-4). (j) UMAP of lineage in (i), colored by the diffusion pseudotime applied as the low-dimensional manifold during velocity inference. (k) Violin plot of the posterior estimate obtained on the CA2-3-4 cell differentiation lineage using both P0 and P5 cells. (l) Bar plots of cell type proportions in the CA and CA2-3-4 clusters at P0 and P5 relative to the total number of cells at each time point. 200 posterior samples were generated in (d) and (k), and the estimation of the total differentiation process duration is indicated in black at the bottom of the x-axis.

Fig 2. Minor comments on the figure caption. In panel l, the bottom plot is not the boxplot instead it is the point plot with the standard deviation.

We thank the reviewer for catching this mistake, we have corrected the figure legend for Fig. 2l to read "point plots with standard deviation" instead of "box plots."

Fig 3. It will be interesting to see the quiver plot and the streamline plot of the RNA velocity vector overlaying the manifold in figure 3a. It may be also worth to see the RNA velocity plot for just two cell cycle genes (a plot where the x, y axis are gene expression of two genes and the quivers are the RNA velocity for this two genes in each single cell).

We show below that both the quiver and streamline plots asked by the reviewer can be

produced and look ideal, which is expected considering our formulation involves a tangency constraint. The reason why we decided not to show it originally was deeply reasoned and not casual. We aimed for consistency between the new philosophy of the modeling framework introduced and the way we display the object we estimate. In other words, we believe that reflecting the change of estimation paradigm with an appropriate change of representations is important.

In our framework, velocity is an explicit function of the low dimensional manifold from which, for each cell, spliced and unspliced counts are considered realizations from random variables with expectations that are functions of a latent manifold coordinate and the velocity function. In that sense, a cell-wise velocity is not an object that exists naturally in our framework. We can force its computation by taking difference of expectation between the

41

current expected spliced counts (ElogS) and the expectations obtained by updating the position (ϕ) and applying the velocity function (ω), but there is no gene-cells object representing a velocity that is ever computed across the genes in any step of our framework.

A typical quiver plot is conceptually different from the above notion: it is computed for each single cell or on a grid. Likewise, it is meant to scrutinize for “tilts of arrows” and is typically computed using the high dimensional gene-wise velocity, with a heuristic projection on a ML embedding. This is all in contrast with our new framework.

Even the streamline plot gives a misleading perspective to our model, as its streams are typically segmented and shows parallel trajectories even for 1D data like ours. As manifold is 1D, trajectories are not realizations of the model, only U and S counts are, and they descend from the velocity and manifold, which are the latent variables hierarchically at the top of the model.

We stress: these plots look ideal and there is no other reason than a conceptual one not to feature them in the paper, but we deeply value the conceptual aspect of our contribution. Thus, if the reviewer does not object, we would be keen on reducing their usage to a minimum and to supplemental figures, bringing them up only in a part of the paper where they can be at least a bit coherent with the narrative.

We added to the revised manuscript Fig. S16, which extends Fig. 4 using the RPE1 dataset; this is the earliest point in the narrative of the paper after we fully introduce both manifold-learning and velocity-learning. These plots include the cell-wise velocity quiver and streamline plots (Fig. S16a) and three gene-wise velocity quiver plots (Fig. S16b) for RPE1 cells (see new figure panels below). The analogous plots for mouse embryonic stem cells asked by the reviewer are provided below as Figure R2.1.

42

Fig. S16. Velocity vector field representations of VeloCycle estimates in RPE1 cells, related to Fig. 4. (a) RNA velocity quiver plot (left) and streamline plot (right) for 14,259 RPE1 cells from Fig. 4a, colored by their categorical phase assignment (G1, S, G2/M). (b) RNA velocity gene-wise quiver plots for three marker gene pairs corresponding to distinct categorical phases in RPE1 cells (G1 marker SON and S marker CCNE2; S marker MCM4 and G2M marker TOP2A; G2M marker MKI67 and G1 marker SON).

43

Fig. R2.1. Velocity vector field representations of VeloCycle estimates in mouse embryonic stem cells, related to Fig. 3. (a) RNA velocity quiver plot (left) and streamline plot (right) for 279 mouse embryonic stem cells (mESC) from Fig. 3a, colored by their FACS-sorted categorical phase (G1, S, G2/M). (b) Gene-wise RNA velocity quiver plots for three marker gene pairs corresponding to distinct FACS-sorted phases in mESCs (G1 marker Hat1 and S marker Trp53; S marker Mcm4 and G2M marker Ube2c; G2M marker Top2a and G1 marker Ccng1).

Fig 4. Minor point. The text “Pearson’s correlation is indicated in red” should be for Panel C but it is currently written for panel b.

We thank the reviewer for catching this mistake, we have corrected this in the figure legend to refer to Fig. 4c instead of 4b.

Fig 7. The author uses HybISS approach and the BoneFight algorithm to reveal the region specific cell cycle pattern along the forebrain-midbrain-hindbrain axis. It may be interesting to directly leverage the single cell data from Stereo-seq ([cell.com/cell/pdf/S0092-8674\(22\)00399-3.pdf](http://cell.com/cell/pdf/S0092-8674(22)00399-3.pdf)) to perform such analyses because it gives the

44

real spatial information and at the same has the intron and exon information. The slices used for the Cover figure in that issue may be a good idea to try.

We thank the reviewer for this suggestion. Stereo-seq is indeed an impressive technology we have looked at with great interest, and this suggestion for an application “out-of-distribution” data type is highly compelling.

We downloaded the data (Chen et al. 2022), specifically the slides suggested by the reviewer, and verified that the resolution and segmented format of the data was suitable for a single-cell analysis of this spatial data (Fig. R2.2a). For cell cycle genes, the level of detection of the spliced were one order of magnitude lower than typical single cell data, which was sufficient in ensemble to define a cell cycle phase in the telencephalon (Fig. R2.2b). Moreover, the cell distribution was in line with the expected contrast between the proliferation-active progenitors located in the ventricular zone and the differentiating mantle zone (also Fig. R2.2b). This was very encouraging. Unfortunately, when inspecting the unspliced counts, we noted an even lower detection than for the spliced (on average 50 times less deep than single-cell data) and, and a low correlation with the levels of unspliced (Fig. R2.2c-d) in the scRNA-seq data used for VeloCycle analyses in our manuscript (see Fig. 7d).

Indeed, when loading the data, the VeloCycle library throws the warning we set (based on the sensitivity analysis in Fig. 2) to inform the user when the model is about to be run very far from the safe detection levels of counts. Note that we were able to infer a cell cycle manifold (Fig. R2.2e), which suggests that there is simply too limited information in the unspliced counts of the cell cycle related genes to extract a velocity (manifold-learning only uses the spliced counts), despite the clear presence of proliferating cells.

Normally, we should stop at the warning, but if we proceed anyways and run inference, we observe: (a) expected unspliced trends that are very low, (b) often not coherent with the spliced, (c) small delays. Most importantly the output of velocity inference is a wide confidence interval with SVI (Fig. 2.2f) and that even spans zero when estimated using MCMC (Fig. R2.2g). In other words, the information in the data is not sufficient to reject the null hypothesis of no cell cycle velocity. We are particularly excited that the model displays this desired behavior facing poorly poor unspliced signal, as other velocity models would just output in any case a result without notion of its confidence, VeloCycle’s credible interval inspection protect users from overfitting without noticing, and this from conclusion that are poorly not supported by the data to be reported in papers.

These results could be interesting as a demonstration to the reader, however we do not see how these results would fit in the narrative at the moment, so we are sharing them through an example notebook on our GitHub page.

45

Fig R2.2. VeloCycle analysis of spatially-defined Stereo-seq forebrain radial glia progenitors from the developing mouse embryo at E15.5. (a) Spatial visualization map of section from developing mouse brain E15.5 on which single cell resolution was recovered in the original study, colored by the cell types annotated (Chen et al. 2022). Subsection of the forebrain utilized for VeloCycle analysis is indicated by the red box. (b) Spatial visualization map of the radial glial forebrain cells from (a), colored by categorical cell cycle phase. Cells actively dividing (S and G2/M phases) are located closer to the ventricular zone. (c) Scatter plots of the total measured spliced (left) and unspliced (right) counts for 1,444 cell cycle genes between Stereo-seq (x-axis) and 10X v1 scRNA-seq (y-axis; data from Fig. 7d). The red dashed line indicates the diagonal, and the mean ratio y/x axis indicates the average gene-wise ratio of counts detected by scRNA-seq compared to Stereo-seq. (d) Box plots of total spliced (left) and unspliced (right) counts per cell for cell cycle genes, compared between Stereo-seq and scRNA-seq data. (e) Scatter plot of the estimated cell cycle phases of data in (b) obtained using manifold-learning and a LDA prior based on the categorical phases. (f) VeloCycle posterior plot of the RNA velocity estimate obtained using the SVI velocity-learning procedure. (g)

VeloCycle posterior plot of the RNA velocity estimate obtained using the MCMC velocity-learning procedure, where the initial conditions given were taken from the posterior output of (f). In both (f) and (g), the white dashed line indicates the mean of 500 posterior samples, and the black bars indicate the credibility interval across the entire posterior distribution.

46

Method sections: more explanation and clarification are needed for several equations. For example, $s: x \mapsto s(x) \in M$, it is confusing to me to understand why latent space coordinate x is mapped to the manifold through the function s . What is the function s ? Is this some function for the spliced RNA?

We have now reworded this part in the methods and added additional explanations to clarify the definition of the manifold, which is indeed done through the function $s(x)$ representing the spliced RNA.

47

Reviewer #3:

Lederer et al. developed a framework, VeloCycle, involving a generative model of RNA velocity and Bayesian inference, to estimate RNA velocity in a statistically-robust way that is consistent with the gene expression manifold which cells traverse. Specifically, they propose to parameterize RNA velocity as a field defined on the corresponding gene expression manifold. This is done in turn by simultaneous estimation of RNA velocity inference and manifold inference.

VeloCycle specifically is focused on the inference of RNA velocity along the cell cycle. In principle, it could be extended to additional topologies, but the parameterization could be a challenge. The authors applied VeloCycle to both in vitro perturb-seq data and in vivo samples, and uncovered proliferation modes and changes in cell cycle speed across samples, conditions and space in different samples.

Indeed, estimation of RNA velocity based on scRNA-seq data is known to suffer from limitations related to inherent noise and lack of robustness, which could be mitigated by the statistically robust method suggested in this paper.

The authors first validated VeloCycle on simulated data, showing that they can reconstruct both the manifold and velocity on top of it - the right phases of circular processes, rate parameters and fourier series coefficients. Next they showed good reconstruction of cell cycle phases for mESC smart-seq2 dataset (for which they also suggest new cell cycle markers), and human fibroblasts 10X chromium data.

Using estimates of cell cycle speed from live-microscopy imaging, the authors were able to, for example, estimate real-time progression, and reason about changing speed along the cycle.

The authors also statistically compared, and found differences in cell cycle progression and characteristics of an adenocarcinoma cancer cell line before and after drug treatment. They were additionally able to quantify differences in cell cycle speed across spatial zones of radial glial cells, which stabilized at later developmental stages. Finally, they propose a method for transfer learning that enables to infer velocity consistent with the cell cycle manifold using small numbers of cells, and demonstrate it in the context of perturb-seq dataset.

The paper is thorough and well written.

We thank the reviewer for the positive comments and accurate summary of our work. We also appreciate their recognition of the important strides VeloCycle makes towards defining an improved framework for RNA velocity estimation and inference.

48

The code is at <https://github.com/lamanno-epfl/velocycle>, and not at <https://github.com/lamanno-lab/velocycle> as is currently noted in the data and code availability section. The code itself is available, organized, documented, and includes multiple tutorials.

We are happy to hear the reviewer found the code and tutorials to be well organized and documented. We apologize for the wrong link to our GitHub page, which was written correctly in the “Data Availability” section but incorrectly in the “Code Availability” section. As the reviewer notes, the working link is <https://github.com/lamanno-epfl/velocycle>. This has been corrected in the “Code Availability” section of the manuscript.

It would be useful to give more intuition in the main text about the underlying assumptions and mathematical application of transfer learning in the context of VeloCycle.

We apologize for our brevity on this aspect in the main text.

In our work, we apply transfer learning to assign a manifold position (i.e., the cell cycle phase) to cells that come from Perturb-seq gene knockdown conditions containing a small total number of cells. This is particularly useful for multiple reasons. First, as shown in Fig. 2d and described in lines 273-278, manifold-learning performance is poor when it is forced on simulated datasets with very few cells and genes. This is similar to circumstances of working with Perturb-seq data or other screening-based experimental setups, where there is a limited number of profiled cells for many conditions (sometimes as few as 75 cells). Second, as an effect of some gene knockdowns, normal cell cycle behavior may become impaired, and cells may be restricted to a single phase of the cell cycle (i.e., G1/G0). Without transfer learning, this would pose a challenge for de novo estimation of the periodic Fourier series components (see Fig. 8c, Gene B/C Knockdowns, as an illustrative example).

Given these challenges, in order to infer accurate cell cycle phases for the Perturb-seq dataset, we first use manifold-learning to estimate gene harmonic coefficients (v_0 , $v_1\sin$, $v_1\cos$) on a larger set of non-targeting control cells, which are more evenly distributed throughout the various phases of the cell cycle. Then, we run manifold-learning again, but on the entire Perturb-seq dataset, conditioning the VeloCycle model on the coefficients learned in the first run. This allows cells belonging to a knockdown condition to be assigned to a position on the cell cycle manifold, but restricts those assignments such that they are based on realistic gene expression patterns learned on a larger and more informative dataset (with the term Δv allowing for batch effect expression differences, see Fig. S1a).

Ultimately, this approach could also be useful in other contexts, such as comparing between datasets belonging to the same species but taken from experiments with different sequencing depths or across different sequencing technologies.

We have now added a more detailed explanation of the transfer learning approach to both the main text (lines 827-841) and the methods section:

49

Main text: “Previous frameworks for RNA velocity have struggled to obtain reliable estimates using cell types or conditions for which only a limited number of cells are profiled. With recent single cell technologies designed to screen the effects of hundreds of genetic, environmental, or drug perturbations, there is a growing need to assess changes in cell dynamics using a small population of cells. VeloCycle, with its manifold-constrained velocity estimates, can explore RNA velocity contexts that were previously challenging: by conditioning the manifold-learning model on gene harmonic coefficients previously inferred from a large reference dataset, one can perform velocity inference using a smaller number of cells or using cells belonging only to one phase of the cell cycle. [...] To scrutinize the effect of individual gene knockout conditions on cell cycle speed, we employed a transfer learning approach in which we conditioned our manifold-learning model on gene harmonics previously inferred from the NT and CC-KO data subsets, assigning phases to a significantly larger population of 167,119 cells from 986 individual knockout conditions, some containing as few as 75 cells.”

Methods: “Condition-independent estimation of the periodic Fourier series components would be especially challenging on Perturb-seq knockdown conditions containing either (1) very few cells or (2) cells belonging to just one phase of the cell cycle. Therefore, in order to infer accurate cell cycle phases for these cells, we first performed manifold-learning for 5,000 training steps to estimate the gene harmonic coefficients (v_0 , $v_1\sin$, $v_1\cos$) on a larger set of non-targeting control (NT) and knockdown (CC-KO) cells, which are more evenly distributed throughout the various phases of the cell cycle. After, we ran manifold-learning again for 5,000 training steps, but on the entire

Perturb-seq dataset of 167,119 cells and 986 gene knockdown conditions. This time, we conditioned VeloCycle on the gene harmonic coefficients learned in the first step. This allowed cells belonging to each stratified condition to be assigned to a position on the cell cycle manifold, but restricted those assignments such that they are based on gene expression patterns learned on a larger and more informative dataset (with the term Δv allowing for batch effect expression differences). Finally, we then performed velocity-learning for 10,000 training steps on the entire dataset, estimating an individual constant velocity for each gene knockdown condition.”

The authors take the learned gene expression manifold as a natural constraint for velocity estimation. It would be interesting to discuss additional types of potential constraints, in the specific context of the cell cycle or potentially broader than that, that could aid in the reliable inference of RNA velocity.

We are interested in the potential of incorporating additional constraints to further refine and enhance the accuracy of RNA velocity predictions. In relation to the cell cycle, a specific constraint that could be considered is the expected drop in counts during cytokinesis and rise

50

during G1 (despite automatically obtaining that in the output of manifold-learning; see Fig. 3m, S5a, and S6a) to further improve performance. This could be implemented as an auxiliary loss, adding it to the ELBO, but it is perhaps not very elegant.

A related but more general constraint would be to assume that datasets contain cells that are spread across the manifold; with this, we could enforce an entropy loss term favoring solutions where cells are evenly distributed across the manifold without “gaps” (part of the latent space without cells) present. However, as with the Perturb-seq data in the previous comment, it is not always reasonable to constrain cells from a dataset to be evenly distributed across the entire cell cycle. Another constraint could be that the matrix of weights to the fourier basis function can be represented as a low rank matrix.

For both the cell cycle or any process for which strong biological priors are available, it would also be possible to provide priors (or harder constraints), thereby encapsulating the temporal dynamics of gene regulation to favor solutions where the sequential activation of genes during the process aligns with the literature. This could be enforced by introducing correlations in the gene-wise priors for basis functions (this is a bit simpler when using packet function-like bases) or via an additional helper loss function. Incorporating prior knowledge of gene regulatory networks into our framework could offer another layer of constraints, yet following that intuition would probably lead to another framework/class of models.

Another interesting form of regularization is to fit kinetic parameters as functions of genomic feature predictors, such as GC content, the number and length of introns, or the presence of intronic polyadenylation signals. This would not only refine kinetic parameter estimates but also embed an additional biological layer into the model.

Due to the limitations in space within the main text, we summarize these considerations into a concise statement in the discussion:

“Fourth, future refinements to VeloCycle could involve the addition of structured priors and other constraints: for the cell cycle, an auxiliary loss could be implemented to favor configurations reflecting total UMI drop after cytokinesis (Fig. 3m). More generally, the introduction of an entropy regularization could encourage an even distribution of cells across the manifold, and the imposition of low-rank constraints on basis coefficients could improve learning of high dimensional manifolds. Similarly, priors that postulate a specific sequential activation of genes or leverage knowledge of gene regulatory networks could inject valuable biological information. Finally, introducing genomic features as predictors for kinetic parameters is another promising strategy for regularizing the model.”

It would be useful to add visualized comparisons of low-d velocity estimates by VeloCycle relative to baselines which do not constrain velocity to lie on the low-d gene expression manifold.

51

We thank the reviewer for this suggestion, we have now performed the visualization of velocity estimates on the low-dimensional embeddings for a dataset of human fibroblasts

(Capolupo et al. 2022) and A549 cells (sci-fate dataset) (Cao et al. 2020) in the context of our comparisons to among five RNA velocity methods: VeloCycle, scvelo (Bergen et al. 2020), cellDancer (Li et al. 2023), VeloVAE (Gu, Blaauw, and Welch 2022), and dynamo (Qiu et al. 2022) (Fig. S13d).

However, we want to stress that we do not necessarily expect to see the inconsistency from these plots even when there are in the high dimensional space; in the process of plotting the velocity estimate in 2d the velocity is smoothed and redirected by a correlation-based heuristic projection procedure (i.e., that one we proposed in the original velocity paper, La Manno et al. Nature 2018). Hence, in other methods, the vector field is made consistent heuristically and post hoc, thus, in a way that neither improves kinetic parameters estimates nor allows for inference. In our method, we have the constraint as a modeling assumption from the beginning so that we can exploit it statistically.

Despite this important consideration, we show that in some instances, results fit by other models show cells do not progress along the cell cycle UMAP but rather move “backward” or “away” from the embedding space, such as on the A549 cells with Dynamo.

Figure S13. Velocity method benchmarking with cross-boundary direction correctness and velocity consistency. (d) Low dimensional embedding plots with grid-wise velocity vector fields computed for all five velocity methods on human fibroblasts (top) and A549 cells (bottom). Cells are colored by their categorical cell cycle phase to enable the visual inspection of vector field direction correctness.

Should we expect VeloCycle to work better/worse on particular types of data/experiments?

52

We appreciate this comment, as one of the general challenges with applying any RNA velocity model to an experiment is gauging the specific applicability in the specific scenario (e.g., is there enough data? Is data of the right quality? Do assumptions hold?). This was in part our motivation behind performing different sensitivity analyses to study how both manifold-learning and velocity-learning performance varies with the number of genes and give boundary estimates in terms of applicability (Fig. 2d and 2o).

In response to the reviewer’s request, the scenarios that trivially fall outside of these tested ranges will now raise a User Warning in our VeloCycle package, discouraging running the method. This is the case for one Stereo-seq spatial transcriptomics dataset that was proposed by Reviewer 2, where unspliced counts were more than one order of magnitude lower than our recommended boundary conditions, leading to very few genes being retained after initial quality control filtering, which is also described in our tutorials on GitHub.

Besides that, we have no evidence leading us to believe there are limits to VeloCycle as compared to other RNA velocity models (e.g. the model outputs consistent estimates for both cell datasets from in vivo or in vitro experiments and with 10X, SmartSeq, and scEU-seq technologies).

In consideration of the reviewer’s comment, in the revised manuscript we decided to further explore the behavior of the model in a specific setting for the cell cycle where one may suspect the method to perform worse: the case where the population of cycling cells analyzed is contaminated by an undetected proportion of non-cycling cell. With new sensitivity analyses, we now show that VeloCycle is resilient to a modest presence of non-cycling cells, yet we warn the users that in the case of a sizable G0 population (i.e. > 40%) can lead to a significant underestimation (Fig. S12).

53

Figure S12. Sensitivity analysis of manifold-learning and velocity-learning on the introduction of non-cycling cells in simulated data and human fibroblasts. (a) Box plots of the circular correlation coefficient between the estimated phase and simulated ground truth (G.T.) across three simulated datasets with varying proportions of non-cycling cells (from 0 to 100 non-cycling cells per 100 cycling cells). (b) Box plots of scaled velocity posterior estimates compared to the simulated G.T.

(red dashed line) across three simulated datasets with varying proportions of non-cycling cells. (c) Box plots of the circular correlation coefficient between the estimated phase with varying proportions of non-cycling cells and the phase without any non-cycling cell contaminants taken from the same dataset of human dermal fibroblasts. Contaminant cells were taken from non-cycling cell types as annotated in the original study (Capolupo et al. 2022). (d) Box plots of scaled velocity posterior estimates compared in human dermal fibroblasts with varying proportions of non-cycling cells. The box plots in (c) and (d) indicate the results across three independently sampled subsets of non-cycling cells.

We note that the Perturb-seq analysis (Fig. 8) is relevant to this question as we tackle a case that cannot be tackled by other velocity methods. In these settings, each single condition is composed of few cells, sometimes as few as 75; nonetheless, we make tractable velocity

54

estimation using transfer learning (see results section “Transfer learning of manifold parameters enables discovery of velocity alterations in genome-wide perturbation screens” and Methods 6.6)

Finally, we would like to stress that with our framework, users can gauge “after the fact” how credible velocity outputs and velocity differences are. This can be simply achieved by fitting the model and inspecting a single (not convolved and not a heuristic!) readout: the posterior probability distribution of the estimated velocity. An example of this scenario is shown in Figure R2.2, where we decided to run the mode despite the raising the Warning described above, and were confronted with a wide credible interval that does not allow us to conclude there is credible velocity information in the data.

55

References

- Aissa, Alexandre F., Abul B. M. M. K. Islam, Majd M. Ariss, Camille C. Go, Alexandra E. Rader, Ryan D. Conrardy, Alexa M. Gajda, et al. 2021. “Single-Cell Transcriptional Changes Associated with Drug Tolerance and Response to Combination Therapies in Cancer.” *Nature Communications* 12 (1): 1628.
- Aivazidis, Alexander, Fani Memi, Vitalii Kleshchevnikov, Brian Clarke, Oliver Stegle, and Omer Ali Bayraktar. 2023. “Model-Based Inference of RNA Velocity Modules Improves Cell Fate Prediction.” *bioRxiv*. <https://doi.org/10.1101/2023.08.03.551650>.
- Auerbach, Benjamin J., Garret A. FitzGerald, and Mingyao Li. 2022. “Tempo: An Unsupervised Bayesian Algorithm for Circadian Phase Inference in Single-Cell Transcriptomics.” *Nature Communications* 13 (1): 6580.
- Bastidas-Ponce, Aimée, Sophie Tritschler, Leander Dony, Katharina Scheibner, Marta Tarquis-Medina, Ciro Salinno, Silvia Schirge, et al. 2019. “Comprehensive Single Cell mRNA Profiling Reveals a Detailed Roadmap for Pancreatic Endocrinogenesis.” *Development* 146 (12). <https://doi.org/10.1242/dev.173849>.
- Battich, Nico, Joep Beumer, Buys de Barbanson, Lenno Krenning, Chloé S. Baron, Marvin E. Tanenbaum, Hans Clevers, and Alexander van Oudenaarden. 2020. “Sequencing Metabolically Labeled Transcripts in Single Cells Reveals mRNA Turnover Strategies.” *Science* 367 (6482): 1151–56.
- Bergen, Volker, Marius Lange, Stefan Peidli, F. Alexander Wolf, and Fabian J. Theis. 2020. “Generalizing RNA Velocity to Transient Cell States through Dynamical Modeling.” *Nature Biotechnology*, August. <https://doi.org/10.1038/s41587-020-0591-3>.
- Bergen, Volker, Ruslan A. Soldatov, Peter V. Kharchenko, and Fabian J. Theis. 2021. “RNA Velocity-Current Challenges and Future Perspectives.” *Molecular Systems Biology* 17 (8): e10282.
- Buettner, Florian, Kedar N. Natarajan, F. Paolo Casale, Valentina Proserpio, Antonio Scialdone, Fabian J. Theis, Sarah A. Teichmann, John C. Marioni, and Oliver Stegle. 2015. “Computational Analysis of Cell-to-Cell Heterogeneity in Single-Cell RNA-Sequencing Data Reveals Hidden Subpopulations of Cells.” *Nature Biotechnology* 33 (2): 155–60.
- Cao, Junyue, Wei Zhou, Frank Steemers, Cole Trapnell, and Jay Shendure. 2020. “Sci-Fate Characterizes the Dynamics of Gene Expression in Single Cells.” *Nature Biotechnology* 38 (8): 980–88.
- Capolupo, Laura, Irina Khven, Alex R. Lederer, Luigi Mazzeo, Galina Glousker, Sylvia Ho, Francesco Russo, et al. 2022. “Sphingolipids Control Dermal Fibroblast Heterogeneity.” *Science* 376 (6590): eabh1623.
- Chen, Ao, Sha Liao, Mengnan Cheng, Kailong Ma, Liang Wu, Yiwei Lai, Xiaojie Qiu, et al. 2022. “Spatiotemporal Transcriptomic Atlas of Mouse Organogenesis Using DNA Nanoball-Patterned Arrays.” *Cell* 185 (10): 1777–92.e21.
- Cui, Haotian, Hassaan Maan, Maria C. Vladioiu, Jiao Zhang, Michael D. Taylor, and Bo Wang.

2024. "DeepVelo: Deep Learning Extends RNA Velocity to Multi-Lineage Systems with Cell-Specific Kinetics." *Genome Biology* 25 (1).
<https://doi.org/10.1186/s13059-023-03148-9>.
Droin, Colas, Eric R. Paquet, and Felix Naef. 2019. "Low-Dimensional Dynamics of Two Coupled Biological Oscillators." *Nature Physics* 15 (10): 1086–94.
Farrell, Spencer, Madhav Mani, and Sidhartha Goyal. 2023. "Inferring Single-Cell Transcriptomic Dynamics with Structured Latent Gene Expression Dynamics." *Cell Reports Methods* 3 (9): 100581.
Gao, Mingze, Chen Qiao, and Yuanhua Huang. 2022. "UniTVelo: Temporally Unified RNA

56

Velocity Reinforces Single-Cell Trajectory Inference." *Nature Communications* 13 (1): 6586.

Gayoso, Adam, Philipp Weiler, Mohammad Lotfollahi, Dominik Klein, Justin Hong, Aaron Streets, Fabian J. Theis, and Nir Yosef. 2024. "Deep Generative Modeling of Transcriptional Dynamics for RNA Velocity Analysis in Single Cells." *Nature Methods* 21 (1): 50–59.

Gorin, Gennady, Meichen Fang, Tara Chari, and Lior Pachter. 2022. "RNA Velocity Unraveled." *PLoS Computational Biology* 18 (9): e1010492.

Gu, Y., D. Blaauw, and J. D. Welch. 2022. "Bayesian Inference of Rna Velocity from Multi-Lineage Single-Cell Data." *bioRxiv*.

<https://www.biorxiv.org/content/10.1101/2022.07.08.499381.abstract>.

Hochgerner, Hannah, Amit Zeisel, Peter Lönnerberg, and Sten Linnarsson. 2018. "Conserved Properties of Dentate Gyrus Neurogenesis across Postnatal Development Revealed by Single-Cell RNA Sequencing." *Nature Neuroscience* 21 (2): 290–99.

La Manno, Gioele, Kimberly Siletti, Alessandro Furlan, Daniel Gyllborg, Elin Vinsland, Alejandro Mossi Albiach, Christoffer Mattsson Langseth, et al. 2021. "Molecular Architecture of the Developing Mouse Brain." *Nature* 596 (7870): 92–96.

La Manno, Gioele, Ruslan Soldatov, Amit Zeisel, Emelie Braun, Hannah Hochgerner, Viktor Petukhov, Katja Lidschreiber, et al. 2018. "RNA Velocity of Single Cells." *Nature* 560 (7719): 494–98.

Levitin, Hanna Mendes, Jinzhou Yuan, Yim Ling Cheng, Francisco Ruiz Jr, Erin C. Bush, Jeffrey N. Bruce, Peter Canoll, et al. 2019. "De Novo Gene Signature Identification from Single-Cell RNA-Seq with Hierarchical Poisson Factorization." *Molecular Systems Biology* 15 (2): e8557.

Liang, Shaoheng, Fang Wang, Jincheng Han, and Ken Chen. 2020. "Latent Periodic Process Inference from Single-Cell RNA-Seq Data." *Nature Communications* 11 (1): 1441.

Li, Shengyu, Pengzhi Zhang, Weiqing Chen, Lingqun Ye, Kristopher W. Brannan, Nhat-Tu Le, Jun-Ichi Abe, John P. Cooke, and Guangyu Wang. 2023. "A Relay Velocity Model Infers Cell-Dependent RNA Velocity." *Nature Biotechnology*, April, 1–10.

Mages, Simon, Noa Moriel, Inbal Avraham-Davidi, Evan Murray, Jan Watter, Fei Chen, Orit Rozenblatt-Rosen, Johanna Klughammer, Aviv Regev, and Mor Nitzan. 2023. "TACCO Unifies Annotation Transfer and Decomposition of Cell Identities for Single-Cell and Spatial Omics." *Nature Biotechnology* 41 (10): 1465–73.

Piran, Zoe, Niv Cohen, Yedid Hoshen, and Mor Nitzan. 2024. "Disentanglement of Single-Cell Data with Biolord." *Nature Biotechnology*, January.

<https://doi.org/10.1038/s41587-023-02079-x>.

Piran, Zoe, and Mor Nitzan. 2024. "SiFT: Uncovering Hidden Biological Processes by Probabilistic Filtering of Single-Cell Data." *Nature Communications* 15 (1): 760.

Qiao, Chen, and Yuanhua Huang. 2021. "Representation Learning of RNA Velocity Reveals Robust Cell Transitions." *Proceedings of the National Academy of Sciences of the United States of America* 118 (49). <https://doi.org/10.1073/pnas.2105859118>.

Qiu, Xiaojie, Yan Zhang, Jorge D. Martin-Rufino, Chen Weng, Shayan Hosseinzadeh, Dian Yang, Angela N. Pogson, et al. 2022. "Mapping Transcriptomic Vector Fields of Single Cells." *Cell* 185 (4): 690–711.e45.

Rand, David A., Archishman Raju, Meritxell Sáez, Francis Corson, and Eric D. Siggia. 2021. "Geometry of Gene Regulatory Dynamics." *Proceedings of the National Academy of Sciences of the United States of America* 118 (38).

<https://doi.org/10.1073/pnas.2109729118>.

Replogle, Joseph M., Reuben A. Saunders, Angela N. Pogson, Jeffrey A. Hussmann, Alexander Lenail, Alina Guna, Lauren Mascibroda, et al. 2022. "Mapping Information-Rich Genotype-Phenotype Landscapes with Genome-Scale Perturb-Seq."

57

Cell 185 (14): 2559–75.e28.

Riba, Andrea, Attila Oravec, Matej Durik, Sara Jiménez, Violaine Alunni, Marie Cerciat, Matthieu Jung, Céline Keime, William M. Keyes, and Nacho Molina. 2022. "Cell Cycle

Gene Regulation Dynamics Revealed by RNA Velocity and Deep-Learning." Nature Communications 13 (1): 2865.

Talamanca, Lorenzo, Cédric Gobet, and Felix Naef. 2023. "Sex-Dimorphic and Age-Dependent Organization of 24-Hour Gene Expression Rhythms in Humans." Science 379 (6631): 478–83.

Zhang, Yan, Xiaojie Qiu, Ke Ni, Jonathan Weissman, Ivet Bahar, and Jianhua Xing. 2023. "Graph-Dynamo: Learning Stochastic Cellular State Transition Dynamics from Single Cell Data." bioRxiv. <https://doi.org/10.1101/2023.09.24.559170>.

58

Version 1:

Decision Letter:

Our ref: NMETH-A54818A

21st Jun 2024

Dear Gioele,

Thank you for submitting your revised manuscript "Statistical inference with a manifold-constrained RNA velocity model uncovers cell cycle speed modulations" (NMETH-A54818A). It has now been seen by the original referees and their comments are below. The reviewers find that the paper has improved in revision, and therefore we'll be happy in principle to publish it in Nature Methods, pending minor revisions to satisfy the referees' final requests (if any) and to comply with our editorial and formatting guidelines.

TRANSPARENT PEER REVIEW

Please note: we allow redactions to authors' rebuttal and reviewer comments in the interest of confidentiality. If you are concerned about the release of confidential data, please let us know specifically what information you would like to have removed. Please note that we cannot incorporate redactions for any other reasons. Reviewer names will be published in the peer review files if the reviewer signed the comments to authors, or if reviewers explicitly agree to release their name. For more information, please refer to our <https://www.nature.com/documents/nr-transparent-peer-review.pdf> target="new">FAQ page.

ORCID

Sincerely,
Madhura

Madhura Mukhopadhyay, PhD
Senior Editor
Nature Methods

Reviewer #1 (Remarks to the Author):

Authors addressed all my concerns. I do not have more questions.

Reviewer #1 (Remarks on code availability):

The code was well written.

Reviewer #2 (Remarks to the Author):

The author did a good job in addressing all my concerns and I don't have further questions.

Reviewer #2 (Remarks on code availability):

I suggest the author to develop a readthedocs page for their package. In addition, they should release the package to PyPi or bioconda.

Reviewer #3 (Remarks to the Author):

The authors have adequately and thoroughly addressed all of my comments, and the manuscript is greatly improved. Overall, I believe this manuscript to be an important contribution to the field.

Reviewer #3 (Remarks on code availability):

As I mentioned in my original review, the code is organized, documented, and includes multiple tutorials.

Version 2:

Decision Letter:

15th Sep 2024

Dear Gioele,

I am pleased to inform you that your Article, "Statistical inference with a manifold-constrained RNA velocity model uncovers cell cycle speed modulations", has now been accepted for publication in Nature Methods. The received and accepted dates will be 18th Dec 2023 and 15th Sep 2024. This note is intended to let you know what to expect from us over the next month or so, and to let you know where to address any further questions.

Over the next few weeks, your paper will be copyedited to ensure that it conforms to Nature Methods style. Once your paper is typeset, you will receive an email with a link to choose the appropriate publishing options for your paper and our Author Services team will be in touch regarding any additional information that may be required. It is extremely important that you let us know now whether you will be difficult to contact over the next month. If this is the case, we ask that you send us the contact information (email, phone and fax) of someone who will be able to check the proofs and deal with any last-minute problems.

Please note that *Nature Methods* is a Transformative Journal (TJ). Authors may publish their research with us through the traditional subscription access route or make their paper immediately open access through payment of an article-processing charge (APC). Authors will not be required to make a final decision about access to their article until it has been accepted. [Find out more about Transformative Journals](https://www.springernature.com/gp/open-research/transformative-journals)

If you have posted a preprint on any preprint server, please ensure that the preprint details are updated with a publication

reference, including the DOI and a URL to the published version of the article on the journal website.

If you are active on Twitter/X, please e-mail me your and your coauthors' handles so that we may tag you when the paper is published.

Best regards,
Madhura

Madhura Mukhopadhyay, PhD
Senior Editor
Nature Methods

** Visit the Springer Nature Editorial and Publishing website at http://editorial-jobs.springernature.com?utm_source=ejP_NMeth_email&utm_medium=ejP_NMeth_email&utm_campaign=ejp_Nmeth for more information about our career opportunities. If you have any questions please click [here](mailto:editorial.publishing.jobs@springernature.com).**
